# A visual–omics foundation model to bridge histopathology with spatial transcriptomics

**Weiqing Chen**[1,2,12], **Pengzhi Zhang**[1,3,4,5,12], **Tu N Tran**[1,3,4,5], **Yiwei Xiao**[1,3,4,5], **Shengyu Li**[1,3,4,5], **Vrutant V. Shah**[4], **Hao Cheng**[6], **Kristopher W. Brannan**[4], **Keith Youker**[3,5], **Li Lai**[3,5], **Longhou Fang**[3,5], **Yu Yang**[7], **Nhat-Tu Le**[3,5], **Jun-ichi Abe**[8], **Shu-Hsia Chen**[9], **Qin Ma**[6], **Ken Chen**[10], **Qianqian Song**[11], **John P. Cooke**[2,3,4,5], **Guangyu Wang**[1,3,4,5,✉]

[1]Center for Bioinformatics and Computational Biology, Houston Methodist Research Institute, Houston, TX, USA.

[2]Department of Physiology, Biophysics & Systems Biology, Weill Cornell Graduate School of Medical Science, Cornell University, New York, NY, USA.

[3]Center for Cardiovascular Regeneration, Houston Methodist Research Institute, Houston, TX, USA.

[4]Center for RNA Therapeutics, Houston Methodist Research Institute, Houston, TX, USA.

[5]Department of Cardiothoracic Surgery, Weill Cornell Medicine, Cornell University, New York, NY, USA.

[6]Department of Biomedical Informatics, College of Medicine, The Ohio State University, Columbus, OH, USA.

✉ **Correspondence and requests for materials** should be addressed to Guangyu Wang. gwang2@houstonmethodist.org.

Author contributions
G.W. supervised the study. W.C. and G.W. designed and developed the visual–omics foundation model and platform. W.C., P.Z., Y.X., T.T. and H.C. analyzed the data. G.W., W.C. and P.Z. wrote the manuscript. V.V.S., K.W.B., L.L., K.Y. and L.F. provided in-house patient tissues. K.Y. and Y.Y. annotated the pathology images. J.P.C., K.W.B., L.L., K.Y., L.F., Q.S., Q.M., K.C., N.-T.L., J.-i.A., Y.X., S.-H.C. and S.L. were involved in the discussion and helped improve the manuscript. All authors approved the final version of the manuscript.

Competing interests
The authors declare no competing interests.

Code availability
Loki is implemented in Python and is available via https://github.com/GuangyuWangLab2021/Loki/. The pretrained OmiCLIP weights are available via https://huggingface.co/WangGuangyuLab/Loki/.

**Extended data** is available for this paper at https://doi.org/10.1038/s41592-025-02707-1.

Reporting summary
Further information on research design is available in the Nature Portfolio Reporting Summary linked to this article.

[7]Department of Pathology, Immunology and Laboratory Medicine, College of Medicine, University of Florida, Gainesville, FL, USA.

[8]Department of Cardiology, The University of Texas MD Anderson Cancer Center, Houston, TX, USA.

[9]Center for Immunotherapy, Neal Cancer Center, Houston Methodist Research Institute, Houston, TX, USA.

[10]Department of Bioinformatics and Computational Biology, The University of Texas MD Anderson Cancer Center, Houston, TX, USA.

[11]Department of Health Outcomes and Biomedical Informatics, College of Medicine, University of Florida, Gainesville, FL, USA.

[12]These authors contributed equally: Weiqing Chen, Pengzhi Zhang.

## Abstract

Artificial intelligence has revolutionized computational biology. Recent developments in omics technologies, including single-cell RNA sequencing and spatial transcriptomics, provide detailed genomic data alongside tissue histology. However, current computational models focus on either omics or image analysis, lacking their integration. To address this, we developed OmiCLIP, a visual–omics foundation model linking hematoxylin and eosin images and transcriptomics using tissue patches from Visium data. We transformed transcriptomic data into 'sentences' by concatenating top-expressed gene symbols from each patch. We curated a dataset of 2.2 million paired tissue images and transcriptomic data across 32 organs to train OmiCLIP integrating histology and transcriptomics. Building on OmiCLIP, our Loki platform offers five key functions: tissue alignment, annotation via bulk RNA sequencing or marker genes, cell-type decomposition, image–transcriptomics retrieval and spatial transcriptomics gene expression prediction from hematoxylin and eosin-stained images. Compared with 22 state-of-the-art models on 5 simulations, and 19 public and 4 in-house experimental datasets, Loki demonstrated consistent accuracy and robustness.

Computational biology has advanced notably with artificial intelligence (AI) for tasks such as gene expression enhancement, single-cell perturbation prediction, tissue annotation, diagnosis, primary tumor origin predictions and image retrieval from hematoxylin and eosin (H&E)-stained images[1–7]. Recently, foundation models like CLIP[8], CoCa[9] and DeCLIP[10] have been adapted to the field, fine-tuned with pathology images and captions, as seen in PLIP and CONCH[11,12]. These visual–language foundation models support applications like text-to-image and image-to-text retrieval, histology image classification, captioning and diagnosis improvement.

Omics data, including transcriptomics and genetics, provide crucial insights into cell types in health and disease, enhancing our understanding of cellular heterogeneity, lineage tracing and disease mechanisms[13–22]. Combining omics data with histology images offers complementary information for both research and clinical applications, and has been used for predicting cancer outcomes, prognosis and response to neoadjuvant chemotherapy[3]. However, existing methods remain task specific and lack a unified multimodal AI model

to integrate histology and omics data. Additionally, challenges remain in developing infrastructure to efficiently analyze sequencing data and pathology images together.

To address these gaps, we introduce omics and image pretraining, OmiCLIP, a transcriptomic–image dual-encoder foundation model and Loki platform, an infrastructure of multimodal analysis using OmiCLIP as a backbone. To train OmiCLIP, we curated the ST-bank dataset with 2.2 million tissue patches from 1,007 samples across 32 organs with paired whole-slide images (WSIs) and 10x Visium spatial transcriptomics (ST) data. Inspired by large language model-based single-cell models like GenePT[23] and Cell2Sentence[24], we represented transcriptomics of a tissue patch by a 'sentence' of top-ranking highly expressed genes, separated by spaces (' '). Using this large-scale set of transcriptomics–histology image pairs, we trained the CLIP-based foundation model, integrating both genomic and image data. Building upon OmiCLIP, the Loki platform offers five core functions: tissue alignment, tissue annotation, cell-type decomposition, image–transcriptomics retrieval and ST gene expression prediction (Fig. 1). Loki provides several distinctive features, including aligning H&E images with ST data, annotating tissue H&E images based on bulk RNA sequencing (RNA-seq) or marker genes and decomposing cell types from H&E images with reference to single-cell RNA sequencing (scRNA-seq). We evaluated Loki's functions against 22 state-of-the-art (SOTA) methods on 5 simulation datasets, 19 publicly available experimental datasets and 4 in-house experimental datasets, showing Loki's consistent accuracy and robustness across tasks. We also investigated OmiCLIP's embeddings for clustering and annotating scRNA-seq data and predicting The Cancer Genome Atlas (TCGA) participants' risk levels (Supplementary Notes 1 and 2).

## Results

### Loki platform powered by contrastive-aligned visual–omics

Transcriptomics provides insights into cellular diversity within tissues, making it a natural indicator of tissue diversity[25]. ST technologies bridge histopathology images and transcriptomics data, enabling the development of a foundation model that integrates both. We introduce OmiCLIP, a visual–transcriptomics foundation model trained on ST-bank, which includes diverse histopathology images and over 2.2 million paired transcriptomics from 113 studies (Fig. 1a–c and Supplementary Table 1). ST-bank covers 32 organ types, including conditions like health, cancer, heart failure and Alzheimer's disease (Fig. 1b,c). We applied a quality-control pipeline to retain ST data with high-resolution H&E images. As the batch effects may strongly affect the generalization ability of the model, the adopted rank-based strategies inspired by recent single-cell foundation models such as GeneFormer[26] and scFoundation[27] successfully eliminate batch effects through rank-based approaches rather than relying directly on raw read counts or normalized gene expression values. Specifically, we standardized text descriptions of the associated images by converting all Ensembl gene IDs to gene symbols and removing housekeeping genes. To format transcriptomics for language models, genes symbols were ranked from high to low by expression levels and structured into sentences for the text encoder (Fig. 1a).

OmiCLIP was fine-tuned using CoCa[9], a SOTA visual–language foundation framework, comprising an image encoder, a text encoder and a multimodal fusion decoder. The image

and transcriptomics modalities were aligned in a common representation space utilizing contrastive learning (Fig. 1a and Extended Data Figs. 1 and 2). In this dual-modality space, paired image and transcriptomic embedding vectors were optimized to be similar.

To evaluate OmiCLIP's reliability to image quality variability across samples due to technological limitations, we simulated low-quality H&E images by adding Gaussian noise and compared the similarity scores between the paired transcriptomic and original image embeddings, with paired transcriptomic and simulated low-quality image embeddings, which were encoded by OmiCLIP's image and transcriptomic encoders. PLIP and OpenAI CLIP served as benchmarks (Extended Data Fig. 3a,b), and results demonstrated that OmiCLIP is robust to variations in image quality.

For sequencing depth variability across technologies, we first analyzed the sequencing depth ranges in ST-bank and categorized samples into high, medium and low sequencing depth groups, identified as 11,792 unique molecular identifier (UMI) counts, 4,512 UMI counts and 615 UMI counts, respectively. Second, we generated low sequencing depth ST simulations using the downsampling function implemented in scuttle[28]. We evaluated transitions from high-to-medium sequencing depth, medium-to-low sequencing depth and high-to-low sequencing depth. We compared similarity scores between paired images and original transcriptomic embeddings, with paired images and downsampled transcriptomic embeddings. These embeddings were encoded using OmiCLIP's image and transcriptomic encoders, using PLIP and OpenAI CLIP as benchmarks (Extended Data Fig. 3c). Results demonstrated OmiCLIP's robustness across sequencing depths, highlighting its adaptability to datasets generated across different technologies.

The key advantage of contrastive-aligned visual–transcriptomics pretraining is its unique capability to drive the development of cross-modality tissue analysis tools. As a proof of concept, we developed Loki, a unified AI platform for multimodal analysis. In Loki, five modules were implemented, including Loki Align for multi-section tissue alignment, Loki Annotate for multimodal tissue annotation, Loki Decompose for cell-type decomposition from transcriptomics or histology, Loki Retrieve for histology image–transcriptomics retrieval and Loki PredEx for ST gene expression prediction from histology images (Fig. 1b). While these initial modules demonstrate its potential, Loki is designed to expand, supporting the development of more tools to further enhance multimodal tissue reconstruction and analysis. Loki could serve as the infrastructure that efficiently transfers transcriptomics such as scRNA-seq, bulk RNA-seq data and even marker genes into pathology image analysis via the pretrained model (OmiCLIP) (Fig. 1d), streamlining workflows, accelerating analysis and minimizing sequencing cost in research areas such as three-dimensional (3D) tissue studies and pathology diagnosis.

### OmiCLIP improves image and transcriptomics representations

OmiCLIP's image embeddings capture the morphology of tissues, while its transcriptomic embeddings represent genomic characteristics. Since OmiCLIP includes both transcriptomics and image encoders, here we evaluated whether contrastive learning enhances the ability of each encoder to represent tissue types better than the initial encoders. To assess clustering performance, we moved beyond qualitative visualizations and

introduced quantitative metrics to assess the quality of the clustering. The uniform manifold approximation and projection (UMAP) visualizations showed that both embeddings clustered similar tissue types (Extended Data Fig. 2); however, the results were limited in their ability to quantify clustering quality and may have appeared unstable in some cases. Therefore, we computed the Calinski–Harabasz (CH) score[29], a widely used clustering validation metric, which balances the dispersion between clusters with the cohesion within clusters (Methods). Higher CH scores reflect better clustering performance by indicating more distinct and internally consistent clusters.

First, we calculated CH scores across 95 tissue samples from the ST-bank dataset, which included expert-annotated cell types from breast, healthy heart, kidney cancer and lung tissues and heart tissue with myocardial infarction (Supplementary Table 2). These annotated cell types served as ground-truth cluster labels. Our results showed a significant increase ($P$ value < 0.001; Extended Data Fig. 1) in CH scores for embeddings after contrastive learning compared to before, demonstrating improved clustering performance.

Second, we expanded the CH score calculations to the rest of the ST-bank samples, where no cell-type annotations are directly available. For these samples, the clusters were identified by the Leiden algorithm on the ST (Methods). After contrastive learning, CH scores significantly increased in all organ types ($P$ value < 0.05; Extended Data Fig. 2). OmiCLIP's image embeddings also outperformed SOTA models like UNI[7] and GigaPath[30] by aligning image and transcriptomic data, not just image–image interactions. The results demonstrated OmiCLIP's ability to capture tissue heterogeneity.

## Loki Align aligns ST-to-ST and H&E image-to-ST data

Researchers recently began investigating spatial biology in 3D, revealing new insights into tissue organization and cellular interactions. This requires tools to align multiple H&E images or ST sections, and even cross-align H&E images with ST slides. However, spatial distortions and biological variations between sections make alignment challenging. To address this, we developed the module Loki Align to align ST-to-ST data, H&E image-to-H&E image, and H&E image-to-ST data. Loki Align first embeds patch-level transcriptomics or H&E images into a 768-dimension space using OmiCLIP, and then applies the adapted coherent point drift (CPD) method[31] to align two embeddings, preserving probability distribution and topology (Fig. 2a and Methods). We evaluated Loki Align on four datasets including two simulation datasets, a set of eight adjacent small intestine tissue sections, and a set of two adjacent ovarian carcinosarcoma sections. To ensure compatibility with datasets that may not be represented in the ST-bank, we used fine-tuning as a default setting for the Loki Align in the alignment tasks. Fine-tuning minimized contrastive loss between image embeddings and the paired text embeddings of the top-expressed gene name sentence (Methods). We further evaluated the zero-shot performance on an ovarian carcinosarcoma dataset.

First, we simulated paired H&E images and ST data by perturbing gene expression and spatial locations with varying noise levels, covering diverse tissue types and disease types (Methods). We measured the distance between Loki-aligned data and the ground truth, and compared Loki Align with PASTE and GPSA, which are designed for ST section

alignment[32,33]. At both high and low noise levels, Loki ST-to-ST alignment and Loki image-to-ST alignment ranked first and second, respectively, among the four methods (Fig. 2b), significantly outperforming PASTE and GPSA ($P$ values < 0.001, Wilcoxon test). This superiority likely stems from PASTE's design for linear transformations, which maintains topological integrity but struggles with spatial warping[32], while GPSA aims to map readouts to a common coordinate system, risking topological fidelity[33].

Second, we tested Loki Align on eight adjacent human small intestine tissues sections[34]. Real-world datasets often present challenges due to distortions such as rotation, tilt, uneven slicing and missing fragments. For better performance, we fine-tuned OmiCLIP using the target slide's H&E image and ST data. We aligned seven source ST datasets to target ST data and seven source H&E images to target ST data using Loki Align and applied PASTE and GPSA to align seven source ST datasets to target ST data. Loki Align successfully aligned all source sections to the target section. To evaluate the performance, we calculated the Pearson correlation coefficient (PCC) and Kendall's tau coefficient. For ST-to-ST scenarios, we compared the aligned ST data and the target ST data. For image-to-ST scenarios, after aligning the H&E image to the target ST dataset, we compared the paired ST data corresponding to the H&E image with the target ST dataset. The median PCC for Loki's image-to-ST and ST-to-ST alignment ranged from 0.67 to 0.80 and 0.62 to 0.83, respectively (Fig. 2c). The median Kendall's tau coefficient ranged from 0.16 to 0.27 for Loki's image-to-ST and 0.18 to 0.27 for ST-to-ST alignment (Supplementary Fig. 1a). On the vertical plane, Loki correctly aligned the same tissue types by image-to-ST and ST-to-ST alignment, while PASTE and GPSA twisted the tissues. PASTE rotated three source sections (sources 1–3; Fig. 2c) and the PCC and Kendall's tau coefficient ranged from −0.25 to 0.39 and −0.06 to 0.13, respectively. GPSA found common coordinates in six of the seven slices but introduced tremendous distortions, resulting in a PCC of 0.27 to 0.56 and Kendall's tau coefficient of 0.06 to 0.13. Overall, Loki ST-to-ST and image-to-ST alignments outperformed the SOTA methods. To isolate the contributions of OmiCLIP embeddings versus the superior registration method (CPD), we applied CPD to both OmiCLIP embeddings and transcriptomic embeddings that was reduced to two principal components using principal component analysis (PCA; Fig. 2c). OmiCLIP embeddings significantly improved the performance of alignment compared to PCA embeddings ($P$ value < 0.001, Wilcoxon test).

Third, we assessed Loki Align's performance on two adjacent human ovarian carcinosarcoma sections[35] (Fig. 2d). With fine-tuning, Loki's ST-to-ST and image-to-ST achieved the best performance, with median PCCs of 0.88 and 0.86, and Kendall's tau coefficients of 0.21 and 0.18, respectively. PASTE, GPSA and CAST[36] had median PCCs of 0.26, 0.43 and 0.71 and median Kendall's tau coefficients of 0.03, 0.04 and 0.09, respectively ($P$ value < 0.01; Fig. 2e and Supplementary Fig. 1b). The spatial expression patterns of representative genes are shown in Supplementary Fig. 2.

Fourth, we evaluated Loki Align on a human breast cancer dataset[37] with paired 10x Visium and Xenium slides (Extended Data Fig. 4). We generated simulation data by performing rotation and translation of Xenium data. To perform the alignment, we first calculated transcriptomic embeddings for the Visium slide using gene sentences derived from Visium

transcriptomic data. For the Xenium slide, we created pseudo-Visium data by averaging gene expression values across pseudo-spots. These pseudo-Visium data were then used to calculate transcriptomic embeddings via the transcriptomic encoder of OmiCLIP. Finally, Loki Align was applied to align the transcriptomic embeddings of the Xenium slide with those of the Visium slide, with performance measured by the mean distance between the aligned and target spots. The resulting distance between the aligned Xenium slide and the target Visium slide was 0.08 mm, demonstrating that Loki Align effectively aligns Visium and Xenium slides with high precision.

Fifth, we evaluated the performance of three training strategies: pretraining plus fine-tuning, pure pretraining and pure training from scratch on ovarian carcinosarcoma samples (Supplementary Fig. 3). The best performance was achieved with pretraining plus fine-tuning, resulting in a median PCC of 0.86 and a Kendall's tau coefficient of 0.17. Pure pretraining showed comparable performance, with a median PCC of 0.85 and a Kendall's tau coefficient of 0.18. In contrast, training from scratch exhibited the lowest performance, with a median PCC of 0.53 and a Kendall's tau coefficient of 0.06. Overall, we recommend fine-tuning as a default setting for Loki Align, as it ensures compatibility with datasets underrepresented in the ST-bank.

Lastly, we examined whether Loki Align could leverage both modalities simultaneously for alignment over a single modality. To evaluate this, we integrated image embeddings and transcriptomic embeddings by averaging them and used the combined embeddings to align two adjacent ovarian carcinosarcoma samples. We then calculated the PCC and Kendall's tau coefficient for the image embeddings, transcriptomic embeddings and averaged embeddings to assess performance (Supplementary Fig. 4). The results indicated that the averaged embeddings did not outperform single-modality embeddings. Altogether, by addressing spatial distortions and biological variability, Loki Align enables the accurate alignment of multiple H&E images and ST sections, thereby supporting advanced 3D reconstructions of tissue organization, particularly for cross-modality studies that combine H&E images and ST data.

## Loki Annotate deciphers H&E images with bulk RNA-seq data

Next, we evaluated Loki's capability to analyze H&E images using bulk RNA-seq data, which is commonly used in both basic research and clinical practice. During OmiCLIP pretraining, the cosine similarities between paired ST and histology images were maximized, allowing the similarity between the H&E image of tissue patches and tissue-type-specific bulk RNA-seq data to indicate tissue-type enrichment. We developed Loki Annotate to annotate H&E images using tissue-type-specific bulk RNA-seq data as a reference. We used OmiCLIP to encode tissue patches from a WSI and the tissue-specific bulk RNA-seq data, then calculated the cosine similarity between the encoded embeddings (Fig. 3a). Higher similarity values indicate greater presence of the tissue type.

We evaluated Loki Annotate on breast cancer, normal breast, and heart failure tissues. In three breast cancer tissues, H&E regions corresponding to tumor tissue showed high similarity with the bulk RNA-seq data from tumor biopsies, which include tumor-related markers such as *COL1A1* (ref. 38) and *ACTB*[39] (Fig. 3b and Supplementary Fig. 5).

Similarity scores within the tumor regions were significantly higher than those outside ($P$ value < 0.05, Wilcoxon test). Additionally, higher similarity scores were consistent with higher diagnostic values of tumors calculated by clustering-constrained-attention multiple-instance learning[40] (CLAM, a SOTA WSI tumor analysis model; Fig. 3b). Next, we tested the similarity between H&E images of heart failure tissues and fibroblast RNA-seq data, as well as between H&E images of normal breast tissues and adipose RNA-seq data. The similarity scores in the corresponding pathology annotated regions were remarkably higher than the non-corresponding regions (Fig. 3b and Supplementary Fig. 5). In summary, Loki Annotate effectively annotates H&E images by using tissue-type-specific bulk RNA-seq data as a reference.

### Loki Annotate annotates H&E images based on marker genes

When bulk RNA-seq is unavailable, Loki Annotate can also annotate tissues using predefined marker genes, similar to the workflow of using bulk RNA-seq data without fine-tuning. We created tissue-specific gene lists using well-established markers, such as 'TP53, EPCAM, KRAS, …, DSP' for tumors (Fig. 4a and Supplementary Table 3). As with the bulk RNA-seq approach, we used OmiCLIP to encode tissue patches from histology images and the gene name sentence composed from the marker gene list. We applied Loki Annotate to four benchmark histopathology datasets including CRC7K[41] (eight tissue types), WSSS-4LUAD[42] (normal and tumor), PatchCamelyon[43] (normal and tumor) and LC25000[44] (benign and malignant). Tissue-type annotation was determined by cosine similarity derived from the dot product of normalized text embeddings and H&E image embeddings, with the highest cosine similarity score assigned as the predicted tissue to the query image. Based on these annotations, precision was defined as the proportion of correctly predicted tissues (true positives) of all predicted tissues, while recall was defined as the proportion of correctly predicted tissues of all actual tissues. The F1 score was calculated as the harmonic mean of precision and recall, which was used to measure classification performance. We measured annotation performance using F1 score and compared our results to the OpenAI CLIP model. Our analysis showed that Loki consistently outperformed OpenAI CLIP across all four datasets (Fig. 4b,c). The F1 scores of Loki ranged from 0.59 to 0.96, while the F1 scores of OpenAI CLIP ranged from 0.03 to 0.34 (Fig. 4c).

Several studies have developed visual–language foundation models using paired histopathology images and captions[11,12]. Given that transcriptomics and natural language provide complementary information, we investigated whether their combination could improve annotation performance without additional training. We applied PLIP, a visual–language foundation model for pathology image analysis, to annotate the tissue images by descriptive prompts, such as converting 'tumor' to 'an H&E image patch of colorectal adenocarcinoma epithelium' in the CRC7K dataset. Overall, PLIP performed comparably to Loki, with F1 scores ranging from 0.5 to 0.93 (Fig. 4d). We then combined Loki and PLIP by averaging their similarity scores of an H&E image and a given tissue type (Fig. 4a and Methods), resulting in the best performance across all four benchmark datasets (Fig. 4d,e). In CRC7K, PLIP misclassified 63% of colorectal adenocarcinoma epithelium images as cancer-associated stroma, while Loki misclassified 15% of tumor images as normal colon mucosa. Notably, combining Loki and PLIP achieved a 93% recall rate, demonstrating that

combining transcriptomic and natural language enhances overall performance compared to each modality alone (Fig. 4f).

### Loki Decompose maps cell types in H&E image using scRNA-seq

Since OmiCLIP can project the Visium ST data and H&E images to a shared embedding space, we developed Loki Decompose, a feature to decompose cell types in both ST data and H&E images, using scRNA-seq as a reference. Inspired by ST decomposition models like Tangram and CytoSPACE[45,46], we used OmiCLIP to encode the patches (the same size as a Visium spot) of an H&E image and scRNA-seq transcriptomic profile into this embedding space. As an application of Tangram with OmiCLIP embeddings instead of gene expression data, Loki Decompose applied Tangram's nonconvex optimization algorithm[47] to deconvolute the OmiCLIP embeddings of an H&E image patch or the embeddings of a Visium spot's transcriptomic profile rather than raw gene expression data, providing the cell-type composition of an image patch or a Visium spot (Fig. 5a). We assessed Loki Decompose on our in-house triple-negative breast cancer (TNBC) dataset, a human colorectal cancer dataset[48] and a brain dataset[49,50].

First, we performed a Xenium experiment on the in-house TNBC sample and captured paired H&E images. We generated pseudo-Visium data from the Xenium data as a benchmark for evaluating Loki Decompose, using publicly available scRNA-seq data as a ref. 51. The Xenium data classified tissue into three main cell types: cancer epithelial cells, immune cells and stromal cells (Fig. 5b and Extended Data Fig. 5a,b). We used Loki to decompose pseudo-Visium spots and H&E images, using paired sequencing and image data from one-fourth of a WSI for fine-tuning followed by cross-validation (Methods). Decomposition accuracy was evaluated using Jensen–Shannon (JS) divergence and the structural similarity index measure (SSIM). These metrics were calculated by comparing the predicted cell-type proportions to the ground truth derived from the Xenium data. Since JS divergence and SSIM operate on different scales, we standardized their values by calculating $z$-scores among different methods (details in Methods). The $z$-score for JS divergence was inverted (that is, multiplied by −1), as lower values indicate better performance. Finally, we averaged the $z$-scores of JS divergence and SSIM to calculate an overall impact score, which provides a unified metric for comparison across methods. Loki Decompose in ST mode and image mode ranked as the top two methods with impact scores of 1.32 and 1.11, respectively, outperforming other SOTA methods[52] including Tangram, Spatial Seurat[53], CARD[54], CytoSPACE, Cell2location[50], SpatialDWLS[55] and RCTD[56], with impact scores ranging from 0.87 to −1.82 (Fig. 5c,d and Extended Data Fig. 5c). As single-cell foundation models such as GeneFormer[26], scGPT[57] and scFoundation[27] can also provide the transcriptomic embeddings, to further evaluate the approach, we replaced OmiCLIP gene expression embeddings with those from single-cell foundation models GeneFormer, scGPT and scFoundation. Results showed that scGPT, scFoundation and GeneFormer ranked 6th, 8th and 9th, respectively (Fig. 5c and Extended Data Fig. 5c).

Second, we evaluated Loki Decompose using pseudo-Visium data generated from whole-genome sequencing Visium-HD data of human colorectal cancer as a benchmark (Fig. 5e). We fine-tuned OmiCLIP on regions with paired sequencing and image data (Methods).

Remarkably, the transcriptomic embeddings for scRNA-seq data effectively captured cell heterogeneity, even without training on scRNA-seq data (Extended Data Fig. 6a and Supplementary Note 1). Loki Decompose successfully predicted the spatial distribution of key cell types (Extended Data Fig. 6b). We developed a technique inspired by non-maximum suppression (NMS)[58] to refine spatial probabilistic maps, enhancing decomposition performance by reducing ambiguity in complex spatial scenarios and focusing predictions on the most confident cell-type assignments. Using JS divergence and SSIM scores, Loki Decompose based on either the ST data or the H&E images was comparable to Tangram, which used gene expression as input (Fig. 5f).

Third, we extended the analysis to the entire WSI (20 mm) of the same human colorectal cancer tissue (Fig. 5g), segmenting it into image patches matching Visium spot size. Similarly, we used OmiCLIP to encode image patches and transcriptomics of scRNA-seq and then decomposed those using scRNA-seq data. Loki Decompose accurately predicted densities of tumor, fibroblast, intestinal epithelial, smooth muscle, immune and inflammatory cells, aligning closely with pathology annotations (Fig. 5g). Additionally, our predicted tumor cell density matched that of CLAM[40], further validating Loki Decompose's robustness (Fig. 5g).

Fourth, to test Loki Decompose in a more challenging scenario, we applied it to a brain tissue, where neurons share similar morphology. Our dataset included vascular and leptomeningeal cells (VLMCs), astrocytes, and neurons from layers 2/3 (L2/3), layers 4/5 (L4/5) and layer 6 (L6), as well as oligodendrocytes (Fig. 5h and Supplementary Fig. 6). VLMCs and astrocytes are concentrated near the cortical surface and pial borders (for example, layer 1), while oligodendrocytes are more prevalent in deeper layers and within white matter tracts[49]. To decompose the mouse brain cortex slice, we applied a workflow similar to the one for other decomposition tasks. First, we fine-tuned OmiCLIP using adjacent Visium data and H&E images, then segmented the WSI into patches, corresponding to Visium spot size. The transcriptomic encoder of OmiCLIP was used to encode the scRNA-seq data from the Allen Institute atlas[49], while the image encoder was used to encode the H&E image. Finally, Loki Decompose was applied to predict cell-type distributions within the brain cortex H&E image. Loki Decompose accurately predicted the distribution of VLMCs, astrocytes, neurons from L2/3, L4/5 and L6 and oligodendrocytes, aligning closely with brain anatomic ref. 49.

Lastly, we tested the performance of decomposition using three training strategies: pretraining plus fine-tuning, pure pretraining and pure training from scratch on TNBC samples (Extended Data Fig. 7). The analysis showed that pretraining plus fine-tuning had the best performance, achieving a mean SSIM score of 0.30 and a mean JS divergence of 0.40. In contrast, pure pretraining resulted in a mean SSIM score of 0.13 and a mean JS divergence of 0.43, while pure training from scratch performed the worst, with a mean SSIM score of 0.00070 and a mean JS divergence of 0.44. Although pure pretraining achieved a comparable JS divergence score to the pretraining plus fine-tuning method (0.43 versus 0.40), it showed a notable decline in the SSIM (0.13 versus 0.30), underscoring the importance of fine-tuning for optimal performance. Therefore, we strongly recommend fine-tuning the model for this task to achieve optimal results.

Altogether, Loki Decompose effectively inferred cell-type fractions from H&E images and ST data, demonstrating its potential to enhance spatial tissue analysis by utilizing H&E images to reduce experimental costs and processing time, particularly in multi-section tissue studies.

### Loki Retrieve enables H&E image-to-transcriptomics retrieval

One of the basic functions of contrastive learning models is retrieval. Leveraging such ability of OmiCLIP, we developed Loki Retrieve to identify and retrieve transcriptomics data corresponding to a given H&E image. Using OmiCLIP's image encoder, query images were encoded to embeddings to retrieve the most similar transcriptomic entries from the ST-bank dataset in the aligned latent space (Fig. 6a). We presented the top 50 most similar transcriptomics results, as demonstrated by the ST-paired images from the ST-bank dataset (Fig. 6b). Then, we systematically evaluated our model on diverse datasets including four independent histopathology datasets of colorectal cancer, lung cancer and lymph node metastasis, along with eight in-house tissues of heart failure, Alzheimer's disease and breast cancer human tissues (Supplementary Fig. 7). Because ground-truth transcriptomics data were unavailable, retrieval accuracy was assessed by measuring similarity between the query image and the retrieved transcriptomics-paired images. Overall, Loki Retrieve significantly outperformed OpenAI CLIP and PLIP by a large margin (Fig. 6c,d; $P$ value < 0.05), achieving median similarity scores ranging from 0.7 to 0.9.

We further evaluated image-to-transcriptomics retrieval performance by calculating the rank of the correct pair using Recall@K (5% and 10%). This metric measures the proportion of correctly retrieved data within the samples retrieved using the top-K quantile (Methods). We used four reserved samples from ST-bank as validation datasets including brain, heart, kidney and breast tissue samples and four independent ST studies as a test dataset, including desmoplastic small round cell tumor, colorectal cancer, vascular and colon samples (Supplementary Table 4). Results demonstrated that Loki notably outperformed both OpenAI CLIP and PLIP across all validation datasets. Specifically, Loki achieved Recall@5% of 0.125 and Recall@10% of 0.227 for brain (average 2.3-fold higher than OpenAI CLIP and 2.5-fold higher than PLIP), Recall@5% of 0.186 and Recall@10% of 0.291 for heart (average 3.2-fold higher than OpenAI CLIP and 3.1-fold higher than PLIP), Recall@5% of 0.173 and Recall@10% of 0.297 for kidney (average 3.2-fold higher than OpenAI CLIP and PLIP) and Recall@5% of 0.140 and Recall@10% of 0.240 for breast (average 2.6-fold higher than OpenAI CLIP and PLIP; Fig. 6e). On the test dataset, Loki further demonstrated substantial improvements, achieving Recall@5% of 0.117 and Recall@10% of 0.208 (average 3.1-fold higher than OpenAI CLIP and 3.0-fold higher than PLIP; Fig. 6e and Supplementary Table 4). Together, these results confirm Loki's superior performance in accurately retrieving paired transcriptomic information from images.

### Loki PredEx predicts ST gene expression from H&E images

Building on the success of Loki Align, Annotate and Decompose in analyzing tissue across the H&E image and transcriptomics data, we developed Loki PredEx to predict gene expression for image patches. Loki PredEx computes a weighted sum of gene expression from reference ST spots where weights are determined by similarity scores between the

query image and ST data, both encoded by OmiCLIP (Supplementary Fig. 8 and Methods). Several studies have explored predicting gene expression from H&E images using AI models[59–62]. We compared Loki PredEx with them on a normal human heart dataset comprising 39 samples. Loki accurately predicted highly variable gene expression, as demonstrated by the spatial distribution of the predicted gene expression (Extended Data Fig. 8). To evaluate the performance, we used mean squared error (MSE) and PCC as two metrics. Loki PredEx demonstrated superior performance, achieving the best results based on MSE scores in 28 of 39 cases, and ranking as the best in 16 of 39 samples based on PCC compared to Hist2ST, HisToGene, BLEEP and mclSTExp (Extended Data Fig. 9a). These results showed the robustness of OmiCLIP in predicting ST data across diverse datasets (Extended Data Fig. 9b). A major limitation of deep learning models like HisToGene is their heavy hardware requirements. Models like HisToGene and Hist2ST were optimized for smaller legacy ST datasets, with fewer spots. For instance, HisToGene is typically trained on less than 7,000 spot–image pairs. However, with modern ST technologies such as Visium, slides contain over 4,000 spots, pushing memory demands above 300 GB and complicating GPU-based training. In our experiments, training HisToGene on over 80,000 spots from 35 tissues required 4 h on 16 2.60 GHz Intel Xeon Gold 6348 CPUs for 100 epochs and Hist2ST took 31 h under similar conditions. Loki PredEx avoids these resource-intensive training needs, providing a more efficient alternative. Together, Loki PredEx delivers accurate ST gene expression predictions, and avoids these resource-intensive training needs, providing a more efficient alternative based upon the use of pretrained weights, highlighting its potential as a scalable infrastructure.

## Discussion

Existing dual-modality foundation models in computational biology[11,12] primarily combine images with textual descriptions, proving their utility in histopathology annotation and analysis. However, the natural language descriptions lack molecular insights for disease characterization. Our study first suggests that publicly available ST datasets provide sufficient volume and diversity to pretrain a foundation model bridging tissue morphology with genomics. The success of the development of our foundation model could represent a substantial step toward understanding molecular mechanisms regulating tissue phenotypes in health and disease.

We presented OmiCLIP, a high-performance histopathology image–omics foundation model by contrastive learning. Unlike visual–language foundation models, OmiCLIP integrates molecular insights with pathology images, complementing language descriptions. Benchmark results indicate that OmiCLIP performs comparably to, and in some cases surpasses visual–language foundation models in tissue annotation, suggesting that marker genes could serve as effective tissue labels independent of language. Notably, our annotation of tissue types incorporating both language description and marker genes shows promise for triple-modal foundation modeling of image, transcriptomics and language. Using marker genes as a label could potentially facilitate molecular investigation-related studies such as drug repurposing, immune response prediction and disease mechanism discovery.

A key question is whether OmiCLIP's transcriptomic encoder generalizes to other sequencing techniques like bulk RNA-seq and scRNA-seq. We evaluated the information of transcriptomic embeddings by cell annotation of scRNA-seq data (Supplementary Note 1) and tumor classification of bulk RNA-seq data (Supplementary Note 2). Our results show that OmiCLIP's transcriptomic embeddings efficiently cluster participants with cancer without specific training and accurately annotate cell types with even 1% of labeled cells.

Loki could potentially enhance 3D tissue analysis by integrating imaging and molecular modalities in a scalable and efficient manner. Emerging 3D histology and omics techniques already show promise in improving diagnostic accuracy by preserving native 3D tissue morphology, leading to better prognostic predictions and ultimately improved patient care[63–66]. However, challenges remain in spatial distortions and aligning molecular data across different modalities. Loki addresses these by aligning tissue slices and integrating ST, histology and scRNA-seq data, enabling a more comprehensive understanding of tissue architecture and cellular interactions, which is crucial for 3D tissue analysis. Incorporating Loki into workflows facilitates detailed molecular and spatial features analysis across tissue sections, supporting automated, scalable and high-resolution 3D tissue analysis.

Loki provides an AI-powered platform supporting the expansion of additional tools in a unified framework. Among the existing modules, Loki Annotate automates annotation and interpretation of molecular and spatial tissue features using associated or external RNA-seq data or marker genes. Loki PredEx predicts spatial gene expression from histology images, reducing reliance on costly and laborious ST experiments. These modules, leveraging contrastively aligned embeddings, enable efficient multimodal tissue reconstruction and analysis, providing a scalable solution to the growing demand for high-resolution tissue studies. Loki's ability to integrate diverse data types across tissue sections minimizes cost and complexity while accelerating workflows in enabling deeper insights into biological systems.

Compared to billion-scale datasets for developing visual–language models in the general machine-learning domain, the major limitation of this study is pretraining data size. We expect that continued use of training datasets may further improve the zero-shot performance. However, several biomedical multimodal foundation models were efficiently trained on million-scale datasets by removing duplicates and noise[11,12,67], a strategy we used to optimize training efficiency.

Notably, as a contrastive learning framework, OmiCLIP is not generative and cannot directly generate the accurate transcriptomic profile of the query image. Instead, it retrieves tissues with the most similar transcriptomic profiles to the query tissue. While it effectively embeds transcriptomic and histology data at the patch level, it does not inherently generate new data, such as reconstructing a WSI with gene expression patterns. However, OmiCLIP's patch-level embeddings could support generative approaches, such as diffusion models, to reconstruct WSIs with ST details. Future studies could refine the transcriptomic encoder using RNA-seq datasets like scRNA-seq and bulk RNA-seq data. Although ST-bank includes 32 organ types, rare conditions may be underrepresented. We suggest fine-tuning

alignment and decomposition tasks to ensure compatibility with datasets that are not covered in ST-bank (Extended Data Fig. 10).

Unlike single-cell foundation models like scGPT[57], Geneformer[26] and scFoundation[27], our approach models omics data as text, effectively bridging molecular and visual modalities. Representing gene expression data as text leverages natural language processing models to embed biological information into a high-dimensional space, offering several advantages over using gene expression values directly. First, text embeddings integrate omics data with various biological entities such as pathways, functional annotations[68] and cell types[69], extending the model's capabilities beyond tissue alignment and decomposition, making it adaptable to a broader range of biological tasks. Second, this approach aligns with other multimodal foundation models, and allows incorporation of proteomics, metabolomics and DAPI images into the same unified space. In contrast, raw gene expression values lack flexibility for such integrations and require additional preprocessing. Third, text-based foundation models trained on billions of tokens provided robust text embeddings, like GenePT[23], demonstrating that gene embeddings from textual descriptions can match or surpass models trained on extensive gene expression datasets. This supports our approach of utilizing text-based embeddings to capture rich biological information efficiently.

While integrating two modalities enhances information capture, it may also introduce noise or misalignment, potentially overshadowing benefits. If one modality dominates, performance gains from dual-modality fusion may be minimal.

Loki Decompose is valuable in scenarios where sequencing costs limit transcriptomic profiling. By estimating cell-type proportions from images, researchers can preselect, screen or perform batch processing of samples cost-effectively for exploratory studies and large-scale screenings. Loki Retrieve utilizes curated reference images for ground-truth comparisons, aiding validation and interpretation, especially when training data for prediction models like Loki PredEx are scarce. Together, our approach contributes to a unified, scalable framework for multimodal analysis.

To conclude, we created ST-bank, a dataset of over 2 million pathology-specific image–transcriptomics pairs. We developed OmiCLIP to integrate these data, forming a visual–omics foundation model. Leveraging OmiCLIP, we built Loki, an infrastructure enabling multimodal analysis for tissue alignment, tissue annotation, cell-type decomposition, histology image–transcriptomics retrieval and ST gene expression prediction. These capabilities represent a fundamental step toward bridging and applying foundation models in genomics for histopathology.

## Online content

## Methods

### Training dataset curation

We curated a large dataset of histopathology image–transcriptomics pairs using publicly available 10x Visium datasets (Supplementary Table 1). H&E images were cropped to match ST spot sizes, and text sentences were generated by combining the top 50 expressed genes per spot into sentences. For example, the top-expressed genes in one spot, for example, *SNAP25*, *ENO2*, *CKB*, *GRIN2C* and *CAMK4*, will be combined into a sentence: 'SNAP25 ENO2 CKB GRIN2C CAMK4 … MTOR VPS13D'. Data preprocessing involved removing duplicates and excluding low-resolution H&E images ($<2{,}000 \times 2{,}000$ pixels), and normalizing raw count matrices following standard protocols using Seurat[70] and Scanpy[71]. For datasets in transcripts per million or fragments per kilobase of transcript per million fragments mapped formats, which cannot be normalized to standard gene expression profiles, were retained unchanged. Quality control was applied to filter out contaminated, extremely low-quality or damaged cells, retaining only those with over 200 expressed genes. Ensembl gene IDs were converted to gene symbols for consistency. Housekeeping genes were removed to ensure a more biologically relevant analysis. These steps resulted in ST-bank, a pathology-specific image–transcriptomics caption dataset of 2,185,571 pairs.

### Downstream evaluation datasets (details in Supplementary Note 3)

**Tissue alignment.** Simulated datasets were generated from ten human tissue slices including two breast cancer[72,73], one colorectal liver cancer[74], one liver cancer[75], one prostate cancer[76], one 10x Genomics prostate cancer, one 10x Genomics colon cancer, one embryonic lung[77], one normal small intestine[34] and one sleep apnea tonsil sample[78]. We simulated new ST experiments by perturbing both gene expression and spatial locations at different levels of noise, generating 10 simulated datasets per real dataset, totaling 200 datasets (100 low-noise, 100 high-noise). Real-world data tests used a normal human small intestine Visium dataset[34] of eight adjacent tissue slices, a human ovarian carcinosarcoma Visium dataset[35] of two adjacent tissue slices and a human breast cancer Visium and Xenium dataset[37].

**Tissue annotation.** Bulk RNA-seq data-based annotation used three normal human breast and three human heart failure histology images[79,80] and three breast cancer histology images from TCGA. Pathology experts annotated different tissue regions. Bulk RNA-seq datasets including 663 human adipose and 504 fibroblast samples from the Genotype-Tissue Expression Portal and three paired tumor biopsy samples from TCGA. Marker gene-based annotation included four datasets: CRC7K (6,333 colorectal adenocarcinoma images), WSSS4LUAD (10,091 LUAD images), LC25000 (25,000 lung and colon images) and PatchCamelyon (32,768 lymph node images).

**Cell-type decomposition.** We downloaded a human colorectal cancer dataset[48] to create pseudo-Visium spots in the Visium-HD capture area. Pathology experts annotated different tissue regions. We collected an in-house TNBC patient-derived xenograft for processing on Xenium slides, to create pseudo-Visium spots with an external scRNA-seq reference of

TNBC[51] for decomposition. We also downloaded a mouse brain Visium dataset[50] and a scRNA-seq dataset[49] from the Allen Institute.

**H&E image-to-ST retrieval.** We collected our in-house heart failure patient tissue, paraffin-embedded Alzheimer's disease patient tissue, and metaplastic breast cancer and TNBC patient-derived xenografts. The validation datasets included brain, heart, kidney and breast samples, and the test dataset included desmoplastic small round cell tumor, colorectal cancer, vascular and colon samples (Supplementary Table 4).

**ST gene expression prediction.** We used a normal human heart sample dataset[81] of paired ST data and H&E images including 39 samples.

## OmiCLIP model training

OmiCLIP consisted of an image encoder and a text encoder following CoCa[9] settings. The image encoder was based on a standard vision transformer (ViT)[82] with an input image size of $224 \times 224$ pixels. The text encoder was based on a causal masking transformer with input text length of 76 tokens. Regarding the initial embeddings of ST data, the initial text encoder was not trained from scratch but on LAION-5B[83], including biological literature, which may explain its tendency to cluster similar tissue patches. The model was trained for 20 epochs, using one NVIDIA A100 80-GB GPU with a local batch size of 64. The output vectors of the image and text encoders with dimensions of 768 were optimized by minimizing the contrastive loss on a given batch. All experiments were run in Python v.3.9. Detailed software versions are: CUDA v.12.2; torch v.2.3.1; torchvision v.0.18.1; scipy v.1.13.1; pillow v.10.4.0; scikit-learn v.1.5.2; pandas v.2.2.3; numpy v.1.25.0; and scanpy v.1.10.3.

## OmiCLIP model fine-tuning

To improve performance on downstream tasks, OmiCLIP allows fine-tuning with user datasets. The fine-tuning dataset is created by preprocessing Visium data using a standard 10x Space Ranger pipeline and generating gene name sentences as describe in 'Training dataset curation', ensuring compatibility with the pretraining dataset format. Fine-tuning is done using contrastive loss[9] between image embeddings and paired text embeddings of the top-expressed gene sentences. The contrastive loss is calculated according to equation (1):

$$L_{Con} = -\frac{1}{N}\left( \sum_{i}^{N} \log \frac{\exp\left(\frac{x_i^T y_i}{\sigma}\right)}{\sum_{j=1}^{N} \exp\left(\frac{x_i^T y_j}{\sigma}\right)} + \sum_{i}^{N} \log \frac{\exp\left(\frac{y_i^T x_i}{\sigma}\right)}{\sum_{j=1}^{N} \exp\left(\frac{y_i^T x_j}{\sigma}\right)} \right),$$

(1)

where $x_i$ and $y_j$ denote the normalized image and text embeddings, respectively. $N$ denotes the batch size, while $\sigma$ represents the temperature parameter. The pretrained model was fine-tuned for ten epochs for the tissue alignment task and five epochs for the cell-type decomposition task, using a local batch size of 64, minimizing the contrastive loss.

## Loki Align

We first fine-tuned OmiCLIP using paired ST data and H&E image of the target sample. The fine-tuned OmiCLIP text encoder and image encoder then encoded ST data and image, respectively. We used a nonrigid point set registration algorithm based on the CPD method[31], which iteratively aligns two point sets by minimizing the statistical discrepancies.

The algorithm initializes the transformation matrix $W$ to zero and sets the variance $\sigma^2$ of point displacements as shown in equation (2):

$$\sigma^2 = \frac{1}{DNM} \sum_{m, n = 1}^{M, N} \|x_n - y_m\|^2.$$

(2)

Where $D$ is the point's dimensionality, $M$, $N$ are the number of points in each set, and $x$, $y$ are the source and target points in sets $X$ and $Y$, respectively. Point sets are modeled as Gaussian mixture samples, with correspondence probability matrix $G$ computed as shown in equation (3):

$$g_{ij} = \exp^{-\frac{1}{2\beta^2} \|y_i - y_j\|^2}.$$

(3)

This forms the basis for expectation–maximization steps, which iterate until convergence. During the E-step, posterior probabilities $P$ of correspondences update as given by equation (4):

$$P_{mn} = \frac{\exp^{-\frac{1}{2\sigma^2} \|x_n - (y_m + G(m, \cdot)W)\|^2}}{\sum_{k = 1}^{M} \exp^{-\frac{1}{2\sigma^2} \|x_n - (y_k + G(k, )W)\|^2} + \frac{w}{1 - w} \frac{\left(2\pi\sigma^2\right)^{D/2} M}{N}}.$$

(4)

In the M-step, $W$ updates according to equations (5)–(7):

$$\left(G + \lambda\sigma^2 d(P1)^{-1}\right)W = d(P1)^{-1}PX - Y,$$

(5)

$$N_p = 1^T P1, T = Y + GW,$$

(6)

$$\sigma^2 = \frac{1}{N_p D}\left(tr\left(X^T d\left(P^T 1\right)X\right) - 2tr\left((PX)^T T\right) + tr\left(T^T d(P1)T\right)\right),$$

(7)

where the transformation weights $W$ are constrained to $0 \leq W \leq 1$. Parameters $\beta > 0$ controls transformation stiffness and the trade-off between data fidelity and smoothness, respectively.

We optimized CPD by adding the first two principal components of embeddings generated by OmiCLIP image encoder or text encoder, along with the original two-dimensional coordinates. The M-step was optimized by updating only the coordinates to minimize loss. We further calculated the homography matrix with translation and rotation between spots before and after alignment to avoid tremendous distortion. For PASTE, GPSA and CAST, we used their default configuration for tissue preparation and alignment in Visium data.

### Loki Annotate

**Bulk RNA-seq data.** OmiCLIP enables zero-shot annotation by learning an aligned latent space for image and transcriptomic embeddings, eliminating the need for retraining. We used OmiCLIP text encoder to encode bulk RNA-seq data and image encoder for H&E images, then calculated cosine similarity between transcriptomic and image embeddings at spot level.

**Marker genes.** Annotation was determined by selecting candidate texts with the highest similarity score to image query. We evaluate this using four datasets: CRC7K, LC25000, PatchCamelyon and WSSS-4LUAD. For Loki, text candidates were generated according to marker genes of each tissue type (Supplementary Table 3). For the PLIP model, text candidates were generated from tissue-type descriptions (Supplementary Table 3). The OmiCLIP image encoder encoded images resized to $20 \times 20$ pixels, consistent with its pretraining. OpenAI CLIP and PLIP models used their default configuration and functions for image and text processing.

**Multimodal annotation.** For jointly using Loki and PLIP, we summed their normalized similarity scores. Let, $s_{Loki}(I,T)$ and $s_{PLIP}(I,T)$ represent the similarity scores between an image $I$ and text $T$ computed by Loki and PLIP, respectively. Normalized scores were obtained according to equations (8)–(10):

$$\hat{s}_{Loki}(I,T) = \frac{s_{Loki}(I,T) - \min_{T'} s_{Loki}(I,T')}{\max_{T'} s_{Loki}(I,T') - \min_{T'} s_{Loki}(I,T')},$$

(8)

$$\hat{s}_{PLIP}(I,T) = \frac{s_{PLIP}(I,T) - \min_{T'} s_{PLIP}(I,T')}{\max_{T'} s_{PLIP}(I,T') - \min_{T'} s_{PLIP}(I,T')},$$

(9)

$$s_{\text{combine}}(I, T) = \hat{s}_{\text{Loki}}(I, T) + \hat{s}_{\text{PLP}}(I, T).$$

(10)

The candidate text $T^*$ with the highest combined similarity score was identified as given by equation (11):

$$T^* = \underset{T}{\text{argmax}} \ \ s_{\text{combine}}(I, T).$$

(11)

## Loki Decompose

To decompose human colorectal cancer slices, we fine-tuned OmiCLIP using paired Visium ST data and H&E images. We then used fine-tuned OmiCLIP text encoder to encode scRNA-seq data and pseudo-Visium ST data, and image encoder to encode H&E images. For in-house TNBC human samples, we fine-tuned OmiCLIP using a quarter of a region (top-right, top-left, bottom-right or bottom-left) of pseudo-Visium ST data and H&E images, then encoded scRNA-seq data and ST data via the text encoder and H&E images via the image encoder. Similarly, for mouse brain cortex slices, we fine-tuned OmiCLIP using adjacent Visium ST data and H&E images, then encoded scRNA-seq data and H&E images accordingly.

We used a nonconvex optimization algorithm implemented by Tangram to co-register OmiCLIP embeddings of scRNA-seq data with those of ST data or H&E images. We aimed to obtain a probabilistic mapping matrix $M$ aligning single cells to specific spots based on embedding similarities between scRNA-seq and ST data or scRNA-seq and H&E images. The mapping matrix $M$ of dimensions spots-by-cells quantifies the likelihood that a given single cell is located within a particular spot. The scRNA-seq data matrix $S$ is structured as cells-by-embeddings, while the ST data or H&E image matrix $G$ is formatted as spots-by-embeddings. The optimal mapping matrix $M$ is derived by minimizing the loss function $L(S, M)$ as shown in equation (12):

$$L(S, M) = \sum_{k}^{n_{\text{embeddings}}} \cos_{\text{distance}}\left(\left(M^T S\right)_{*, k}, G_{*, k}\right).$$

(12)

Here, $\cos_{\text{distance}}$ denotes the cosine distance between OmiCLIP embeddings of the mapped single cells and those of ST data or H&E images. The loss function aims to minimize the cosine distance between the projected single-cell embeddings $M^T S$ and the embeddings of ST data or H&E images $G$, thereby ensuring that the embeddings of the single cells, when mapped, resemble those observed in the spatial data as closely as possible. Each element $M_{ij}$ in the matrix represents the probability that $\text{cell}_i$ correspond to $\text{spot}_j$, integrating

the cellular composition of the spatial spot. For Tangram, we used a uniform density prior for each spot without target count, aligning with Loki Decompose. To enhance efficiency, we adapted the mapping at the cell cluster level. The same settings were used for Loki, while Spatial Seurat, CARD, CytoSPACE, RCTD, Cell2location and spatialD-WLS utilized their default configurations and tissue preparation and decomposition functions. For scGPT, scFoundation and GeneFormer, we used default configuration and tissue preparation functions before using the Tangram method with same default configurations to decompose cell types. To evaluate their performance, we used cell-type information from Xenium, Visium-HD and pathology annotation as ground truth.

To improve decomposition performance in regions with complex cellular heterogeneity, we developed a refinement strategy inspired by NMS[58]. This method prioritizes the most probable cell type within each spot, reducing overlapping or ambiguous assignments when multiple cell types have comparable probabilities. This refined method is recommended in complex spatial scenarios, such as colorectal cancer. For $N$ total spots (indexed by $i = 1, …, N$), and $C$ cell types, we defined $P_{i,c}$ as the original probability of cell type $c$ at spot $i$. The NMS-based refinement follows two steps: selecting the highest probability cell type and suppressing others. The most likely cell type at each spot $i$ was determined as given by equation (13):

$$c_i^* = \arg \max_{c \in C} P_{i,c}.$$

(13)

Then refined probabilities $P_{i,c}^{(\mathrm{NMS})}$ was defined according to equation (14):

$$P_{i,c}^{(\mathrm{NMS})} = \begin{cases} P_{i,c}, & if \ \ c = c_i^*, \\ 0, & \text{otherwise}. \end{cases}$$

(14)

This NMS-based refinement ensured that only the cell type with the highest likelihood remained at each spot, eliminating competing probabilities and improving spatial decomposition accuracy.

## Loki Retrieve

Similarly to Loki Annotate, the retrieval results were decided by choosing candidate transcriptomics with the highest similarity score to the image query.

$$k* = \arg \max_{k \in K} sim(I_q, I_k).$$

(15)

Here, as shown in equation (15), $K$ indicates the set of all pairs, $I_q$ indicates the image embeddings of a given query, $I_k$ indicates the transcriptomics embeddings and $k*$ indicates the candidate transcriptomics with the highest similarity score. We then calculated the

similarity between the embeddings of the query image and the image that is paired with the retrieved transcriptomics as the ground truth.

### Loki PredEx

We applied 10-fold cross-validation to evaluate Loki PredEx's performance. In each fold, OmiCLIP was fine-tuned on the training set for ten epochs, and then we used the fine-tuned OmiCLIP text encoder to encode the ST data of training sets and the image encoder to encode the image of validation sets. For each spot in the validation set, cosine similarity between its image embeddings and all the transcriptomic embeddings in the training set was computed, and these weights were used to generate ST gene expression prediction for validation set spots via a weighted average as given by equation (16):

$$X_i = \frac{\sum_{j \epsilon T} w_{i,j} \cdot X_j}{\sum_{j \epsilon T} w_{i,j}},$$

(16)

where $T$ is the set of all spots in the training set, $X_i$ is the predicted gene expression for validation spot $i$, $w_{i,j}$ is the similarity score between validation spot $i$ and training spot $j$, and $X_j$ is the gene expression for training spot $j$.

To benchmark performance, we compared Loki PredEx against HisToGene, Hist2ST, BLEEP and mclSTExp, on the same dataset. In each fold, the top 300 expressed genes in the validation set were selected for prediction. We followed default training settings: 100 epochs for HisToGene, 4 epochs for BLEEP, 90 epochs for mclSTExp and 110 epochs reduced from 350 due to computational resource constraints for Hist2ST. By applying the same cross-validation procedure and evaluating the top 300 expressed genes in each fold, we ensured a fair comparison between Loki PredEx and baseline models.

### Evaluation metrics and statistical analysis

In 'OmiCLIP improves image and transcriptomics representations', we used the Leiden algorithm in Scanpy[71] to cluster ST with default parameters including a resolution of 1 and a sparse adjacency matrix derived from neighbor connectivity. We then calculated the UMAP embeddings with an effective minimum distance of 0.5 and three dimensions.

The CH score, also referred to as the variance ratio criterion, was used to evaluate clustering quality for a given dataset by comparing between-cluster dispersion and within-cluster dispersion. It was computed using two sets of ground truth, a benchmarked dataset containing 95 tissue samples from the ST-bank, which included expert-annotated cell types (Supplementary Table 2) and the Leiden clustering (described above) labels for samples without cell-type annotations. For a dataset with $n$ points $\{x_1, ..., x_n\}$ divided into $k$ clusters $\{C_1, ..., C_k\}$, CH score is the ratio normalized by the number of degrees of freedom for between-cluster and within-cluster dispersions, respectively, as given by equation (17):

$$CH = \frac{BCSS/(k-1)}{WCSS/(n-k)}.$$

(17)

Between-cluster sum of squares (BCSS) is calculated as the weighted sum of squared Euclidean distances from each cluster's centroid to overall centroid, as given by equation (18):

$$BCSS = \sum_{i=1}^{k} n_i \|c_i - c\|^2.$$

(18)

Here, $n_i$ is the number of points in cluster $C_i$, $c_i$ is the centroid of cluster $C_i$, and $c$ is the overall centroid. BCSS quantifies separation between clusters, with higher value indicating better separation. Within-cluster sum of squares (WCSS) measures the cohesion of the clusters with smaller values indicating tighter clustering and is the total squared Euclidean distances from each data point to its cluster centroid, as given by equation (19):

$$WCSS = \sum_{i=1}^{k} \sum_{x \in C_i} \|x - c_i\|^2.$$

(19)

The PCC, which ranges from $-1$ to $1$, assessed tissue alignment and gene expression prediction. Given paired data $\{(x_1, y_1), \ldots, (x_n, y_n)\}$ consisting of $n$ pairs, PCC represented by $r_{xy}$ is defined in equation (20):

$$r_{xy} = \frac{n \sum x_i y_i - \sum x_i \sum y_i}{\sqrt{n \sum x_i^2 - (\sum x_i)^2} \sqrt{n \sum y_i^2 - (\sum y_i)^2}},$$

(20)

where $n$ is the sample size, and $x_i$, $y_i$ are the individual sample points indexed with $i$.

Kendall's tau coefficient, which ranges from $-1$ to $1$, assessed tissue alignment, as given by equation (21):

$$\tau = \frac{P - Q}{\sqrt{(P + Q + T)(P + Q + U)}},$$

(21)

where $P$ denotes the number of concordant pairs, $Q$ is the number of discordant pairs, while $T$ and $U$ represent ties occurring solely in $x$ or solely in $y$, respectively.

JS divergence, which ranges from 0 to 1, assessed cell-type decomposition. To calculate JS divergence between two probability distributions $P$ and $Q$, we first computed the pointwise average distribution, as given by equation (22):

$$M = \frac{1}{2}(P + Q).$$

(22)

Then, we calculated Kullback–Leibler (KL) divergence of each distribution with respect to $M$: $D_{\mathrm{KL}}(P \| M)$ and $D_{\mathrm{KL}}(Q \| M)$. KL divergence is a measure of how one probability distribution diverges from a second distribution, as given by equation (23):

$$\mathrm{KL}(P \| Q) = \sum P(x) \log \left( \frac{P(x)}{Q(x)} \right).$$

(23)

JS divergence is the average of these two KL divergences as given by equation (24):

$$D_{\mathrm{JS}}(P \| Q) = \frac{1}{2} D_{\mathrm{KL}}(P \| M) + \frac{1}{2} D_{\mathrm{KL}}(Q \| M).$$

(24)

The SSIM, which ranges from −1 to 1, assessed cell-type decomposition, where we considered the cell-type distribution in spatial as image. For two images $x$ and $y$, as shown in equation (25):

$$\mathrm{SSIM}(x, y) = \frac{(2\mu_x\mu_y + C_1)(2\sigma_{xy} + C_2)}{(\mu_x^2 + \mu_y^2 + C_1)(\sigma_x^2 + \sigma_y^2 + C_2)}.$$

(25)

where $u_x$ and $u_y$ are the mean intensities of images $x$ and $y$, $\sigma_x^2$ and $\sigma_y^2$ are the variances of $x$ and $y$, $\sigma_{xy}$ is the covariance between $x$ and $y$, $C_1$ and $C_2$ are small constants to stabilize the division when the denominators are close to zero.

MSE assessed ST gene expression prediction by comparing the Euclidean distance of the highly expressed gene expression between ground truth and prediction for each method within the same location.

The impact score assessed the performance of cell-type decomposition. For each decomposition method $m$, we computed the mean JS divergence, $\mathrm{JS}_m$, and the mean SSIM, $\mathrm{SSIM}_m$, across all cell types as given by equations (26) and (27):

$$\mathrm{SSIM}_m = \frac{1}{N} \sum_{c=1}^{N} \mathrm{SSIM}(p_c, q_c),$$

(26)

$$JS_m = \frac{1}{N} \sum_{c=1}^{N} JS(p_c, q_c),$$

(27)

where $p_c$ and $q_c$ represent the ground truth and predicted proportions, respectively. $N$ represents the total number of cell types. We standardized SSIM and JS divergence across methods to enable direct comparison, as they operate on different scales. The standardized metrics $Z_{SSIM_m}$ and $Z_{JS_m}$ are calculated according to equation (28):

$$Z_{SSIM_m} = \frac{SSIM_m - \mu_{SSIM}}{\sigma_{SSIM}},$$

(28)

where $\mu_{SSIM}$ and $\sigma_{SSIM}$ are the mean and standard deviation of SSIM across methods. Because lower JS divergence indicates better performance, we inverted the standardized JS divergence values by multiplying them by −1, as given by equation (29):

$$Z_{JS_m} = - \frac{JS_m - \mu_{JS}}{\sigma_{JS}},$$

(29)

where $\mu_{JS}$ and $\sigma_{JS}$ are the mean and standard deviation of JS divergence across methods. To generate a unified metric for decomposition accuracy, we averaged the inverted JS divergence $z$-scores and the SSIM $z$-scores for each method as given by equation (30):

$$Impact\ score_m = \frac{Z_{JS_m} + Z_{SSIM_m}}{2}.$$

(30)

F1 score, which ranges from 0 to 1, assessed zero-shot and linear probing methods as given by equation (31):

$$F1 = \frac{2 \times precision \times recall}{precision + recall} = \frac{2 \times TP}{2 \times TP + FP + FN}.$$

(31)

Here, TP represents true positives, FP represents false positives and FN represents false negatives. A higher F1 score indicates better overall performance in classification tasks. The weighted F1 score was calculated by averaging the F1 scores for each class, with each class's contribution weighted based on its frequency in the data.

Recall@K assessed image-to-transcriptomics retrieval. Let $Q$ be the set of all queries, and $N$ be the total number of queries. For each query $q \in Q$, the retrieval model outputs a ranked list of candidate targets as given by equation (32):

$$R_q = [c_{q,1}, c_{q,2}, \ldots, c_{q,i}],$$

(32)

where $c_{q,i}$ is the $i^{th}$ highest-ranked candidate for query $q$ based on cosine similarity, and quantile $(q)$ is the quantile of the smallest index $i$ of the ground-truth target. Recall@K is defined as the fraction of queries for which the ground-truth target occurs at rank $K$ or better as given by equation (33):

$$\text{Recall@K} = \frac{1}{N} \sum_{q \in Q} \text{I}[\text{quantile}(q) \leq K],$$

(33)

where $\text{I}[\cdot]$ is an indicator function that takes the value of 1 if quantile $(q) \leq K$ and 0 otherwise.

Two-sided Student's $t$-test and Wilcoxon rank-sum test were used to assess statistical significance between models.

## Extended Data

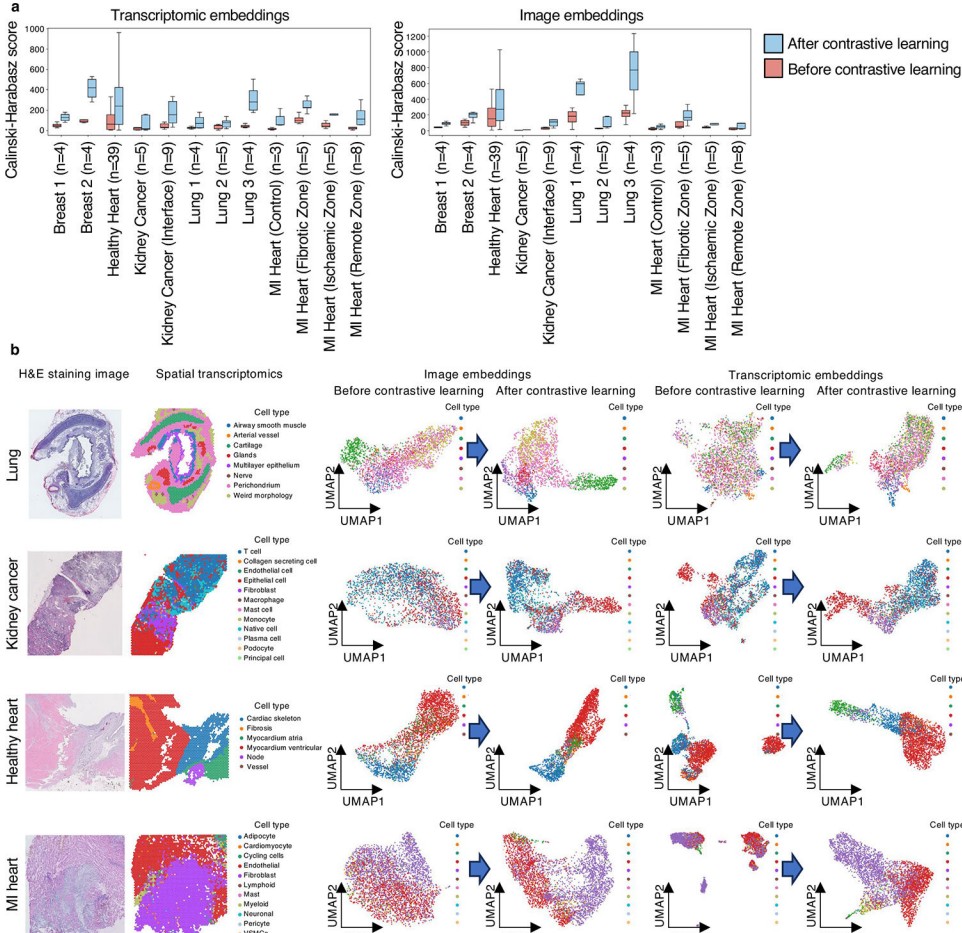

**Extended Data Fig. 1 |. Image and transcriptomic representations.**
**a**, Clustering performance on ST-bank data with cell type annotation. Left: clustering performance using transcriptomic embeddings generated from OmiCLIP model before and after training. Right: clustering performance usings image embeddings from OmiCLIP model before and after training. The Calinski-Harabasz scores were calculated on the embeddings (Methods) using the pretrained OmiCLIP transcriptomic (left) and image (right) encoders, evaluated for each organ type. Higher Calinski-Harabasz scores indicate better separation capability between clusters of the embeddings. In the box plots, the middle line represents the median, the box boundaries indicate the interquartile range, and the whiskers extend to data points within 1.5× the interquartile range. **b**, Image and transcriptomic embeddings of the lung, kidney cancer, healthy heart, and Myocardial Infarction (MI) heart samples. Each row corresponds to a WSI and showcases information from two modalities. The first column are H&E images showing tissue morphology; the second column are the heatmaps of ST data with the colors indicating the cell types; the third column are the UMAP of image embeddings colored by cell types before and after contrastive learning; the fourth column are the UMAP of transcriptomics embeddings colored by cell types before and after contrastive learning.

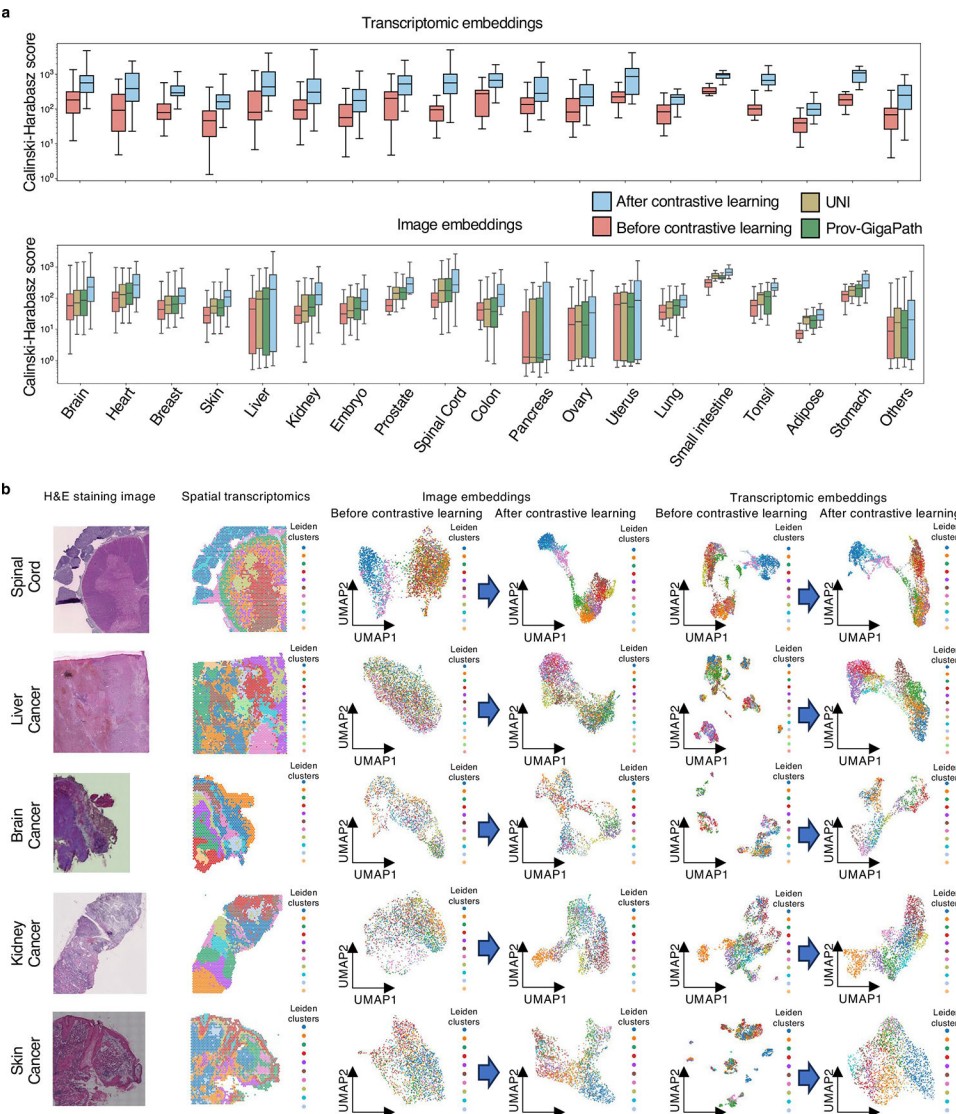

**Extended Data Fig. 2 |. Image and transcriptomic representations analysis.**
**a**, Clustering performance on all ST-bank data. Top: clustering performance using transcriptomic embeddings generated from OmiCLIP model before and after training. Bottom: clustering performance usings image embeddings from OmiCLIP model before and after training, and image embeddings generated from UNI and Pro-GigaPath, respectively. The Calinski-Harabasz scores were calculated on the embeddings using the pre-trained OmiCLIP transcriptomic (top) and image (bottom) encoders, evaluated for each organ type. Higher Calinski-Harabasz scores indicate better separation capability between clusters of the embeddings. In the box plots, the middle line represents the median, the box boundaries indicate the interquartile range, and the whiskers extend to data points within 1.5× the interquartile range. Sample sizes are skin: 163, brain: 119, breast: 97, heart: 73, kidney: 73, embryo: 73, others: 64, liver: 57, prostate: 49, spinal cord: 44, ovary: 32, colon: 29, pancreas: 25, lung: 22, tonsil: 18, uterus: 17, adipose: 15, small intestine: 14, and stomach: 12. **b**, Image and transcriptomic embeddings of the spinal cord, liver

cancer, brain cancer, kidney cancer and skin cancer samples. Each row corresponds to a WSI and showcases information from two modalities. The first column are H&E images showing tissue morphology; the second column are the heatmaps of ST data with the colors indicating the ST data clustering using Leiden algorithm (Methods); the third column are the UMAP of image embeddings colored by ST Leiden clusters before and after contrastive learning; the fourth column are the UMAP of transcriptomics embeddings colored by ST Leiden clusters before and after contrastive learning.

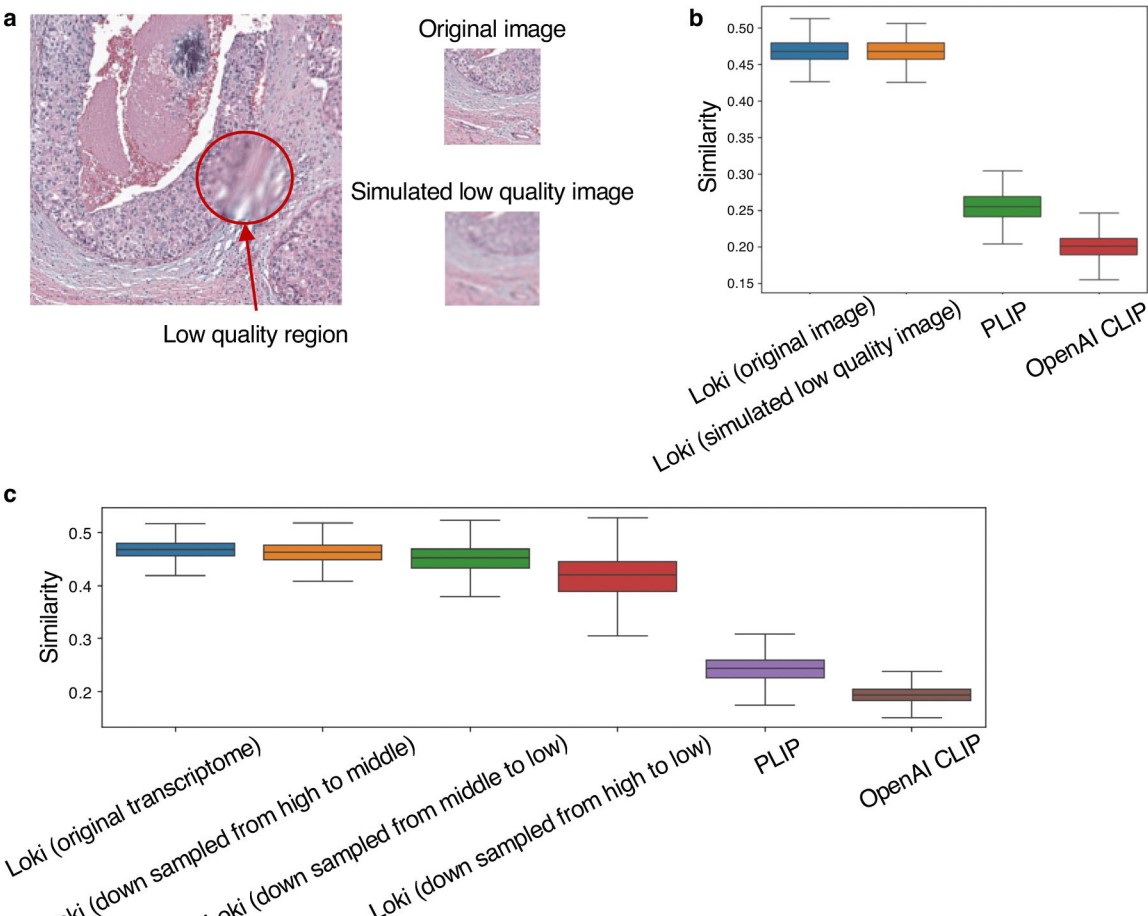

**Extended Data Fig. 3 |. OmiCLIP's robustness for image quality and sequencing depth.**
**a**, Example image with low-quality region marked in red line and simulated low-quality image by adding Gaussian noise. **b**, Cosine similarity of paired transcriptomic and image embeddings using OmiCLIP (original image and simulated low-quality image), PLIP (original image), and OpenAI CLIP (original image). In the box plots, the middle line represents the median, the box boundaries indicate the interquartile range, and the whiskers extend to data points within 1.5× the interquartile range. Sample sizes are 10 for each simulated condition. **c**, Cosine similarity of the paired image with transcriptomic embeddings using OmiCLIP (original transcriptomes and down sampled transcriptome from high sequencing depth to middle sequencing depth, middle sequencing depth to low sequencing depth, and high sequencing depth to low sequencing depth, respectively), PLIP

(original transcriptome), and OpenAI CLIP (original transcriptome). In the box plots, the middle line represents the median, the box boundaries indicate the interquartile range, and the whiskers extend to data points within 1.5× the interquartile range, n = 500.

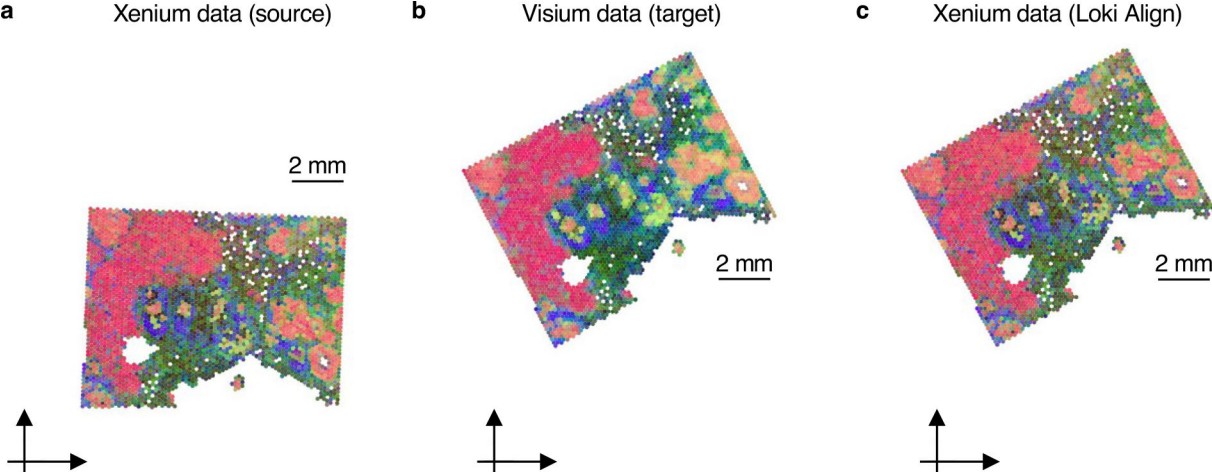

**Extended Data Fig. 4 |. Visium and Xenium tissue alignment.**
Tissue alignment results on breast cancer sample using Loki Align. **a**, Source Xenium ST data. **b**, Target Visium ST data. **c**, Xenium ST data after Loki alignment.

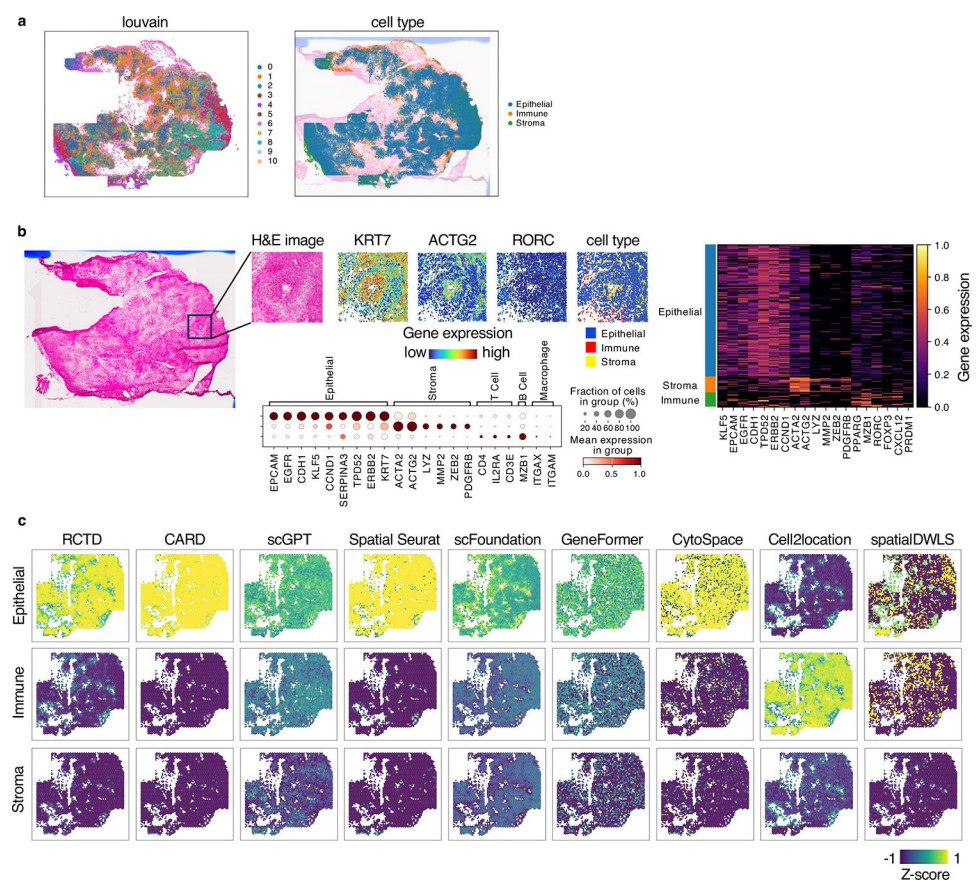

**Extended Data Fig. 5 |. Cell type decomposition of TNBC case study.**
**a,** Xenium data from our in-house TNBC patient sample, colored by Louvain clusters and cell types, respectively. **b**, H&E image, marker gene expression (KRT7, ATCG2, RORC), and cell type distribution in an example zoom-in region of the TNBC sample. **c,** Cell type decomposition results on 3 major cell types of the TNBC sample using ST by RCTD, CARD, scGPT, Spatial Seurat, scFoundation, GeneFormer, CytoSPACE, Cell2location, and SpatialDWLS, respectively. The color of the heatmap reflects the z-score, calculated by the enrichment of each cell type.

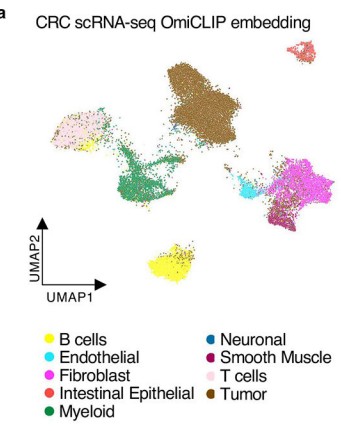

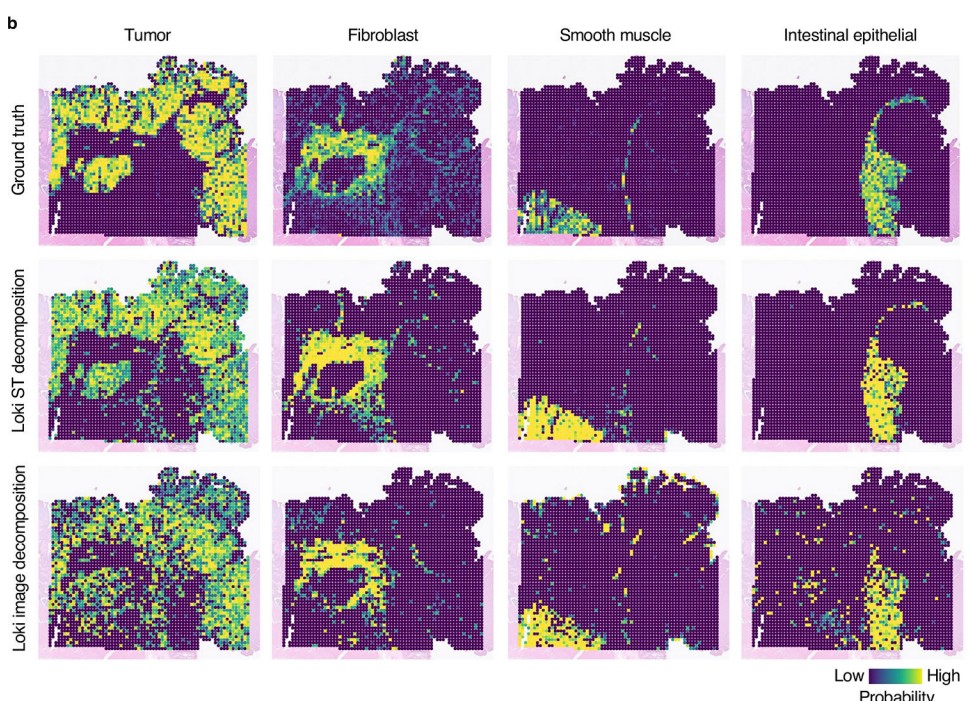

**Extended Data Fig. 6 |. Cell type decomposition of colorectal case study.**
**a**, UMAP representation of the OmiCLIP transcriptomic embeddings colored by cell types, where each dot represents a spot. **b**, Cell type decomposition result using Loki ST decomposition and Loki image decomposition respectively on human colorectal sample within the Visium HD capture area, and ground truth. Heatmap shows the cell type

distribution of tumor, fibroblast, smooth muscle, and intestinal epithelial, respectively, with color reflecting the probability of each cell type.

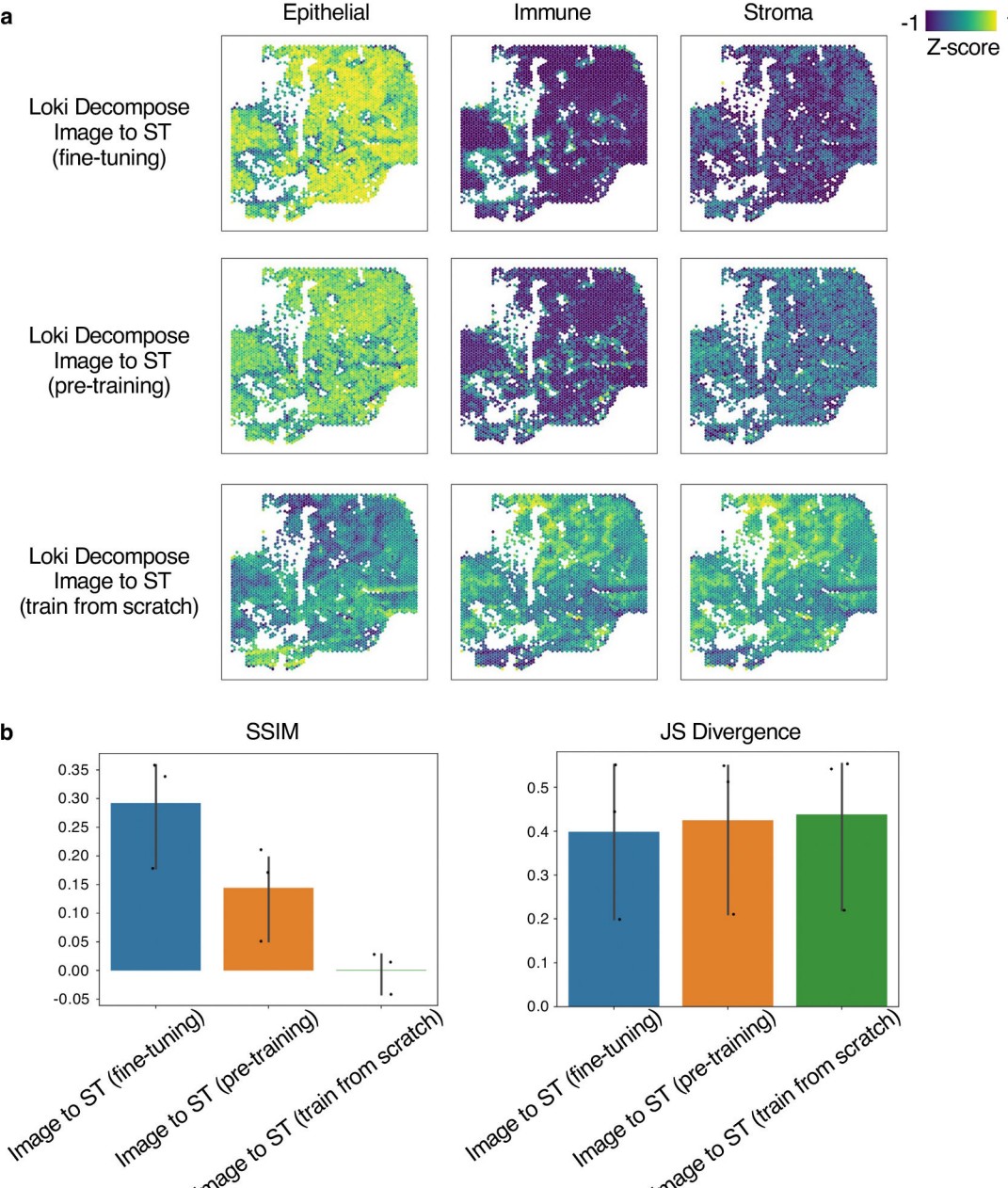

**Extended Data Fig. 7 |. Cell type decomposition of fine-tuning, pre-training, and train from scratch.**

**a**, Cell type decomposition results on 3 major cell types of the TNBC sample using Loki Decompose Image-to-ST (fine-tuning, pre-training, and train from scratch). The color of the heatmap reflects the z-score, calculated by the enrichment of each cell type. **b**, Bar plot shows the accuracy of decomposition of 3 major cell types by Loki Decompose Image-to-ST (fine-tuning, pre-training, and train from scratch). Error bar is standard deviation with center measured by mean.

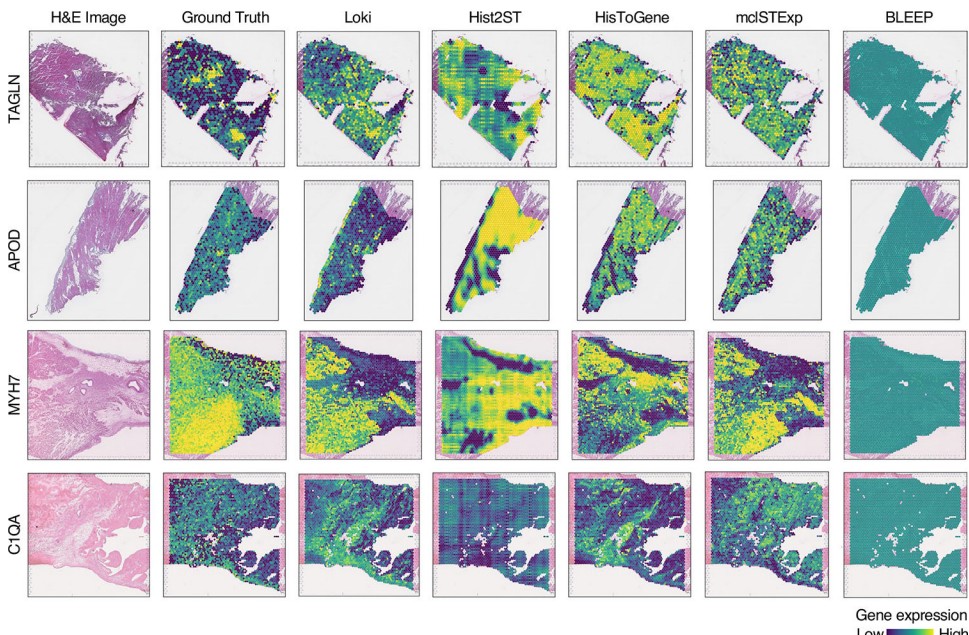

**Extended Data Fig. 8 |. Examples of ST gene expression prediction.**
H&E images, ground truth ST gene expression, and ST gene expression predicted by Loki, Hist2ST, HisToGene, BLEEP, and mclSTExp, respectively.

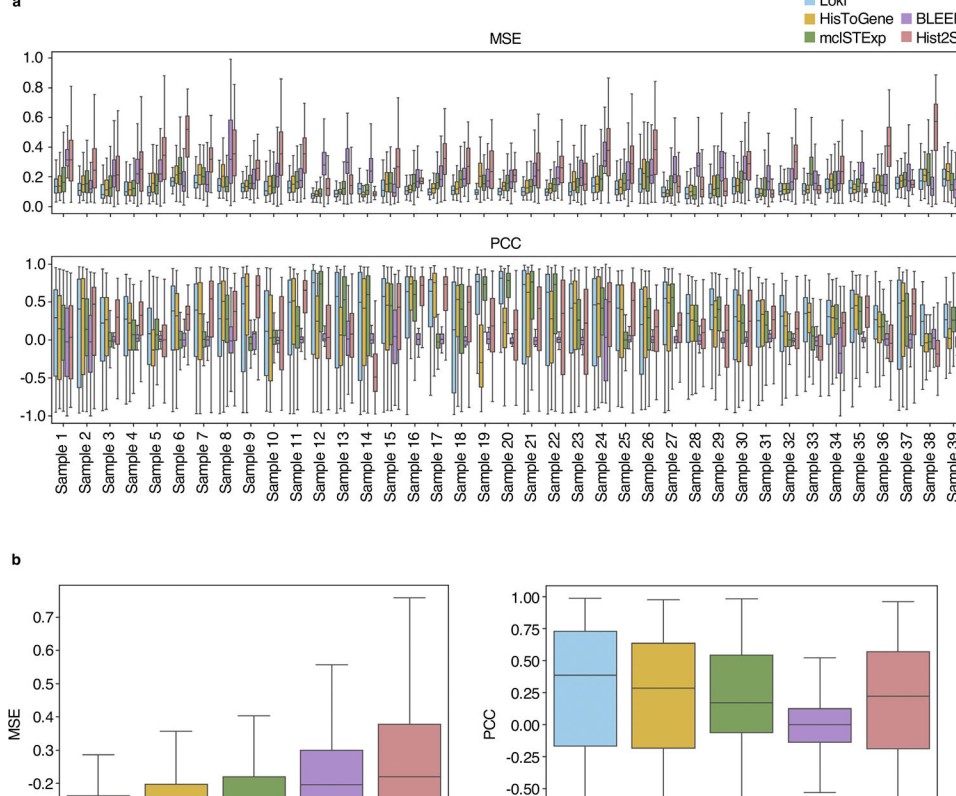

**Extended Data Fig. 9 |. Comparison of ST gene expression prediction performances.**
**a**, Comparison of ST gene expression prediction performances, represented by MSE and PCC respectively on 39 normal heart tissues using Loki, Hist2ST, HisToGene, BLEEP, and mclSTExp, respectively. In the box plots, the middle line represents the median, the box boundaries indicate the interquartile range, and the whiskers extend to data points within 1.5× the interquartile range. **b**, Summarized comparison of ST gene expression prediction performances, represented by MSE and PCC respectively across all samples using Loki, HisToGene, mclSTExp, BLEEP, and Hist2ST respectively. In the box plots, the middle line represents the median, the box boundaries indicate the interquartile range, and the whiskers extend to data points within 1.5× the interquartile range.

| | Function | Software | Fine-tune |
|---|---|---|---|
| Loki Align | H&E image to ST alignment | **Loki only** | Suggested |
| Loki Align | ST to ST alignment | Loki PASTE … | No need |
| Loki Decompose | scRNA-seq to H&E image | **Loki only** | Suggested |
| Loki Decompose | scRNA-seq to ST mapping | Loki Tangram … | No need |
| Loki Annotate | Tissue annotation by Bulk RNA-seq | **Loki only** | No need |
| Loki Annotate | Tissue annotation by marker genes | **Loki only** | No need |
| Loki Retrieve | H&E image-to-ST retrieval | **Loki only** | No need |
| Loki PredEx | ST gene expression prediction by H&E image | Loki HisToGene … | Suggested |

**Extended Data Fig. 10 |. Summary of the fine-tuning settings for downstream tasks.**
Recommendation settings for downstream tasks.

## Supplementary Material

Refer to Web version on PubMed Central for supplementary material.

## Acknowledgements

This work was supported in part by the grant R35GM150460 (to G.W.) from the National Institute of General Medical Sciences (NIGMS) and grant R01HL169204–01A1 (to L.L.) from the National Institutes of Health (NIH)-National Heart, Lung, and Blood Institute (NHLBI). K.W.B. is supported by NIH/National Institute of Neurological Disorders and Stroke (NINDS) award K22 NS112678, NIH/National Cancer Institute (NCI) award R01 CA284315 and Cancer Prevention and Research Institute of Texas (CPRIT) award RR220017. We acknowledge J. Chang in Houston Methodist Research Institute for support and assistance in facilitating access to clinical resources essential to this study.

## Data availability

The normal human small intestine dataset used for the tissue alignment task can be found in https://doi.org/10.1038/s41467-023-36071-5 (ref. 34). The human ovarian carcinosarcoma dataset used for the tissue alignment task can be found at https://doi.org/10.1016/j.xgen.2021.100065 (ref. 35). The human breast cancer dataset used for the tissue alignment task can be found at https://doi.org/10.1038/s41467-023-43458-x (ref. 37). The human colorectal cancer dataset including Visium, Visium-HD and scRNA-seq data of serial slices used for cell-type decomposition task can be found at https://doi.org/10.1101/2024.06.04.597233 (ref. 48). The TNBC scRNA-seq data used for the cell-type decomposition task can be found at https://doi.org/10.1038/s41467-018-06052-0 (ref. 51). The TNBC Xenium data generated in this study have been deposited in the Gene Expression Omnibus database under accession code GSE293199. The brain dataset including Visium data of serial slices used for cell-type decomposition task can be found at https://doi.org/10.1038/s41587-021-01139-4 (ref. 50). The brain scRNA-seq dataset used for cell-type

decomposition task can be found at https://doi.org/10.1038/s41586-018-0654-5 (ref. 49). The histology images of the heart failure patient dataset used for the tissue annotation task can be found at https://doi.org/10.1038/s41586-022-05060-x (ref. 80). The histology images of the normal human breast dataset used for the tissue annotation task can be found at https://doi.org/10.1038/s41586-023-06252-9 (ref. 79). The histology images of TCGA BRCA dataset used for the tissue annotation task are available from the NIH Genomic Data Commons (https://portal.gdc.cancer.gov/). The bulk RNA-seq data used for tissue annotation task are available from the Genotype-Tissue Expression Portal (https://gtexportal.org/home/) and TCGA (https://portal.gdc.cancer.gov/). CRC7k image patch data and labels can be found at Zenodo via https://doi.org/10.5281/zenodo.1214456 (ref. 84). WSSS4LUAD image patches and labels can be found at https://wsss4luad.grand-challenge.org/. LC25000 image patches and labels can be found at https://github.com/tampapath/lung_colon_image_set/. PatchCamelyon image patches and labels can be found at https://patchcamelyon.grand-challenge.org/. The validation and test datasets used for the image–transcriptomics retrieval task can be found in Supplementary Table 4. The normal human heart samples used for the ST gene expression prediction task can be found at https://doi.org/10.1038/s41586-023-06311-1 (ref. 81). The ST-bank database is available at https://github.com/GuangyuWangLab2021/Loki/.

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

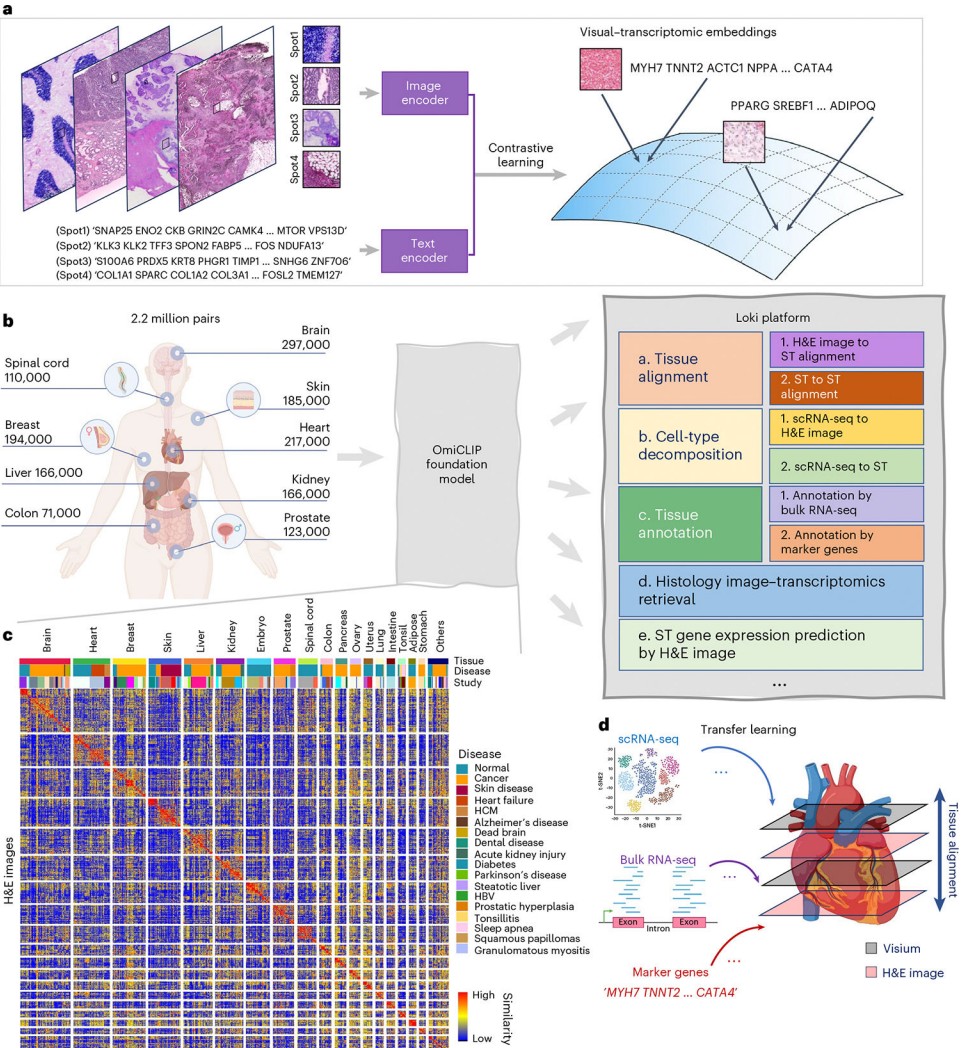

**Fig. 1 |. Overview of the study.**

**a**, The workflow of pretraining the OmiCLIP model with paired image–transcriptomics dataset via contrastive learning. **b**, Workflow of the Loki platform using the OmiCLIP foundation model as an engine. Left diagram illustrates the size of the training data in different organs. Right diagram lists the existing modules of the Loki platform, including tissue alignment, cell-type decomposition, tissue annotation, ST gene expression prediction and histology image–transcriptomics retrieval. Created in BioRender.com. **c**, The heat map represents image embeddings and transcriptomic embeddings similarity across various organs and disease conditions. The color of the heat map reflects the OmiCLIP's embedding similarities, with red indicating high similarity and blue indicating low similarity. HCM, hypertrophic cardiomyopathy; HBV, hepatitis B virus infection. **d**, Schematic illustration of Loki platform with transfer learning for 3D tissue analysis. Created in BioRender.com.

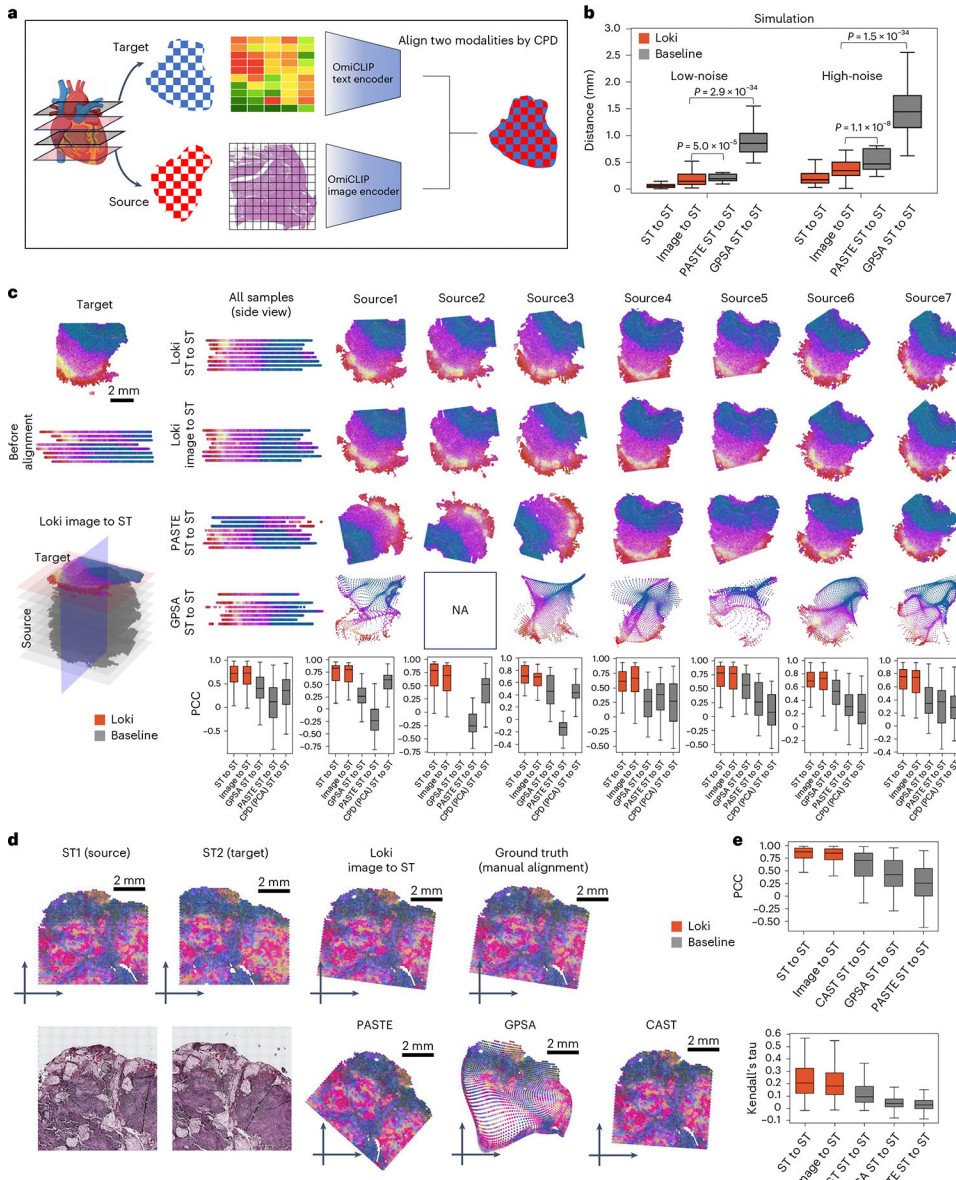

**Fig. 2 |. Tissue alignment.**

**a**, Schematic illustration of tissue alignment using ST and histology image with Loki Align. Created in BioRender.com. **b**, Performance comparison of tissue alignment on 100 low-noise and 100 high-noise simulated datasets, represented by the distance between ground truth and aligned simulated sample using Loki (ST-to-ST and image-to-ST) and baseline methods PASTE (ST-to-ST) and GPSA (ST-to-ST), respectively. *P* values were calculated using a one-sided Wilcoxon test. **c**, Alignment results on eight adjacent normal human small intestine samples using Loki (ST-to-ST and image-to-ST) and baseline methods PASTE (ST-to-ST), GPSA (ST-to-ST) and CPD (ST-to-ST), respectively. We colored the samples using the top three PCA components of OmiCLIP transcriptomic embeddings, mapped to red, green and blue color channels, respectively. For visualization, we stacked the eight samples together along the perpendicular axis before and after different alignment methods,

respectively, and visualized from the side view. The source2 that has no spatial variable gene selected by GPSA to run it is marked as 'not applicable' (NA). Box plots show the comparison of tissue alignment performances on these seven source samples respectively and combined, represented by the PCC (and Kendall's tau coefficient in Supplementary Fig. 1) of highly variable gene expression between target and source samples after alignment at the same location, using Loki and baseline methods (PASTE, GPSA and CPD using PCA embeddings as input), respectively. In the box plots, the middle line represents the median, the box boundaries indicate the interquartile range, and the whiskers extend to data points within 1.5 times the interquartile range. **d**, Tissue alignment of two adjacent human ovarian carcinosarcoma samples using Loki (ST-to-ST and image-to-ST) and baseline methods PASTE (ST-to-ST), GPSA (ST-to-ST) and CAST (ST-to-ST), respectively. We colored the samples as described in **c**. **e**, Alignment performance comparison using PCC and Kendall's tau coefficient of the highly expressed gene expression between the target sample and the source sample at aligned locations, using Loki (ST-to-ST and image-to-ST) and baseline methods PASTE (ST-to-ST), GPSA (ST-to-ST) and CAST (ST-to-ST), respectively. In the box plots, the middle line represents the median, the box boundaries indicate the interquartile range, and the whiskers extend to data points within 1.5 times the interquartile range; $n = 147$.

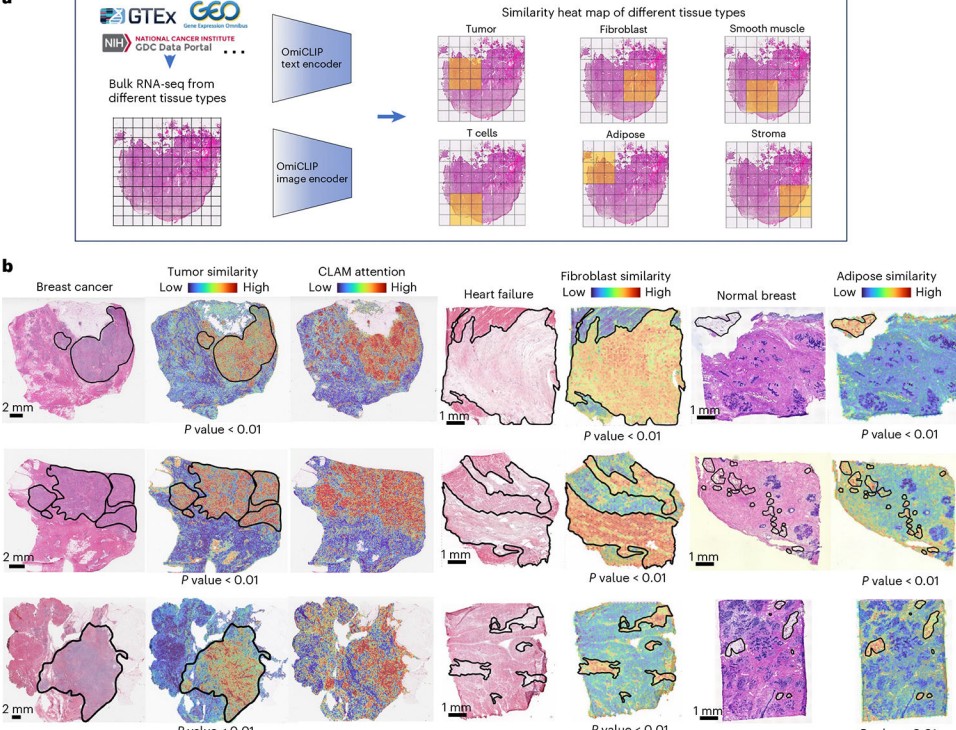

**Fig. 3 |. Tissue annotation using bulk RNA-seq data.**
**a**, Schematic illustration of tissue annotation using H&E image and reference bulk RNA-seq data from different sources, with OmiCLIP paired image and transcriptomic embeddings. **b**, Histology WSIs of breast cancer, heart failure and normal breast samples. The major tumor regions, fibroblast cell-enriched regions and adipose regions are annotated by pathology experts in black lines. Heat map shows the similarity of WSIs to the corresponding reference bulk RNA-seq of tumor, fibroblast and adipose, respectively. The color of the heat map reflects the similarities between WSIs and reference bulk RNA-seq data, with red indicating high similarity and blue indicating low similarity. CLAM attention heat maps were generated using CLAM with default parameters.

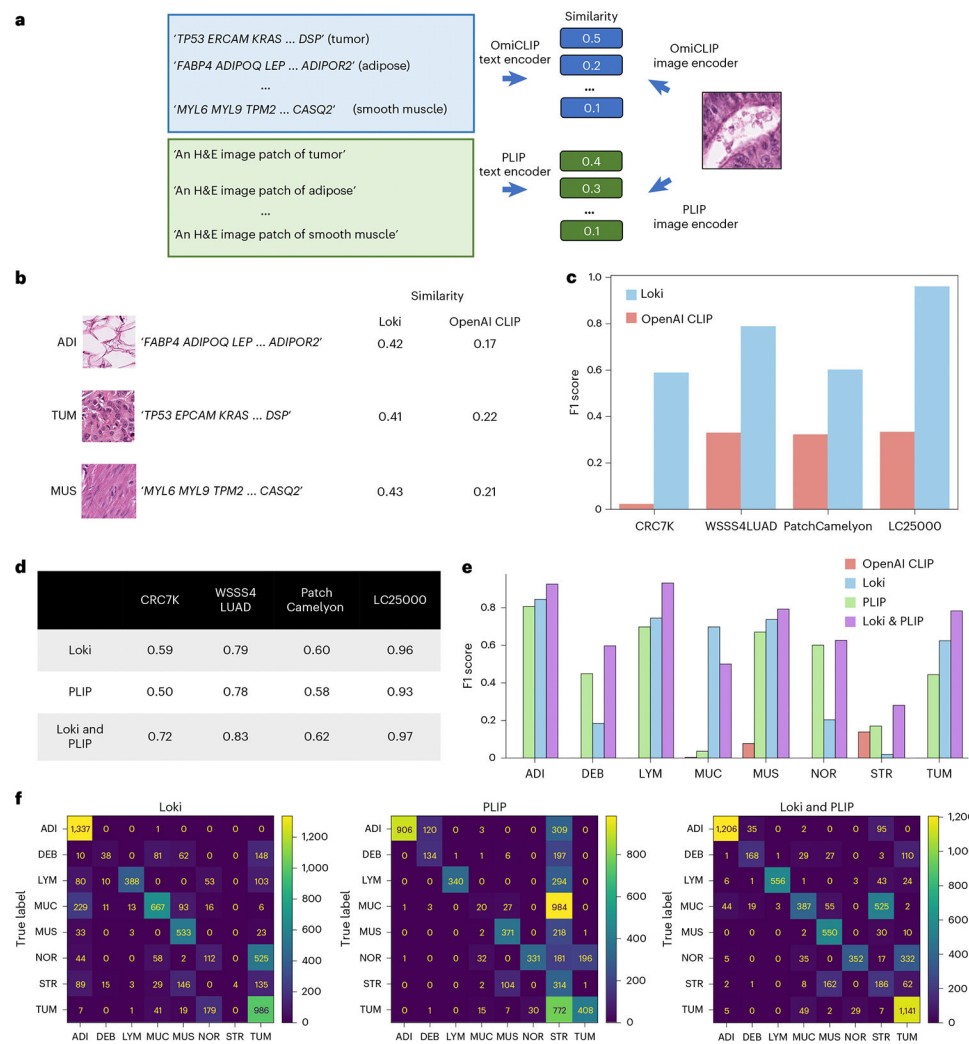

**Fig. 4 |. Tissue annotation using marker genes.**

**a**, Schematic illustration of tissue annotation using H&E image and reference marker genes. The annotation result is decided by choosing the candidate texts with the highest similarity score to the input image query. For Loki, we used the text content of marker gene symbols of each tissue type. For the PLIP model, we used the text content of natural language description of each tissue type. **b**, Examples of similarity scores of images and texts calculated by Loki and OpenAI CLIP model, respectively. **c**, Comparison of zero-shot performances, represented by weighted F1 scores, across four datasets using Loki and OpenAI CLIP, respectively. Number of test samples for each dataset: CRC7K ($n = 6{,}333$); WSSS4LUAD ($n = 10{,}091$); LC25000 ($n = 15{,}000$); and PatchCamelyon ($n = 32{,}768$). **d**, Comparison of zero-shot performances, represented by weighted F1 scores, across four datasets using Loki, PLIP and incorporating Loki and PLIP models by average similarity (shown in **a**; Methods), respectively. **e**, Comparison of zero-shot performances, represented by weighted F1 scores of each tissue type in the CRC7K dataset using OpenAI CLIP model, Loki, PLIP model and incorporating Loki and PLIP models, respectively. **f**, Confusion matrix of the CRC7K dataset using Loki (left), PLIP model (middle) and incorporating Loki

and PLIP models (right), respectively. The ground-truth labels are presented in rows and the predicted labels are presented in columns. ADI, adipose tissue; NOR, normal colon mucosa; TUM, colorectal carcinoma epithelium; LYM, lymphocytes; MUC, mucus; DEB, debris; MUS, smooth muscle; STR, cancer-associated stroma.

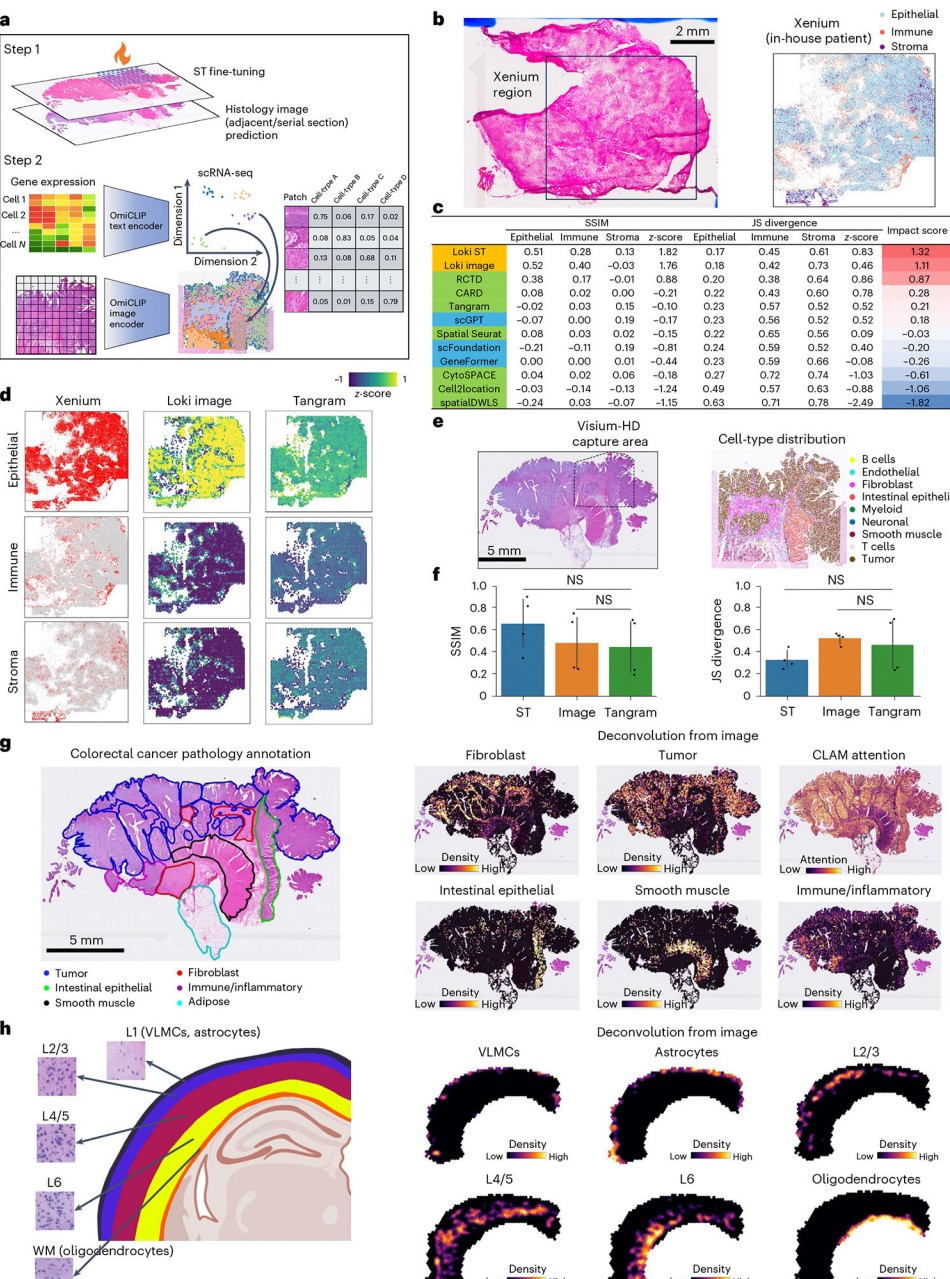

| | SSIM | | | | JS divergence | | | | Impact score |
|---|---|---|---|---|---|---|---|---|---|
| | Epithelial | Immune | Stroma | z-score | Epithelial | Immune | Stroma | z-score | |
| Loki ST | 0.51 | 0.28 | 0.13 | 1.82 | 0.17 | 0.45 | 0.61 | 0.83 | 1.32 |
| Loki image | 0.52 | 0.40 | −0.03 | 1.76 | 0.18 | 0.42 | 0.73 | 0.46 | 1.11 |
| RCTD | 0.38 | 0.17 | −0.01 | 0.88 | 0.20 | 0.38 | 0.64 | 0.86 | 0.87 |
| CARD | 0.08 | 0.02 | 0.00 | −0.21 | 0.22 | 0.43 | 0.60 | 0.78 | 0.28 |
| Tangram | −0.02 | 0.03 | 0.15 | −0.10 | 0.23 | 0.57 | 0.52 | 0.52 | 0.21 |
| scGPT | −0.07 | 0.00 | 0.19 | −0.17 | 0.23 | 0.56 | 0.52 | 0.52 | 0.18 |
| Spatial Seurat | 0.08 | 0.03 | 0.02 | −0.15 | 0.22 | 0.65 | 0.56 | 0.09 | −0.03 |
| scFoundation | −0.21 | −0.11 | 0.19 | −0.81 | 0.24 | 0.59 | 0.52 | 0.40 | −0.20 |
| GeneFormer | 0.00 | 0.00 | 0.01 | −0.44 | 0.23 | 0.59 | 0.66 | −0.08 | −0.26 |
| CytoSPACE | 0.04 | 0.02 | 0.06 | −0.18 | 0.27 | 0.72 | 0.74 | −1.03 | −0.61 |
| Cell2location | −0.03 | −0.14 | −0.13 | −1.24 | 0.49 | 0.57 | 0.63 | −0.88 | −1.06 |
| spatialDWLS | −0.24 | 0.03 | −0.07 | −1.15 | 0.63 | 0.71 | 0.78 | −2.49 | −1.82 |

**Fig. 5 |. Cell-type decomposition.**

**a**, Schematic illustration of tissue alignment using ST, reference scRNA-seq data and histology images with OmiCLIP paired transcriptomic and image embeddings after fine-tuning. **b**, H&E image of our in-house TNBC sample, characterized by Xenium into three major cell types: cancer epithelial, immune and stromal cells. **c**, Performance comparison of 12 decomposition methods using JS divergence, SSIM and impact scores. z-scores of JS divergence (or SSIM) across methods were calculated based on the average JS divergence (or SSIM) among cell types. The impact score of each method is the average of the z-score of JS divergence and SSIM (Methods). The green color indicates decomposition tools. The blue color indicates the performance of replacing OmiCLIP embeddings with other

transcriptomic foundation models' embeddings. **d**, Cell-type decomposition results on three major cell types of the TNBC sample using the image by Loki and using ST by Tangram, with Xenium data as ground truth. The color of the heat map reflects the $z$-score, calculated by the probability distribution of each cell type. **e**, H&E image of the human colorectal cancer sample and cell-type distribution within the Visium-HD capture area. **f**, Bar plot shows the accuracy of decomposition on four major cell types by Loki using ST or image mode, and by Tangram using ST. Error bars indicate the standard deviation and the center values represent the mean. For both JS divergence and SSIM, adjusted $P$ value > 0.1 using a two-sided Wilcoxon test. **g**, Whole-slide (20 mm × 13 mm) human colorectal cancer cell-type decomposition. Different tissue regions are annotated by the pathologist as ground truth. Heat map shows the cell-type distribution of fibroblast, tumor, intestinal epithelial, smooth muscle and immune/inflammatory cells, with color reflecting the density of each cell type. CLAM attention heat maps were generated using CLAM with default parameters. **h**, Cell-type decomposition results on the brain sample. Left, brain anatomic references with zoom-in H&E image patches of L1 (VLMCs, astrocytes), L2/3, L4/5, L6 and white matter (WM; oligodendrocytes), respectively. Created in BioRender.com. Right, heat map shows the cell-type distribution of VLMCs, astrocytes, L2/3, L4/5, L6 and oligodendrocytes, with color reflecting the distribution of each cell type.

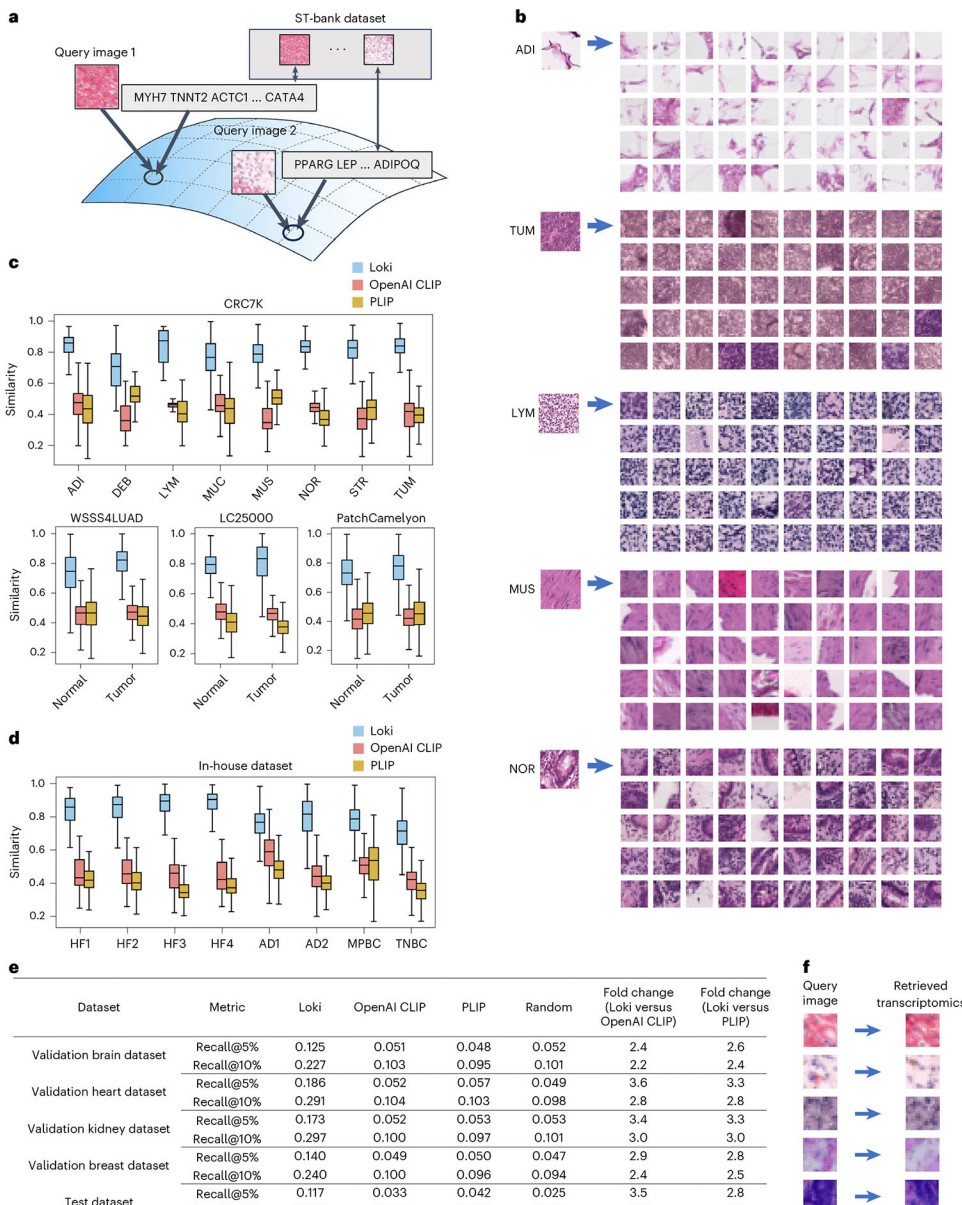

**Fig. 6 |. Image-to-transcriptomics retrieval.**
**a**, Schematic illustration of image-to-transcriptomics retrieval on the ST-bank dataset. **b**, Example image-to-transcriptomics retrieval results. For each example image from adipose tissue, colorectal adenocarcinoma epithelium, lymphocytes, smooth muscle and normal colon mucosa, the retrieved top 50 most similar transcriptomics are shown by the paired image from the ST-bank dataset. **c**, Image-to-transcriptomics retrieval similarity scores across the four validation datasets—CRC7K, WSSS4LUAD, LC25000 and PatchCamelyon —using Loki, OpenAI CLIP and PLIP. In the box plots, the middle line represents the median, the box boundaries indicate the interquartile range, and the whiskers extend to data points within 1.5 times the interquartile range. **d**, Image-to-transcriptomics retrieval similarity scores across the eight in-house human tissues: heart failure (HF), Alzheimer's disease (AD), metaplastic breast cancer (MPBC) and TNBC, using Loki, OpenAI CLIP

and PLIP. In the box plots, the middle line represents the median, the box boundaries indicate the interquartile range, and the whiskers extend to data points within 1.5 times the interquartile range. **e**, Image-to-transcriptomics retrieval evaluation across four validation datasets and one test dataset using Loki, OpenAI CLIP and PLIP, with random baseline. The top-K quantile most similar transcriptomics were retrieved. We report Recall@K for K ∈ {5%, 10%} (Methods). **f**, Example image-to-transcriptomics retrieval results. The retrieved transcriptomics are shown by the paired image.

