## [Peer Review File · Nature methods]

A visual–omics foundation model to bridge histopathology image with transcriptomics

Corresponding Author: Dr Guangyu Wang

Version 0:

Decision Letter:

6th Nov 2024

Dear Dr Wang,

Your Article, "A visual–omics foundation model to bridge histopathology image with transcriptomics", has now been seen by three reviewers. As you will see from their comments below, although the reviewers find your work of considerable potential interest, they have raised a number of concerns. We are interested in the possibility of publishing your paper in Nature Methods, but would like to consider your response to these concerns before we reach a final decision on publication. We therefore invite you to revise your manuscript to address these concerns.

We ask that you focus your revision on adding the requested benchmarking and additional quantitative performance assessments and metrics requested by the reviewers, in addition to addressing calls for clarification. We also ask that you provide the pre-trained model upon resubmission. Please note, because we do not know the outcome of the benchmarking results at this time, we cannot promise to send your paper back for review until we've carefully considered the revised version.

Link Redacted

We hope to receive your revised paper within three months. If you cannot send it within this time, please let us know. In this event, we will still be happy to reconsider your paper at a later date so long as nothing similar has been accepted for publication at Nature Methods or published elsewhere.

OPEN SCIENCE REQUIREMENTS

REPORTING SUMMARY AND EDITORIAL POLICY CHECKLISTS

DATA AVAILABILITY

All novel DNA and RNA sequencing data, protein sequences, genetic polymorphisms, linked genotype and phenotype data, gene expression data, macromolecular structures, and proteomics data must be deposited in a publicly accessible database, and accession codes and associated hyperlinks must be provided in the "Data Availability" section.

CODE AVAILABILITY

Please include a "Code Availability" subsection in the Online Methods which details how your custom code is made available. Only in rare cases (where code is not central to the main conclusions of the paper) is the statement "available upon request" allowed (and reasons should be specified).

MATERIALS AVAILABILITY

Authors reporting new chemical compounds must provide chemical structure, synthesis and characterization details. Authors

reporting mutant strains and cell lines are strongly encouraged to use established public repositories.

ORCID

Nature Methods is committed to improving transparency in authorship. As part of our efforts in this direction, we are now requesting that all authors identified as 'corresponding author' on published papers create and link their Open Researcher and Contributor Identifier (ORCID) with their account on the Manuscript Tracking System (MTS), prior to acceptance. This applies to primary research papers only. ORCID helps the scientific community achieve unambiguous attribution of all scholarly contributions. You can create and link your ORCID from the home page of the MTS by clicking on 'Modify my Springer Nature account'. For more information please visit <http://www.springernature.com/orcid>.

Sincerely,
Rita

Rita Strack, Ph.D.
Senior Editor
Nature Methods

Reviewers' Comments:

Reviewer #1 (Remarks to the Author):

Chen et al. presents a visual-omics foundation model, OmiCLIP, which is pretrained on 2.2 million paired tissue images and transcriptomic data. Building on OmiCLIP, the authors developed a platform called Loki, offering a variety of key downstream analyses, including tissue alignment and cell type decomposition. Overall, this work is interesting and contributes to the community. However, it appears that the results are not as promising as expected. Notably, small model trained on small datasets outperforms the foundation model in tasks like cell type decomposition. Moreover, I have concerns about the practical applications of the proposed foundation model, as the image quality in spatial transcriptomics can be low and inconsistent across samples due to technological limitations. In summary, this paper needs to be substantially improved before publication. In particular, the reliability and superiority of OmiCLIP need further validation. The authors should make the full codes, database ST-bank, and platform available so that our referees can access them before publication.

Major comments:

1. To date, several transcriptomics-based foundation models have been developed for single cell data, including GeneFormer (Theodoris et al., 2023), scGPT (Cui et al., 2024), scFoundation (Hao et al., 2024). The authors should conduct benchmark tests to compare the performance of OmiCLIP with these methods in common downstream tasks, such as cell type decomposition.
2. Similar to large language models, OmiCLIP creates 'sentences' for transcriptomics by concatenating top-expressed gene symbols. However, unlike text data, genes are unordered. I am not sure whether this approach is appropriate and whether it would lose some key biological information. Additionally, OmiCLIP ranks genes from high to low by expression levels. How is the transcriptomics data preprocessed before ranking? Given that sequencing depth varies across technologies. How do you address this issue? How is the data from different technological platforms standardized?
3. For clustering, the authors claim that both embeddings clustered similar tissue types. However, the conclusion is difficult to verify from the UMAPs (Extended Fig. 1), as the results appear unstable, with some seemingly random results. Are the color labels shared between both the image and transcriptomics embeddings? Moreover, legends should be provided for each figure to improve clarity.
4. For Loki Align, the authors select fine-tuning and zero-shot settings for different real-world datasets. If this is case, how should the users choose the appropriate settings when applying the model to their datasets? It is clear how to measure alignment accuracy in ST-to-ST scenarios, but how do you measure accuracy in image-to-ST scenarios?
5. Fine-tuning is a critical step for dataset- or task-specific analysis, yet the fine-tuning details for each task are only briefly mentioned. More detailed steps should be provided, including the loss function and fine-tuning framework.
6. For Loki Decompose, the authors compare it with baseline methods such as Tangram (Biancalani et al. 2021), Spaital Seurat (Stuart et al., 2022), CARD (Lu et al., 2021), and CytoSPACE (Vahid et al., 2023). However, it would be beneficial to also include Cell2location (Kleshchevnikov et al., 2022), SpatialDWLS (Dong et al., 2021), and RCTD (Jin et al., 2024), which have been shown to be the top-performing methods for cell type deconvolution in the benchmarking paper by Li et al. 2022.
7. The results in Loki Decompose are not particularly solid. In Fig. 3f, the authors highlight that Loki Decompose is comparable to Tangram results. However, the results obviously show that Loki Decompose underperforms Tangram in terms of both SSIM and JS. Additionally, in Fig.3g, some color bars are missing.
8. A single metric is insufficient to validate the results. The authors use MSE to evaluate the results for Loki Align and Loki Decompose, but relying on a single metric makes it difficult to ensure fairness. For a fair comparison, more metrics should be included.

9. OmiCLIP is demonstrated to have promising results in tissue annotation and cell type decomposition. However, the cell types in the tissue samples used in the experiments seem somewhat coarse. For example, the human colorectal cancer samples are annotated with only 6 cell types. This might not be particularly challenging for the model, as the features distributions between cell types may be distinct enough. To further enhance the model's reliability, testing the model on tissue samples with more fine-grained cell types could be beneficial.

10. The full codes, platform, and the database ST-bank are currently unavailable. The authors should make these resources accessible so that our referees can access them before publication.

Minor comments

1. Some references are missing, such as nonconvex optimization algorithms.

2. Some results cannot be found. For example, "Additionally, our predicted tumor cell density was consistent with the predictions by CLAM, a SOTA model specifically designed for tumor analysis in WSIs." Where are the results?

Reference

1. Theodoris, Christina V., et al. "Transfer learning enables predictions in network biology." *Nature* 618.7965 (2023): 616-624.
2. Cui, Haotian, et al. "scGPT: toward building a foundation model for single-cell multi-omics using generative AI." *Nature Methods* (2024): 1-11.
3. Hao, Minsheng, et al. "Large-scale foundation model on single-cell transcriptomics." *Nature Methods* (2024): 1-11.
4. Biancalani, Tommaso, et al. "Deep learning and alignment of spatially resolved single-cell transcriptomes with Tangram." *Nature methods* 18.11 (2021): 1352-1362.
5. Stuart, T., et al. Comprehensive integration of single-cell data. *cell* 177, 1888-1902. e1821 (2019).
6. Ma, Y. & Zhou, X. Spatially informed cell-type deconvolution for spatial transcriptomics. *Nature biotechnology* 40, 1349-1359 (2022).
7. Vahid, Milad R., et al. "High-resolution alignment of single-cell and spatial transcriptomes with CytoSPACE." *Nature biotechnology* 41.11 (2023): 1543-1548.
8. Kleshchevnikov, Vitalii, et al. "Cell2location maps fine-grained cell types in spatial transcriptomics." *Nature biotechnology* 40.5 (2022): 661-671.
9. Dong, Rui, and Guo-Cheng Yuan. "SpatialDWLS: accurate deconvolution of spatial transcriptomic data." *Genome biology* 22.1 (2021): 145.
10. Jin, Xing, et al. "RCTD: Reputation-Constrained Truth Discovery in Sybil Attack Crowdsourcing Environment." *Proceedings of the 30th ACM SIGKDD Conference on Knowledge Discovery and Data Mining*. 2024.
11. Li, Bin, et al. "Benchmarking spatial and single-cell transcriptomics integration methods for transcript distribution prediction and cell type deconvolution." *Nature methods* 19.6 (2022): 662-670.

Reviewer #1 (Remarks on code availability):

The authors only provided part of the codes and cannot find the pretrained model. Therefore, I cannot reproduce the results according to the codes.

Reviewer #2 (Remarks to the Author):

The study introduces **OmiCLIP**, a dual-encoder foundation model linking H&E images with transcriptomic data from Visium samples. By transforming the top-expressed genes within tissue patches into "sentences," this method adapts CLIP-style contrastive learning to bring histology and transcriptomic data into a shared embedding space. This shared space enables five downstream applications within their **Loki** platform:

1. **Loki Align** \- Aligns spatial transcriptomics (ST) and H&E images through an adapted coherent point drift method.
2. **Loki Decompose** \- Uses OmiCLIP embeddings with Tangram for cell type decomposition.
3. **Loki Annotate** \- Embeds bulk RNA-seq data, allowing similarity comparisons with regions in H&E images or disease-specific gene sets.
4. **Loki Retrieve** \- Accepts an H&E image to identify similar ST panels from the training or finetuning data.
5. **Loki PredEx** \- Predicts ST gene expression from H&E by averaging gene expression across the ST slides predicted to be most similar.

Additionally, they introduce **ST-bank**, a dataset of 2.2 million tissue patches from over 1,000 Visium samples, to support multi-modal AI integration of histology and omics data.

Overall, I have major concerns about the methodology and the data analysis. Many of these issues are due to inadequate descriptions of key details of the method and analysis. However, even when things were clear, I was confused about several of the interpretations of the results. While the method is very intriguing and novel, the writing and results need major improvements.

Major Concerns About Methodology/Approach

1. **Evaluation of UMAP Embeddings and Contrastive Learning**:

* The reference to UMAP embeddings in Extended Data Fig. 1 is misleading. The claim that similar tissue types are clustered across modalities doesn't align with the actual figure, which shows UMAP embeddings before and after contrastive learning, organized by tissue type. Furthermore, it appears that contrastive learning negatively impacts the transcriptomics embeddings, casting doubt on the efficacy of the approach. Also unclear is the source of the spatial transcriptomics (ST) labels—these are

reportedly derived from Leiden clustering on UMAP embeddings but lack clear description in the Method section. It seems like their initial embeddings of ST data is fairly well clustered, except that the annotations are incorrect.

2. **Clarifications on Loki Decompose**:

* The description of Loki Decompose needs refinement to highlight that it essentially employs Tangram's nonconvex optimization approach with OmiCLIP embeddings rather than raw gene expression data. Further, critical details in the methods section are challenging to follow: the role of PCA, computation of expected cell counts per spot (and if these are also given to base Tangram), and the concept of adapting mappings at the cell cluster level all require elaboration. Moreover, the conflicting statements about the cosine similarity loss function, first described as minimizing and then as maximizing, add unnecessary ambiguity. Consistency in the Tangram-based comparisons, using identical adjustments for both methods, would improve the validity of the assessment.

Major Concerns About Data Analysis

1. **Baseline and Metric Selection for Loki Align**:

* For Loki Align, a fairer baseline would involve applying the Coherent Point Drift method to PCA embeddings of the transcriptomic data, as it remains unclear whether the alignment accuracy results from the OmiCLIP embeddings themselves or simply from a superior registration method compared to PASTE or GPSA. Additionally, using Pearson correlation, a standard metric for transcriptomic comparison, may be more appropriate than MSE, especially when evaluating transcriptomic alignment accuracy.

2. **Interpretability of Figure 2**:

* Figure 2 contains notable visual ambiguities. The right-side plots in Fig. 2d are improperly framed, obscuring interpretation. Additionally, the alignment for Source6 in Fig. 2c appears mirrored on the X-axis, an effect that PASTE's method typically doesn't perform. If intentional, this mirroring needs clarification to explain its relevance to the alignment method.

3. **Metric Choice and Consistency in Loki Decompose**:

* The use of MSE for comparing cell type proportions in Fig. 3c, split by cell type, is unconventional and confusing, particularly for comparing spatial spot vectors. A unified approach would be beneficial across evaluations, as the use of JS divergence and SSIM in Fig. 3e is inconsistent with the preceding analyses in 3c-d, despite targeting the same evaluation objective.

4. **Evaluation Strategy for Loki Retrieve**:

* The evaluation approach for Loki Retrieve is problematic. Rather than comparing retrieved images with unpaired image data, it would be far more informative to use images that have paired transcriptomic data. This would offer a direct assessment of the model's ability to retrieve accurate transcriptomic information from images. As it stands, the title and evaluation give a misleading impression about the model's performance on this task.

* Furthermore, in looking at the python notebook, the retrieved images do not look like they would necessarily have a similar number of cells, or the same composition as the query images. It doesn't seem like the model would be able to accurately represent the transcriptomic profile based on the query image.

5. **F1 Score in Loki Annotate**:

* The computation of the F1 score in Fig. 5c is unclear, particularly concerning how cosine similarity data produces an F1 score. Additional explanation of the scoring framework and its connection to cosine similarity would be valuable to substantiate this metric's relevance.

6. **Interpretation of UMAP and Embedding Quality**:

* The UMAP-based evaluation in Supp Fig. 2 is confusing, with mismatches between annotations in the UMAP and the confusion matrix, particularly for misannotated cell types. This discrepancy complicates the assessment of OmiCLIP's performance, as the annotations should align for a valid comparison. Further investigation into why OmiCLIP generates two distinct clusters for TCGA LUAD, shown in Supp Fig. 3, would also aid interpretation, as this outcome suggests potential biases or unique properties within this embedding.

7. **Potential Overestimation of CLIP Performance**:

* The unexpectedly strong performance of CLIP for single-cell annotation raises questions about the advantages and potential limitations of the gene expression textual representation method. Exploring ways to evaluate these representations independently, such as UMAP visualizations based on CLIP and OmiCLIP embeddings, would clarify the distinct advantages or drawbacks of this approach.

Minor Comments

1. **Visual Clarity in Figures**:

* Fig. 3d suffers from stretching artifacts and inconsistent color scales across methods, which complicates direct comparisons. Standardizing color bars across methods would facilitate more straightforward interpretation of results.

2. **Organization of Results Section**:

* Loki Annotate, which is arguably the most compelling task within the results, is positioned third despite its prominence in the figures. Reordering this section could better emphasize its significance.

3. **Terminology Consistency in Loki Decompose**:

* Describing Loki Decompose as an application of Tangram with OmiCLIP embeddings instead of gene expression data would enhance clarity and reflect the underlying methodology more accurately.

4. **Code base lacks methods to format Visium data**:

* The code as it is currently given does not have any methods to take in the Visium slide and properly segment the histology image and convert the gene expression data into the text-based format. This is a critical piece of processing that could impact the results and it is absent.

Reviewer #2 (Remarks on code availability):

Code base lacks methods to format Visium data:

* The code as it is currently given does not have any methods to take in the Visium slide and properly segment the histology image and convert the gene expression data into the text-based format. This is a critical piece of processing that could impact the results and it is absent.

There also was no README file with instructions for installing (although it doesn't seem like this is an actual software package yet, since it just uses other libraries so far). I think it would be much better if they created a python package for Loki, since it would be much easier to just call methods like `loki.retrieve()` instead of copying a python notebook and adapting it to your needs.

Since this is poised as a foundation model with wide applicability (especially within medical imaging fields), the code should really be more user-friendly.

Reviewer #3 (Remarks to the Author):

The manuscript introduces OmiCLIP, a dual-encoder foundation model designed to integrate histopathology images with transcriptomics data, alongside the Loki platform for tasks such as tissue alignment, cell type decomposition, tissue annotation, and ST gene expression prediction. The authors demonstrate superior performance when compared to several methods across multiple datasets, particularly in integrating visual and omics data. While the modeling ideas are innovative, the authors need to carefully validate the pre-training approach and demonstrate the superiority of their method by comparing it with more recent methods. Additionally, as an infrastructure model, greater transparency around the pre-training data is necessary.

1. For pre-training, how did the authors handle the batch effect between different data resources? It is quite important to guarantee the model has good generalization ability without fine-tuning.
2. The authors fine-tuned the pre-trained model on many downstream tasks, leaving the meaning of pre-training unclear. First, a table summarizing which downstream task needs further fine-tuning model would help readers. Second, it is important to compare the performance of pure pre-training, pure training from scratch for downstream, and pre-training + fine-tuning to show the benefit of pre-training. Also, it would be better if the author could guide readers on when further fine-tuning is needed.
3. The authors use the Calinski-Harabasz (CH) score for evaluation, but it is unclear how the ground truth labels were determined for this metric. A more detailed explanation in the methods section is needed.
4. Spatial transcriptomics data typically include both image and expression information. It would be valuable to explore whether Loki Align could leverage both modalities simultaneously for alignment, rather than relying on just one. The inclusion of more recent methods such as STAlign and CAST for comparison is also recommended.
5. Since Loki Align can be performed based on pure image, it would be better if the authors could demonstrate its performance on aligning 10X Visium and Xenium slides to show the model's contribution as a bridge between image and omic.
6. PLIP should also be used for benchmarking the retrieval task. Moreover, the benefits and significance of retrieval should be discussed in greater depth. For instance, demonstrating how the retrieved gene expression patterns map onto the queried WSI (whole slide image) would provide more tangible insights into the retrieval functionality.
7. New baselines should be included for Loki PredEx. Works like BLEEP (Spatially Resolved Gene Expression Prediction from Histology Images via Bi-modal Contrastive Learning) and mclSTExp (Multimodal contrastive learning for spatial gene expression prediction using histology images) have a similar clip-retrieval style, also scalable and already showed a higher performance than HisToGene. The authors should compare Loki with these methods.
8. Related to the last comment, I suggest adding Pearson correlation across spots as an additional evaluation metric, which would better reflect the accuracy of recovering spatial gene expression patterns.
9. I appreciate the authors on their ST-bank dataset construction and I think it would be a valuable resource to the community. However I only found a demo dataset in code and there is no availability in the manuscript to access the data. I think it should be released as a condition of publication.
10. An intuitive alternative to the current method is to combine existing single-cell foundation models with pathology foundation models. The authors should discuss why they chose to model omics data as text, rather than directly using gene expression values.
11. I recommend the authors add the introduction of tasks in supplementary notes to the introduction of the main manuscript. This would help better contextualize the generalizable contributions of this work and make it easier for the readers to grasp the scope of the research.

Reviewer #3 (Remarks on code availability):

I installed the environment, ran the code on Codeocean, and also reviewed the code in the supplementary material. Codes for downstream applications are executable and documented, but are without the pre-training one.

Version 1:

Decision Letter:

Our ref: NMETH-A58071A

7th Mar 2025

Dear Guangyu,

Thank you for submitting your revised manuscript "A visual-omics foundation model to bridge histopathology image with transcriptomics" (NMETH-A58071A). It has now been seen by the original referees and their comments are below.

We sent your responses to the reviewers, and they agree that a manuscript revised as described would address these concerns. Therefore we'll be happy in principle to publish your paper in Nature Methods, pending these revisions to satisfy the referees' final requests and some to comply with our editorial and formatting guidelines. Please provide the point-by-point rebuttal upon resubmission.

TRANSPARENT PEER REVIEW

Nature Methods offers a transparent peer review option for new original research manuscripts submitted from 17th February 2021. We encourage increased transparency in peer review by publishing the reviewer comments, author rebuttal letters and editorial decision letters if the authors agree. Such peer review material is made available as a supplementary peer review file.

Please state in the cover letter 'I wish to participate in transparent peer review' if you want to opt in, or 'I do not wish to participate in transparent peer review' if you don't. Failure to state your preference will result in delays in accepting your manuscript for publication.

ORCID

Sincerely,
Rita

Rita Strack, Ph.D.
Senior Editor
Nature Methods

Reviewer #1 (Remarks to the Author):

The authors have addressed most comments provided while some comments are still not adequately addressed like comment 3. Specifically, the authors have not answered the reason why the results appear unstable, with some seemingly random results.

Reviewer #1 (Remarks on code availability):

The authors has provided the codes via private link and jupyter notebook files. README file with enough instructions for installing and running the application was provided.

I haven't run the code due to the lacking of GPU resources. However, it appears the code and data are reliable and reproducible.

Additionally, while the author provided private link, it is still recommended to make code and data public.

Reviewer #2 (Remarks to the Author):

Overall, I am very happy with the progress made on the manuscript. The revisions clearly demonstrate that substantial effort was invested, resulting in a much-improved presentation and analysis.

Loki Align

The analysis on Loki Align has been significantly enhanced by the addition of extra metrics and the inclusion of CPD as a baseline, which substantially improves this section. However, I am curious as to why only two principal components were used in the PCA analysis. An explanation of this decision would help clarify whether the chosen dimensionality captures the essential variance or if additional components might provide further insights.

Additionally, the final sentence of this section—stating that “Altogether, Loki Align serves as a generalized cross-modality alignment tool for spatial tissue analysis, reducing costs associated with spatial transcriptomics sequencing”—feels somewhat disconnected from the main motivation. This should be changed to ensure that it aligns clearly with the overall objectives of Loki Align.

Loki Decompose

The conclusion for Loki Decompose remains somewhat weak. It would be useful to more explicitly describe the scenarios in which one might prefer using Loki Decompose for extracting cell type proportions from images, particularly in cases where acquiring the transcriptomic profile of the spots is not necessary. Clarifying these use cases would enhance the reader's understanding of its practical applications.

Loki Retrieve

The new analysis presented for Loki Retrieve raises a couple of points that need further clarification. First, what is the random baseline for Recall@5% in this setting? Including this information would allow readers to better contextualize the performance of Loki Retrieve relative to competing methods such as OpenAI CLIP and PLIP.

Secondly, while I understand that Loki Retrieve is not a generative model, the manuscript would benefit from a clearer discussion of its biological motivation. Specifically, why is retrieving similar images a valuable downstream task from a biological standpoint, especially in contrast to Loki PredEx? Elaborating on this aspect could help underscore the relevance and novelty of the approach.

Supplementary Figures

In Supplementary Figure 2, I was surprised to see many T cells classified as B cells, given that these are distinct cell types in the UMAP. It would be helpful to provide some intuition or discussion on why this misclassification occurs, highlighting any challenges inherent in distinguishing these cell types. Moreover, both Supplementary Figures 1 and 2 would benefit from a revised color scheme that more clearly differentiates B cells from T cells, as these are key cell types that warrant clear visual separation.

Extended Data Figures

The modifications to Extended Data Fig. 2 are a welcome improvement. However, I suggest that Extended Data Fig. 1b be reformatted to more closely resemble Extended Data Fig. 2b, which effectively shows the UMAPs before and after contrastive learning using ground truth labels for coloring. This change would provide a more informative visualization and facilitate a direct comparison of the effects of contrastive learning.

Cancer Patient Risk Stratification

The section on cancer patient risk stratification appears to be biologically significant. Considering its potential impact, it might be worth discussing whether this section should remain in the supplementary material or be promoted to the main text, as it could provide valuable insights that are more compelling than some of the other downstream tasks currently highlighted.

In summary, while the manuscript has improved considerably, addressing these specific points would further enhance its clarity, impact, and overall scientific contribution.

Reviewer #2 (Remarks on code availability):

The additional links for the source code gave me confidence that it will be possible to build off of Loki and reproduce their results. While I did not install the code, a README is available and has detailed instructions. Furthermore, the inclusion of python notebooks for many of the downstream tasks serve as good tutorials.

Reviewer #3 (Remarks to the Author):

I would like to thank the authors for their thorough responses to my comments; they have addressed most of my concerns. I am also pleased to see that the model weight and pre-training data will be made available to the community.

I think the paper is now much stronger, but I have a few additional comments:

1. In Revised Extended Data Fig. 17, the Loki PredEx module is missing. Could the authors please include it?
2. In the Loki align part, why the two modalities embeddings didn't outperform the single one? I suggest adding an explanation for this result.
3. On line 1005, the authors state, "The OmiCLIP model was fine-tuned on the training set for 10 epochs." However, line 606 mentions that "Loki PredEx avoids these resource-intensive training needs, providing a more efficient alternative." Since OmiCLIP is a foundation model, it would typically require more resources to train compared to smaller models. Could the authors clarify this?
4. It is difficult to compare the performance across methods from Extended Data Fig. 16. It would be helpful to summarize the performance across all samples in a single figure for better comparison. Additionally, Since there is not a significant performance gain in the PredEx module, I believe it would be valuable for the authors to provide some insights into the success and failure cases of OmiCLIP.
5. Since fine-tuning is necessary for several of the Loki functions, I suggest the authors make the code more user-friendly. Upon reviewing the code, I noticed that the fine-tuning code is available via the command line but not integrated into the API.

Version 2:

Decision Letter:

15th Apr 2025

Dear Guangyu,

I am pleased to inform you that your Article, "A visual–omics foundation model to bridge histopathology image with transcriptomics", has now been accepted for publication in Nature Methods. The received and accepted dates will be September 30, 2024 and April 15, 2025. This note is intended to let you know what to expect from us over the next month or so, and to let you know where to address any further questions.

Please note, I updated the title of your paper to: A visual–omics foundation model to bridge histopathology with spatial transcriptomics. If you want to change it again, you will have an opportunity on the proofs.

Over the next few weeks, your paper will be copyedited to ensure that it conforms to Nature Methods style. Once your paper is typeset, you will receive an email with a link to choose the appropriate publishing options for your paper and our Author Services team will be in touch regarding any additional information that may be required. It is extremely important that you let us know now whether you will be difficult to contact over the next month. If this is the case, we ask that you send us the contact information (email, phone and fax) of someone who will be able to check the proofs and deal with any last-minute problems.

To assist our authors in disseminating their research to the broader community, our SharedIt initiative provides you with a unique shareable link that will allow anyone (with or without a subscription) to read the published article. Recipients of the link

with a subscription will also be able to download and print the PDF.

If you are active on Twitter/X or Bluesky, please e-mail me your and your coauthors' handles so that we may tag you when the paper is published.

Best regards,
Rita

Rita Strack, Ph.D.
Senior Editor
Nature Methods

Visit the Springer Nature Editorial and Publishing website at http://editorial-jobs.springernature.com?utm_source=ejP_NMeth_email&utm_medium=ejP_NMeth_email&utm_campaign=ejp_Nmeth for more information about our career opportunities. If you have any questions please click [here](mailto:editorial.publishing.jobs@springernature.com).

Open Access This Peer Review File is licensed under a Creative Commons Attribution 4.0 International License, which permits use, sharing, adaptation, distribution and reproduction in any medium or format, as long as you give appropriate credit to the original author(s) and the source, provide a link to the Creative Commons license, and indicate if changes were made. In cases where reviewers are anonymous, credit should be given to 'Anonymous Referee' and the source.

Summary: We would like to first thank the reviewers for the rich feedback and constructive comments. Below is the summary of our main changes. We appreciate that all reviewers acknowledged the significance and novelty of our study. For example:

- *“Overall, this work is interesting and contributes to the community.” (Reviewer #1)*
- *“the method is very intriguing and novel.” (Reviewer #2)*
- *“modeling ideas are innovative.” (Reviewer #3)*
- *“I appreciate the authors on their ST-bank dataset construction and I think it would be a valuable resource to the community.” (Reviewer #3)*

Second, all reviewers have read our manuscript carefully and provided many constructive suggestions. The main comments include improving benchmarking, adding quantitative performance assessments, and clarifying details regarding the methods and analyses.

We have now adequately addressed those concerns in the revised manuscript. Briefly:

1. For the alignment task, we added the benchmark of CAST (revised Figure 2d-e) and CPD using PCA embeddings of the transcriptome instead of OmiCLIP transcriptomic embeddings (revised Figure 2c). We also replaced the MSE score with PCC and Kendall's tau to evaluate the performance. Moreover, we added breast cancer benchmark to evaluate the performance to align Visium data and Xenium data.
2. For the decomposition task, we added the benchmark of Cell2location, SpatialDWLS, RCTD, GeneFormer, scGPT, and scFoundation (revised Figure 5c). We used SSIM and JS divergence to evaluate the performance in all decomposition case studies. Moreover, we added new benchmark data, brain cortex tissue, as a more challenging case study to evaluate Loki's performance.
3. For the retrieval task, we added the PLIP model as a benchmark (revised Figure 6c-e). We also added an evaluation for image-to-transcriptomics retrieval by calculating the rank of the correct pair using Recall@K.
4. For the gene expression prediction task, we added the benchmark of BLEEP, and mclSTExp (revised Extended Data Fig. 16). We also added PCC as another evaluation metric.
5. We tested the robustness of OmiCLIP in the aspect of image quality and sequencing depth (revised Extended Data Fig. 3).
6. We substantially improved descriptions of key details of the method and analysis.
7. We reordered the Loki Annotation to the second task to emphasize its significance as suggested by Reviewer #2.
8. We added a table to summarize the recommendation settings for fine-tuning the model in different tasks. (revised Extended Data Fig. 17).
9. We provided a user-friendly package, a Readme file, pre-train weights (<https://drive.google.com/file/d/1UTJKAqoiUT55h9sA1Ee8iJeOcINLBvWD/view?usp=sharing>) and ST-bank database (https://drive.google.com/drive/folders/1J15cO-pXTwkTjRAR-v-_nQkqXNfcCNn3?usp=sharing). We updated a website to introduce Loki platform including functions and notebooks in the “source_code” file of Additional Review Material.

With these improvements in this study, we believe that the manuscript has been significantly strengthened. Please see below for our detailed point-by-point responses. We have also highlighted the major revisions in yellow color in the revised manuscript.

Response to Reviewer #1:

Remarks to the Author:

Chen et al. presents a visual-omics foundation model, OmiCLIP, which is pretrained on 2.2 million paired tissue images and transcriptomic data. Building on OmiCLIP, the authors developed a platform called Loki, offering a variety of key downstream analyses, including tissue alignment and cell type decomposition. Overall, this work is interesting and contributes to the community. However, it appears that the results are not as promising as expected. Notably, small model trained on small datasets outperforms the foundation model in tasks like cell type decomposition. Moreover, I have concerns about the practical applications of the proposed foundation model, as the image quality in spatial transcriptomics can be low and inconsistent across samples due to technological limitations. In summary, this paper needs to be substantially improved before publication. In particular, the reliability and superiority of OmiCLIP need further validation. The authors should make the full codes, database ST-bank, and platform available so that our referees can access them before publication.

Response: Thank you very much for your valuable comments. We are delighted that Reviewer #1 found our method interesting and recognized its potential contribution to the community. We also appreciate the concerns raised about the practical application of Loki, particularly regarding the variability in image quality across samples due to technological limitations. We trained the OmiCLIP model using 2.2 million tissue patches collected from 1,007 samples spanning 32 organs, derived from 113 studies with a wide range of image quality (**Response Figure 1a; Revised Extended Data Fig. 3a**). Furthermore, we applied image augmentation techniques during training to enhance the model's robustness.

To further evaluate the reliability of OmiCLIP, we generated a simulation dataset with low-quality images by applying Gaussian noise to raw H&E images. Using this dataset, we compared the similarity scores between the paired transcriptomic and original image embeddings, with paired transcriptomic and simulated low-quality image embeddings, which were encoded by OmiCLIP's image and transcriptomic encoders. PLIP and OpenAI CLIP were used as benchmarks for this analysis. The results demonstrated that OmiCLIP is robust to variations in image quality. This additional analysis has been incorporated into the manuscript on lines 112-119.

We have also addressed the reviewer's additional comments through detailed, point-by-point responses below.

Response Figure 1 (Revised Extended Data Fig. 3a, b): **a**, Example image with low quality region marked in red line and simulated low quality image by adding gaussian noise. **b**, Cosine similarity of paired transcriptomic and image embeddings using OmiCLIP (original image and simulated low quality image), PLIP (original image), and OpenAI CLIP (original image). In the box plots, the middle line represents the median, the box boundaries indicate the interquartile range, and the whiskers extend to data points within 1.5× the interquartile range.

Major comments:

Question 1. To date, several transcriptomics-based foundation models have been developed for single cell data, including GeneFormer (Theodoris et al., 2023), scGPT (Cui et al., 2024), scFoundation (Hao et al., 2024). The authors should conduct benchmark tests to compare the performance of OmicCLIP with these methods in common downstream tasks, such as cell type decomposition.

Response: Thank you for this insightful comment. In response, we have included a comparison between our model and GeneFormer, scGPT, and scFoundation in the cell type decomposition task. Specifically, we replaced the OmiCLIP gene expression embeddings with those generated by GeneFormer, scGPT, and scFoundation, and conducted the cell type decomposition analysis on our TNBC patient sample. To evaluate the performance, we calculated Jensen-Shannon (JS) divergence and structure similarity index measure (SSIM) scores following the suggestion of reviewer #2. These metrics were calculated by comparing the predicted cell type proportions to the ground truth derived from the Xenium data. Since JS divergence and SSIM operate on different scales, we standardized their values by calculating z-scores among different methods (details in Methods). To ensure consistency in interpretation, we inverted the z-score for JS divergence (i.e., multiplied by -1), as lower JS divergence indicates better performance. Finally, we averaged the z-scores of JS divergence and SSIM to calculate an overall impact score, which provides a unified metric to compare decomposition performance across methods. Results showed that scGPT, scFoundation, and GeneFormer ranked 6th, 8th, and 9th, respectively. This additional analysis has been incorporated into the manuscript on lines 434-451.

	SSIM				JS Divergence				Impact score
	Epithelial	Immune	Stroma	Z-score	Epithelial	Immune	Stroma	Z-score	
Loki ST	0.51	0.28	0.13	1.82	0.17	0.45	0.61	0.83	1.32
Loki Image	0.52	0.40	-0.03	1.76	0.18	0.42	0.73	0.46	1.11
RCTD	0.38	0.17	-0.01	0.88	0.20	0.38	0.64	0.86	0.87
CARD	0.08	0.02	0.00	-0.21	0.22	0.43	0.60	0.78	0.28
Tangram	-0.02	0.03	0.15	-0.10	0.23	0.57	0.52	0.52	0.21
scGPT	-0.07	0.00	0.19	-0.17	0.23	0.56	0.52	0.52	0.18
Spatial Seurat	0.08	0.03	0.02	-0.15	0.22	0.65	0.56	0.09	-0.03
scFoundation	-0.21	-0.11	0.19	-0.81	0.24	0.59	0.52	0.40	-0.20
GeneFormer	0.00	0.00	0.01	-0.44	0.23	0.59	0.66	-0.08	-0.26
CytoSPACE	0.04	0.02	0.06	-0.18	0.27	0.72	0.74	-1.03	-0.61
Cell2location	-0.03	-0.14	-0.13	-1.24	0.49	0.57	0.63	-0.88	-1.06
spatialDWLS	-0.24	0.03	-0.07	-1.15	0.63	0.71	0.78	-2.49	-1.82

Response Figure 2 (Revised Fig. 5c): Table of JS divergence, SSIM, and impact scores of 12 methods. z-score of JS divergence (or SSIM) across methods was calculated by the average JS divergence (or SSIM) among cell types. The impact score of each method is the average of the z-score of JS divergence

and SSIM. The green color indicates decomposition tools. The blue color indicates the performance of replacing OmiCLIP embeddings with other transcriptomic foundation models' embeddings.

Question 2. Similar to large language models, OmiCLIP creates 'sentences' for transcriptomics by concatenating top-expressed gene symbols. However, unlike text data, genes are unordered. I am not sure whether this approach is appropriate and whether it would loss some key biological information. Additionally, OmiCLIP ranks genes from high to low by expression levels. How is the transcriptomics data preprocessed before ranking? Given that sequencing depth varies across technologies. How do you address this issue? How is the data from different technological platforms standardized?

Response: Thank you for this insightful comment. We understand the concern about the unordered nature of genes when creating "sentences" for transcriptomics and its potential impact on capturing key biological information. However, recent studies provide strong support for using text-based representations of transcriptomics data as effective inputs for large language models. For example, the study by Mengzhou Hu et al. (*Nature Methods*, 2024 a)¹ demonstrated that the effective use of "gene sentences" constructed from transcriptomic data, as inputs for large language models such as GPT-4 and Llama can discover gene set function. Wenpin Hou et al. also highlight that "gene sentences" of marker genes in single-cell RNA sequencing data can accurately annotate cell types by GPT-4 (*Nature Methods*, 2024 b)². Additionally, the work of GenePT (*Nature Biomedical Engineering*, 2024)³ illustrates that similar text-like inputs can be effectively encoded to represent cellular states, further supporting the "gene sentences" approach. These studies indicated that although genes in transcriptomics data are inherently unordered compared to structured text data, representing them as "sentences" leverages the power of natural language processing (NLP) models to encode meaningful relationships among genes. The embeddings generated by these models capture co-expression patterns, functional similarities, and other latent biological structures, making this approach both biologically relevant and computationally efficient. Therefore, in OmiCLIP, we chose this approach to bridge transcriptomics and histopathology data by creating a unified embedding space, which allows the model to learn relationships between molecular and visual modalities. Furthermore, as supported by recent studies, this approach not only preserves critical biological information but also generalizes well across diverse datasets and tasks¹⁻³. These results, along with references to the supporting literature, have been added to the manuscript on lines 697-718 to clarify and substantiate the appropriateness of this approach.

For data preprocessing before ranking and data standardization, we manually curated Visium datasets from publicly available databases (**Revised Supplementary Table 1**) and removed duplicate samples. We excluded ST datasets that included only low-resolution H&E images (<2,000 × 2,000 pixels). For datasets with raw count matrices, we normalized them into gene expression profiles following standard protocols using the Seurat⁴ and Scanpy⁵ packages. For datasets already provided in transcripts per million (TPM) or fragments per kilobase of transcript per million fragments mapped (FPKM) formats, which cannot be normalized to standard gene expression profiles, we retained them unchanged. To filter out contaminated droplets, extremely low-quality cells, and damaged cells, we applied quality control criteria, retaining only cells with over 200 expressed genes (nonzero values >200), processed using Seurat and Scanpy. Additionally, for datasets using Ensembl gene IDs, we converted these IDs to gene symbols to

maintain consistency across all datasets. Housekeeping genes were further removed to ensure a more biologically relevant analysis. We added this pipeline on lines 736-749.

For the concern about the effect of varying sequencing depths, we trained OmiCLIP using a comprehensive dataset including a wide range of sequencing depths, which ensures OmiCLIP is robust and reliable across different sequencing depth variations. To further evaluate the robustness, we performed a detailed evaluation of Loki's robustness. First, we analyzed the sequencing depth ranges in ST-bank and categorized samples into high, medium, and low sequencing depth groups, identified as 11792 UMI counts, 4512 UMI counts, and 615 UMI counts, respectively. Next, we generated a simulation dataset with low sequencing depth ST by applying the downsampling function implemented in scuttle⁶. Using this dataset, we compared similarity scores between paired images and original transcriptomic embeddings, with paired images and downsampled transcriptomic embeddings. We evaluated transitions from high to medium sequencing depth, medium to low sequencing depth, and high to low sequencing depth. These embeddings were encoded using OmiCLIP's image and transcriptomic encoders. PLIP and OpenAI CLIP were used as benchmarks for this analysis. The results demonstrated that OmiCLIP is robust to variations in sequencing depth, highlighting its adaptability to datasets generated by different technologies. This additional analysis has been incorporated into the manuscript on lines 121-133.

Response Figure 3 (Revised Extended Data Fig. 3c): Cosine similarity of paired image with transcriptomic embeddings using OmiCLIP (original transcriptomes and down sampled transcriptome from high sequencing depth to middle sequencing depth, middle sequencing depth to low sequencing depth, and high sequencing depth to low sequencing depth, respectively), PLIP (original transcriptome), and OpenAI CLIP (original transcriptome). In the box plots, the middle line represents the median, the box boundaries indicate the interquartile range, and the whiskers extend to data points within 1.5× the interquartile range.

Question 3. For clustering, the authors claim that both embeddings clustered similar tissue types. However, the conclusion is difficult to verify from the UMAPs (Extended Fig. 1), as the results appear unstable, with some seemingly random results. Are the color labels shared between both the image and transcriptomics embeddings? Moreover, legends should be provided for each figure to improve clarity.

Response: Thank you for this valuable comment. We apologize for the confusion in this analysis. We agree that the UMAPs used in Extended Data Fig. 1 were primarily for visualization and may not serve as a quantitative evaluation of the clustering performance. Therefore, we have calculated the Calinski-Harabasz (CH) score (**Revised Extended Data Fig. 1 and 2**; Extended Data Fig. 2 in Manuscript Version 1), a widely used clustering validation metric that measures how well data points have been partitioned into distinct clusters⁷. The CH score compares the separation between clusters with the cohesion within clusters, balancing the distance between points in the same cluster against the separation between clusters. Higher CH scores indicate clusters that are more distinct and internally consistent, reflecting better clustering performance.

In the revised manuscript, we first evaluated the CH scores using 95 tissue samples from the ST-bank dataset with expert-annotated cell types, including samples from breast, healthy heart, kidney cancer, lung, and myocardial infarction (MI) heart tissues (**Revised Supplementary Table 2**). The annotated cell types were used as ground truth cluster labels. Our results showed a significant increase (p value <0.001 , **Response Figure 4**; **Revised Extended Data Fig. 1**) in the CH scores for embeddings after contrastive learning compared to before, demonstrating improved clustering performance.

Additionally, we expanded the CH score calculations to the rest of the ST-bank samples, where no cell type annotations are directly available. For these samples, we used the Leiden algorithm to cluster spatial transcriptomics data in the UMAP representation and treated these Leiden clusters as the ground truth cluster labels. The results from both the annotated cell types and the Leiden-generated clusters were consistent, showing a significant improvement (p value <0.001 , **Revised Extended Data Fig. 2**) in CH scores after contrastive learning.

We have updated the manuscript to ensure that legends are provided for all figures to improve clarity, and the color labels have been standardized to represent shared information between both image and transcriptomics embeddings. These updates and the CH score result have been included in the revised manuscript on lines 165-182, 1028-1041 and **Revised Extended Data Fig. 1 and 2**.

Response Figure 4 (Revised Extended Data Fig. 1): a, Clustering performance on ST-bank data with cell type annotations. Left: clustering performance using transcriptomic embeddings generated from OmiCLIP model before and after training. Right: clustering performance using image embeddings from OmiCLIP model before and after training. The Calinski-Harabasz scores were calculated on the embeddings (Methods) using the pretrained OmiCLIP transcriptomic (left) and image (right) encoders, evaluated for each organ type. Higher Calinski-Harabasz scores indicate better separation capability between clusters of the embeddings. In the box plots, the middle line represents the median, the box boundaries indicate the interquartile range, and the whiskers extend to data points within 1.5× the interquartile range. **b**, Example samples of lung, kidney cancer, healthy heart, and MI heart, colored by cell types, respectively.

Question 4.1 For Loki Align, the authors select fine-tuning and zero-shot settings for different real-world datasets. If this is case, how should the users choose the appropriate settings when applying the model to their datasets?

Response: Thank you for this valuable comment. For the Loki Align, the ST-to-ST alignment doesn't require fine-tuning. For the image-to-ST alignment, we utilized the fine-tuning setting in all datasets, including the simulation dataset and two experimental datasets (small intestine and ovarian carcinosarcoma). The purpose of adding a zero-shot setting in the ovarian carcinosarcoma dataset is to evaluate the zero-shot capability of OmiCLIP embeddings. Our analysis indicates that while zero-shot alignment slightly reduces accuracy, the fine-tuning still achieved a better performance. Moreover, in response to Reviewer 3's Question 2, we added an evaluation between pre-training plus fine-tuning, pure pre-training, and pure training from scratch. The results indicated that pre-training plus fine-tuning has the best performance (Details in Manuscript on lines 268-276 and 491-501). We recommend fine-tuning as a default setting for the Loki Align, as it ensures compatibility with datasets that may not be represented in the ST-bank. To further clarify these points, we also included a table in the manuscript (**Response Table 1; Revised Extended Data Fig. 17**) summarizing the settings of Loki.

	Function	Software	Fine-tune
Loki Align	H&E image to ST alignment	Loki only	Suggested
Loki Align	ST to ST alignment	Loki PASTE ...	No need
Loki Decompose	scRNA-seq to H&E image	Loki only	Suggested
Loki Decompose	scRNA-seq to ST mapping	Loki Tangram ...	No need
Loki Annotate	Tissue annotation by Bulk RNA-seq	Loki only	No need
Loki Annotate	Tissue annotation by marker genes	Loki only	No need
Loki Retrieve	H&E image-to-ST retrieval	Loki only	No need

Response Table 1 (Revised Extended Data Fig. 17): Summary of the fine-tuning settings for downstream tasks.

Question 4.2 It is clear how to measure alignment accuracy in ST-to-ST scenarios, but how do you measure accuracy in image-to-ST scenarios?

Response: Thank you for this question. For image-to-ST scenarios, if our alignment is highly accurate, the paired ST corresponding to the H&E image should be close to the target ST dataset. Therefore, after aligning the H&E image to the target ST dataset, we evaluated alignment accuracy by comparing the paired ST data corresponding to the H&E image with the target ST dataset. This approach allowed us to apply the same accuracy metrics as those used in the ST-to-ST scenarios.

In response to the suggestion to use additional evaluation metrics and concerns about the appropriateness of mean square error (MSE) (raised by Reviewer 2), we replaced MSE with the

Pearson correlation coefficient (PCC) and Kendall's tau coefficient for a more robust evaluation. These updates and clarifications have been incorporated into the manuscript on lines 224-229.

Question 5. Fine-tuning is a critical step for dataset- or task-specific analysis, yet the fine-tuning details for each task are only briefly mentioned. More detailed steps should be provided, including the loss function and fine-tuning framework.

Response: Thank you very much for this important comment. We have added a detailed workflow in the manuscript on lines 691-695 and 854-868 (Results and Methods) to guide users in performing fine-tuning on their own datasets. This includes comprehensive information on the loss function and fine-tuning framework. Below is the description of the fine-tuning method.

“To achieve better performance on downstream tasks, OmiCLIP allows users to fine-tune the pretrained model using their own datasets. To create the fine-tuning dataset, we first preprocessed the Visium data by the standard 10x Space Ranger pipeline. Then, we generated the gene name sentence using the same workflow described in the ‘Training dataset curation’ section. This preprocessed workflow could ensure the fine-tuning dataset has the same format as pretraining dataset. To fine-tune OmiCLIP, we used the contrastive loss between the image embeddings and the paired text embeddings of the top expressed gene names sentences. The contrastive loss is calculated by

$$L_{con} = -\frac{1}{N} \left(\sum_i^N \log \frac{\exp\left(\frac{x_i^T y_i}{\sigma}\right)}{\sum_{j=1}^N \exp\left(\frac{x_i^T y_j}{\sigma}\right)} + \sum_i^N \log \frac{\exp\left(\frac{y_i^T x_i}{\sigma}\right)}{\sum_{j=1}^N \exp\left(\frac{y_i^T x_j}{\sigma}\right)} \right),$$

where x_i represents the normalized embeddings of the image in the i^{th} pair, and y_j denotes the normalized embeddings of the text in the j^{th} pair. N denotes the batch size, while σ represents the temperature parameter used to scale the logits. The pretrained model was fine-tuned for 10 epochs for tissue alignment task and 5 epochs for cell type decomposition task, using a local batch size of 64 by minimizing the contrastive loss on a given batch.”

Additionally, we have provided notebooks in “source_code/website” folder (please see “source_code /README.md” file with instructions for view it) in Additional Review Material, to facilitate the fine-tuning process.

Question 6. For Loki Decompose, the authors compare it with baseline methods such as Tangram (Biancalani et al. 2021), Spaital Seurat (Stuart et al., 2022), CARD (Lu et al., 2021), and CytoSPACE (Vahid et al., 2023). However, it would be beneficial to also include Cell2location (Kleshchevnikov et al., 2022), SpatialDWLS (Dong et al., 2021), and RCTD (Jin et al., 2024), which have been shown to be the top-performing methods for cell type deconvolution in the benchmarking paper by Li et al. 2022.

Response: Thank you for this valuable suggestion. We have now included a comparison of Loki Decompose with Cell2location, SpatialDWLS, and RCTD, in addition to the previously compared methods (**Response Figure 2; Revised Fig 5c**). In total, Loki Decompose has been benchmarked against seven state-of-the-art (SOTA) models for this task. Using the same workflow as described in response to Question 1, we evaluated their performance. Our results

demonstrate that Loki Decompose achieves the highest performance among all the methods. This analysis has been incorporated into the manuscript on lines 434-451.

Question 7. The results in Loki Decompose are not particularly solid. In Fig. 3f, the authors highlight that Loki Decompose is comparable to Tangram results. However, the results obviously show that Loki Decompose underperforms Tangram in terms of both SSIM and JS. Additionally, in Fig.3g, some color bars are missing.

Response: Thank you for this valuable comment. We acknowledge that, while the p -value from Wilcoxon test did not indicate a significant difference between Loki Decompose and Tangram, Loki Decompose underperforms Tangram in terms of both SSIM and JS metrics. To address this, we developed a refinement inspired by the Non-Maximum Suppression (NMS)⁸ technique to enhance decomposition performance, particularly in complex spatial scenarios. The rationale behind this approach stems from the need to handle overlapping and ambiguous cell type assignments in spatial probabilistic maps. These maps quantify the likelihood of a single cell being located within a specific spot, but in regions with high cellular heterogeneity, multiple cell types may have comparable probabilities, leading to noise and reduced decomposition precision. By adapting the concept of NMS, we refined the spatial probabilistic maps to retain only the cell type with the highest likelihood at each spot, suppressing other probabilities. This process effectively reduces ambiguity and focuses the model's predictions on the most confident assignments. After applying this refinement, we observed notable improvements in Loki Decompose's SSIM and JS scores (**Response Figure 5; Revised Fig. 5f**). This refinement demonstrates the utility of leveraging probabilistic filtering to improve spatial decomposition, especially in datasets with high variability or noise. These updated results and their implications have been added to the manuscript on lines 459-462.

Additionally, we have corrected Fig. 3g to include the missing color bars, improving clarity and completeness. These updates provide a clearer and more compelling comparison between Loki Decompose and Tangram.

Response Figure 5 (Revised Fig. 5f): Barplot shows the accuracy of decomposition by Loki using ST or image, and by Tangram using ST. Error bar is the standard deviation. For both JS divergence and SSIM, adjusted p -value > 0.1 using a two-sided Wilcoxon test.

Question 8. A single metric is insufficient to validate the results. The authors use MSE to evaluate the results for Loki Align and Loki Decompose, but relying on a single metric makes it difficult to ensure fairness. For a fair comparison, more metrics should be included.

Response: Thank you for this valuable comment. To address your concern, we incorporated additional metrics for both the alignment and decomposition tasks to ensure a fair and comprehensive evaluation.

For the alignment task, we replaced MSE with the Pearson correlation coefficient (PCC) and Kendall's tau coefficient, which are more appropriate metrics for evaluating alignment accuracy as suggested by Reviewer #2. The revised analysis consistently showed that Loki outperforms other tools based on both PCC and Kendall's tau coefficient. Detailed results have been added to the manuscript on lines 224-251.

For the decomposition task, we used JS divergence and SSIM scores as complementary metrics to evaluate performance. The updated results demonstrate that Loki outperforms other state-of-the-art (SOTA) models based on these two metrics. These analyses and their results are described in detail in the manuscript on lines 434-451 and illustrated in **Revised Fig. 2c, d, 3c, f, and Revised Extended Data Fig. 4a**.

Response Figure 6 (Revised Fig. 2c, d, and Revised Extended Data Fig. 4a): a, Tissue alignment results of 8 adjacent normal human small intestine samples, represented by the PCC and Kendall's tau coefficient of the highly expressed gene expression between target sample and source sample after alignment at the same location, using Loki (ST-to-ST and Image-to-ST) and baseline methods PASTE

(ST-to-ST), GPSA (ST-to-ST), and CPD method (ST-to-ST), respectively. In the box plots, the middle line represents the median, the box boundaries indicate the interquartile range, and the whiskers extend to data points within 1.5× the interquartile range. **b**, Comparison of tissue alignment performances, represented by the PCC and Kendall's tau coefficient of the highly expressed gene expression between target sample and source sample after alignment at the same location, using Loki (ST-to-ST and Image-to-ST) and baseline methods PASTE (ST-to-ST), GPSA (ST-to-ST), and CAST (ST-to-ST), respectively. In the box plots, the middle line represents the median, the box boundaries indicate the interquartile range, and the whiskers extend to data points within 1.5× the interquartile range.

Question 9. OmiCLIP is demonstrated to have promising results in tissue annotation and cell type decomposition. However, the cell types in the tissue samples used in the experiments seem somewhat coarse. For example, the human colorectal cancer samples are annotated with only 6 cell types. This might not be particularly challenging for the model, as the features distributions between cell types may be distinct enough. To further enhance the model's reliability, testing the model on tissue samples with more fine-grained cell types could be beneficial.

Response: Thank you for this insightful comment. As suggested, we conducted an additional experiment to evaluate OmiCLIP on a more challenging task involving tissue samples with finer-grained cell type annotations. Specifically, we tested OmiCLIP on a mouse brain cortex sample containing vascular and leptomeningeal cells (VLMCs), astrocytes, neurons from layers 2/3 (L2/3), layers 4/5 (L4/5), and layer 6 (L6), as well as oligodendrocytes. VLMCs and astrocytes are concentrated near the cortical surface and pial borders (e.g., layer 1), while oligodendrocytes are more prevalent in deeper layers and within white matter tracts⁹. This dataset presents a significantly more complex scenario, as neurons from different cortical layers share similar morphology and molecular features, making it a stringent test for the model.

To decompose the mouse brain cortex slice, we first fine-tuned the OmiCLIP model using adjacent Visium data and H&E images. The whole slide image (WSI) was divided into image patches corresponding to the size of Visium spots. We used the fine-tuned OmiCLIP text encoder to encode the scRNA-seq data and the image encoder to encode the H&E image. Loki Decompose was then applied to predict the cell type distributions from the H&E image.

The results demonstrated that Loki Decompose accurately predicted the distributions of VLMCs, astrocytes, L2/3, L4/5, L6 neurons, and oligodendrocytes, aligning closely with brain anatomical references. These findings further validate the robustness and reliability of OmiCLIP for tasks involving more nuanced and fine-grained cell-type decomposition. We have added this analysis in the manuscript on lines 475-489.

Response Figure 7 (Revised Fig. 5h): Cell type decomposition results on the brain sample. Left: brain anatomical references with zoom-in H&E image patches of L1 (VLMCs, astrocytes), L2/3, L4/5, L6, and WM (oligodendrocytes), respectively. Right: heatmap shows the cell type distribution of VLMCs, astrocytes, L2/3, L4/5, L6, and oligodendrocytes, with color reflecting the distribution of each cell type.

Question 10. The full codes, platform, and the database ST-bank are currently unavailable. The authors should make these resources accessible so that our referees can access them before publication.

Response: Thank you for this comment. In the re-submission, we provided a user-friendly package, a Readme file, pre-train weights (<https://drive.google.com/file/d/1UTJKAqoiUT55h9sA1Ee8iJeOciNLBvWD/view?usp=sharing>) and ST-bank database (<https://drive.google.com/drive/folders/1J15cO-pXTwkTjRAR-v-nQkqXNfcCNn3?usp=sharing>). We uploaded a website to introduce Loki platform including functions and notebooks in the “source_code” file the Additional Review Material. We plan to release the full codes, platform, pre-trained model, website, and database ST-bank to the public upon acceptance of the manuscript, ensuring accessibility for the broader research community.

Minor comments:

Question 1. Some references are missing, such as nonconvex optimization algorithms.

Response: Thank you for this comment. As suggested, we have included the reference of nonconvex optimization algorithms.

Question 2. Some results cannot be found. For example, “Additionally, our predicted tumor cell density was consistent with the predictions by CLAM, a SOTA model specifically designed for tumor analysis in WSIs.” Where are the results?

Response: Thank you for this comment. We have included the results of CLAM in manuscript version 1 Fig. 3g which is **Revised Fig. 5g**.

Reference

1. Theodoris, Christina V., et al. "Transfer learning enables predictions in network biology." *Nature* 618.7965 (2023): 616-624.
2. Cui, Haotian, et al. "scGPT: toward building a foundation model for single-cell multi-omics using generative AI." *Nature Methods* (2024): 1-11.
3. Hao, Minsheng, et al. "Large-scale foundation model on single-cell transcriptomics."

Nature Methods (2024): 1-11.

4. Biancalani, Tommaso, et al. "Deep learning and alignment of spatially resolved single-cell transcriptomes with Tangram." *Nature methods* 18.11 (2021): 1352-1362.
5. Stuart, T., et al. *Comprehensive integration of single-cell data.* *cell* 177, 1888-1902. e1821 (2019).
6. Ma, Y. & Zhou, X. *Spatially informed cell-type deconvolution for spatial transcriptomics.* *Nature biotechnology* 40, 1349-1359 (2022).
7. Vahid, Milad R., et al. "High-resolution alignment of single-cell and spatial transcriptomes with CytoSPACE." *Nature biotechnology* 41.11 (2023): 1543-1548.
8. Kleshchevnikov, Vitalii, et al. "Cell2location maps fine-grained cell types in spatial transcriptomics." *Nature biotechnology* 40.5 (2022): 661-671.
9. Dong, Rui, and Guo-Cheng Yuan. "SpatialDWLS: accurate deconvolution of spatial transcriptomic data." *Genome biology* 22.1 (2021): 145.
10. Jin, Xing, et al. "RCTD: Reputation-Constrained Truth Discovery in Sybil Attack Crowdsourcing Environment." *Proceedings of the 30th ACM SIGKDD Conference on Knowledge Discovery and Data Mining.* 2024.
11. Li, Bin, et al. "Benchmarking spatial and single-cell transcriptomics integration methods for transcript distribution prediction and cell type deconvolution." *Nature methods* 19.6 (2022): 662-670.

Repones: Thanks for this reference list. We have cited them in the revised manuscript.

Remarks on code availability:

The authors only provided part of the codes and cannot find the pretrained model. Therefore, I cannot reproduce the results according to the codes.

Repones: We appreciate the reviewer's comment regarding code availability. We acknowledge the importance of reproducibility and transparency. We plan to release the full codes, platform, pre-trained model, and database ST-bank to the public upon acceptance of the manuscript. This will ensure full accessibility for the broader research community and facilitate the reproduction of the results.

Response to Reviewer #2:

Remarks to the Author:

The study introduces OmiCLIP, a dual-encoder foundation model linking H&E images with transcriptomic data from Visium samples. By transforming the top-expressed genes within tissue patches into "sentences," this method adapts CLIP-style contrastive learning to bring histology and transcriptomic data into a shared embedding space. This shared space enables five downstream applications within their Loki platform:

- 1. Loki Align - Aligns spatial transcriptomics (ST) and H&E images through an adapted coherent point drift method.*
- 2. Loki Decompose - Uses OmiCLIP embeddings with Tangram for cell type decomposition.*
- 3. Loki Annotate - Embeds bulk RNA-seq data, allowing similarity comparisons with regions in H&E images or disease-specific gene sets.*
- 4. Loki Retrieve - Accepts an H&E image to identify similar ST panels from the training or finetuning data.*
- 5. Loki PredEx - Predicts ST gene expression from H&E by averaging gene expression across the ST slides predicted to be most similar.*

Additionally, they introduce ST-bank, a dataset of 2.2 million tissue patches from over 1,000 Visium samples, to support multi-modal AI integration of histology and omics data.

Overall, I have major concerns about the methodology and the data analysis. Many of these issues are due to inadequate descriptions of key details of the method and analysis. However, even when things were clear, I was confused about several of the interpretations of the results. While the method is very intriguing and novel, the writing and results need major improvements

Response: Thank you very much for reviewing our work. We are thrilled that you find our method intriguing and novel. We apologize for the inadequate descriptions of key details regarding the method and analysis in the original submission. In this revision, we have thoroughly revised the manuscript to address these concerns, adding key details to clarify both the methodology and data analysis. Below, we provide a point-by-point response to your comments.

Major Concerns About Methodology/Approach:

Question 1. Evaluation of UMAP Embeddings and Contrastive Learning:

The reference to UMAP embeddings in Extended Data Fig. 1 is misleading. The claim that similar tissue types are clustered across modalities doesn't align with the actual figure, which shows UMAP embeddings before and after contrastive learning, organized by tissue type. Furthermore, it appears that contrastive learning negatively impacts the transcriptomics embeddings, casting doubt on the efficacy of the approach. Also unclear is the source of the spatial transcriptomics (ST) labels—these are reportedly derived from Leiden clustering on UMAP embeddings but lack clear description in the Method section. It seems like their initial embeddings of ST data is fairly well clustered, except that the annotations are incorrect.

Response: Thank you very much for these thoughtful comments. In response, we substantially revised this section to improve the analysis. We carefully revised Extended Data Fig. 1 to ensure clarity and provide proper context.

We apologize for any confusion caused by the earlier version of the figure. Since OmiCLIP includes both transcriptomics and image encoders, in this section, we evaluated whether contrastive learning enhances the ability of each encoder to represent tissue types better than the initial encoders, rather than the ability to align tissues cross modality. We have added an explanation on lines 162-164 and revised the title of this section. To do so, we first visualized the UMAP embeddings generated from both the transcriptomics and image encoders, before and after contrastive learning, and then quantitatively evaluated the clustering performance. Further, we calculated the Calinski-Harabasz (CH) score, a metric that compares the separation between clusters with their internal cohesion, to improve the quantitative evaluation. In the revised manuscript, we first added a new benchmark using 95 tissue samples from the ST-bank dataset with expert-annotated cell types, including breast, healthy heart, kidney cancer, lung, and myocardial infarction (MI) heart tissues (**Revised Supplementary Table 2**). These expert-annotated cell types were used as ground truth cluster labels. The results demonstrated a significant increase in CH scores for embeddings after contrastive learning compared to before, indicating improved clustering performance (p -value <0.001 , **Response Figure 4; Revised Extended Data Figure 1**). Details of the annotations for these datasets are provided in **Revised Supplementary Table 2** and described in the Methods section.

We also extended the CH score analysis to the remaining samples in the ST-bank dataset, where no expert-annotated cell type labels are available. For these samples, we used the Leiden algorithm to cluster ST data based on their UMAP representations and treated these Leiden clusters as ground truth labels. The results for both annotated cell types and Leiden-generated clusters were consistent, showing a significant improvement in CH scores after contrastive learning (p -value <0.001 , **Revised Extended Figure 2**).

Regarding the initial embeddings of ST data, the initial text encoder was not trained from scratch but from the pretraining weights on LAION-5B¹⁰ including biological literatures, which may explain its tendency to cluster similar tissue patches together. However, the clustering performance was significantly improved after fine-tuning on ST-bank, as reflected in the higher CH scores post-contrastive learning. These updates have been incorporated into the revised manuscript on lines 165-182, 845-847, 1028-1041 and Figure/Table references.

Clarifications on Loki Decompose:

Question 2.1: The description of Loki Decompose needs refinement to highlight that it essentially employs Tangram's nonconvex optimization approach with OmiCLIP embeddings rather than raw gene expression data.

Response: Thanks for this comment. We have highlighted that in Loki Decompose, it is essential to employ the nonconvex optimization¹¹ introduced by Tangram¹² with OmiCLIP embeddings rather than raw gene expression data (lines 418-423).

Question 2.2: Further, critical details in the methods section are challenging to follow: the role of PCA, computation of expected cell counts per spot (and if these are also given

to base Tangram), and the concept of adapting mappings at the cell cluster level all require elaboration.

Response: Thank you for raising this point. In our original approach, PCA was used to reduce the dimensions of the image and text embeddings. However, to ensure a fairer comparison with Tangram, which didn't use PCA, we removed the PCA step in the revised analysis and updated the results. Additionally, we did not compute the expected cell counts per spot in either Loki or Tangram, as read count information was not available in our dataset. We have described these in the Methods section.

Question 2.3: Moreover, the conflicting statements about the cosine similarity loss function, first described as minimizing and then as maximizing, add unnecessary ambiguity. Consistency in the Tangram-based comparisons, using identical adjustments for both methods, would improve the validity of the assessment.

Response: Thank you very much for pointing it out! We replaced the cosine similarity with cosine distance and kept the description consistent stating minimizing cosine distance. We have used the same settings for Tangram and Loki to ensure we fairly compared Tangram and Loki. We have modified the description of the methods (lines 958-973).

Major Concerns About Data Analysis:

*Question 3. Baseline and Metric Selection for Loki Align:
For Loki Align, a fairer baseline would involve applying the Coherent Point Drift method to PCA embeddings of the transcriptomic data, as it remains unclear whether the alignment accuracy results from the OmiCLIP embeddings themselves or simply from a superior registration method compared to PASTE or GPSA. Additionally, using Pearson correlation, a standard metric for transcriptomic comparison, may be more appropriate than MSE, especially when evaluating transcriptomic alignment accuracy.*

Response: Thank you for this important comment. We acknowledge that the observed alignment accuracy could result from the use of OmiCLIP embeddings or from the superior registration method (Coherent Point Drift, CPD) compared to PASTE or GPSA. To address this, we applied the CPD method to both OmiCLIP embeddings and PCA embeddings of the transcriptomic data and evaluated their performance on eight adjacent normal human small intestine samples. As suggested, we used Pearson correlation coefficient (PCC) and Kendall's tau coefficient, widely accepted metrics for tissue alignment accuracy. The results show that OmiCLIP embeddings significantly improved the performance of alignment comparing to PCA embeddings (p -value <0.001 , Wilcoxon test, **Response Figure 6a**). We have added this analysis in manuscript on lines 238-244.

Interpretability of Figure 2:

Question 4. Figure 2 contains notable visual ambiguities. The right-side plots in Fig. 2d are improperly framed, obscuring interpretation. Additionally, the alignment for Source6 in Fig. 2c appears mirrored on the X-axis, an effect that PASTE's method typically doesn't perform. If intentional, this mirroring needs clarification to explain its relevance to the alignment method.

Response: Thank you very much for this comment. To improve the clarity of Figure 2, we replaced the frame with coordinate axes in Figure 2d for better interpretation. Additionally, we carefully reviewed the alignment of PASTE for Source 6 in Figure 2c and confirmed that PASTE aligns this slide with a mirrored X-axis, although PASTE showed a robust performance on source slices 4, 5, and 7. To further clarify, we have included a spot-check figure demonstrating this alignment process. Specifically, we used PASTE to align Source6 to the target slice using the default parameters of PASTE. Results show that Source 6 was mirrored on the X-axis (**Response Figure. 8**).

Response Figure 8. Left panel shows the target and source samples before PASTE alignment, which are consistent with the input data of other compared tools. Right panel shows the target and source samples after PASTE alignment. The target and source samples are colored in red and blue, respectively.

Metric Choice and Consistency in Loki Decompose:

Question 5. The use of MSE for comparing cell type proportions in Fig. 3c, split by cell type, is unconventional and confusing, particularly for comparing spatial spot vectors. A unified approach would be beneficial across evaluations, as the use of JS divergence and SSIM in Fig. 3e is inconsistent with the preceding analyses in 3c-d, despite targeting the same evaluation objective.

Response: Thanks for this valuable comment. To ensure consistency across evaluations, we have replaced MSE with JS divergence and SSIM in **Response Figure 2** and **revised Fig. 5c**, aligning it with the metrics used in **revised Fig. 5f**. Additionally, as suggested by Reviewer 1, we added more benchmarking analyses to provide a comprehensive evaluation. These updates have been detailed in the manuscript on lines 434-451.

Evaluation Strategy for Loki Retrieve:

Question 6. The evaluation approach for Loki Retrieve is problematic. Rather than comparing retrieved images with unpaired image data, it would be far more informative to use images that have paired transcriptomic data. This would offer a direct assessment of the model's ability to retrieve accurate transcriptomic information from images. As it stands, the title and evaluation give a misleading impression about the model's performance on this task.

Response: Thank you for highlighting this important issue. In response, we incorporated an additional evaluation approach for Loki Retrieve by calculating the rank of the correct pair using

Recall@K. This metric measures the proportion of correctly retrieved data within the top-K quantile retrieved samples (Methods). We have reserved 4 samples from ST-bank as validation datasets including brain, heart, kidney, and breast tissue samples. We compiled four independent ST studies as a test dataset, including desmoplastic small round cell tumor, colorectal cancer, vascular, and colon samples (**Revised Supplementary Table 4**). Results demonstrated that Loki significantly outperformed both OpenAI CLIP and PLIP across all validation datasets. Specifically, Loki achieved Recall@5% of 0.125 and Recall@10% of 0.227 for brain (averagely 2.3-fold higher than OpenAI CLIP and 2.5-fold higher than PLIP), Recall@5% of 0.186 and Recall@10% of 0.291 for heart (averagely 3.2-fold higher than OpenAI CLIP and 3.1-fold higher than PLIP), Recall@5% of 0.173 and Recall@10% of 0.297 for kidney (averagely 3.2-fold higher than OpenAI CLIP and PLIP), and Recall@5% of 0.140 and Recall@10% of 0.240 for breast (averagely 2.6-fold higher than OpenAI CLIP and PLIP) (**Response Figure 9; Revised Fig. 6e**). On the test dataset, Loki further demonstrated substantial improvements, achieving Recall@5% of 0.117 and Recall@10% of 0.208 (averagely 3.1-fold higher than OpenAI CLIP and 3.0-fold higher than PLIP) (**Response Figure 9; Revised Fig. 6e and Revised Supplementary Table 4**).

These results confirm Loki's superior performance in accurately retrieving paired transcriptomic information from images, addressing the limitations of the original evaluation approach.

Dataset	Metric	Loki	OpenAI CLIP	PLIP	Fold change (Loki versus OpenAI CLIP)	Fold change (Loki versus PLIP)
Validation brain dataset	Recall@5%	0.125	0.051	0.048	2.4	2.6
	Recall@10%	0.227	0.103	0.095	2.2	2.4
Validation heart dataset	Recall@5%	0.186	0.052	0.057	3.6	3.3
	Recall@10%	0.291	0.104	0.103	2.8	2.8
Validation kidney dataset	Recall@5%	0.173	0.052	0.053	3.4	3.3
	Recall@10%	0.297	0.100	0.097	3.0	3.0
Validation breast dataset	Recall@5%	0.140	0.049	0.050	2.9	2.8
	Recall@10%	0.240	0.100	0.096	2.4	2.5
Test dataset	Recall@5%	0.117	0.033	0.042	3.5	2.8
	Recall@10%	0.208	0.075	0.067	2.8	3.1

Response Figure 9 (Revised Fig. 6e): Image-to-transcriptomics retrieval evaluation across four validation datasets and one test dataset using Loki, OpenAI CLIP, and PLIP, respectively. The top-K quantile most similar transcriptomics were retrieved. We report Recall@K for $K \in \{5\%, 10\%\}$ (Methods).

Question 7. Furthermore, in looking at the python notebook, the retrieved images do not look like they would necessarily have a similar number of cells, or the same composition as the query images. It doesn't seem like the model would be able to accurately represent the transcriptomic profile based on the query image.

Response: Thank you for this comment. The query images in the notebook were collected from the CRC7K pathology database, where the image sizes are larger than tissue patches in the ST-bank dataset. As a result, the retrieved images do not have a similar number of cells as the query images. Notably, since contrastive learning frameworks, such as OmiCLIP, are not generative AI approaches, they are not capable of generating an accurate transcriptomic profile

for the query image. Instead, the retrieved tissues are the most similar image to the query tissue. We also discussed developing a diffusion model on WSI to generate the transcriptomic profile based on the query image as a future direction. We have acknowledged this limitation of OmiCLIP in the Discussion section on lines 678-680.

Question 8. F1 Score in Loki Annotate:

The computation of the F1 score in Fig. 5c is unclear, particularly concerning how cosine similarity data produces an F1 score. Additional explanation of the scoring framework and its connection to cosine similarity would be valuable to substantiate this metric's relevance.

Response: Thank you for pointing out the need for clarification regarding the computation of the F1 score in Fig. 5c. To calculate the F1 score, we first calculated the dot product of text embeddings and H&E image embeddings as the cosine similarity to annotate tissue type. The tissue type with the highest cosine similarity score was assigned as the predicted tissue to that query image. Specifically, in the OmiCLIP model, the text is the sentence of marker genes. Based on these annotations, the F1 score was calculated as the harmonic mean of precision and recall, which was used to measure classification performance. Precision was defined as the proportion of correctly predicted tissues (true positives) out of all predicted tissues, while recall was defined as the proportion of correctly predicted tissues out of all actual tissues. We added this clarification in the manuscript on lines 360-367.

Interpretation of UMAP and Embedding Quality:

Question 9.1 The UMAP-based evaluation in Supp Fig. 2 is confusing, with mismatches between annotations in the UMAP and the confusion matrix, particularly for misannotated cell types. This discrepancy complicates the assessment of OmiCLIP's performance, as the annotations should align for a valid comparison.

Response: Thank you for pointing out the issue in Supplementary Figure 2. We have updated and relabeled the annotations in **Revised Supplementary Figure 2** to ensure clarity and consistency. The revised figures now have consistent annotations between the predicted and ground truth datasets, addressing the discrepancies and providing a more accurate assessment of OmiCLIP's performance.

Response Figure 10 (Revised Supplementary Figure 2): Few-shot cell type annotation. **a**, Confusion matrix between predicted and actual cell types of the kidney cancer dataset. The matrix is colored from light to dark blue by the precision values. The color boxes along the axes represent the cell types. The projection of the actual and predicted cell type labels to the original UMAP embeddings for the breast cancer dataset. 1% randomly sampled cells were used for training and the rest were plotted in the UMAPs. **b**, same as in (a) for the prostate cancer dataset.

Question 9.2 Further investigation into why OmiCLIP generates two distinct clusters for TCGA LUAD, shown in Supp Fig. 3, would also aid interpretation, as this outcome suggests potential biases or unique properties within this embedding.

Response: For the question about Supp Fig. 3, as suggested by the reviewer, in the revision, we investigated OmiCLIP's transcriptomic encoder on the TCGA bulk RNA-seq data. OmiCLIP's transcriptomic embeddings segregated LUAD patients into two groups (**Revised Supplementary Fig. 4a**) as LUAD 1 and LUAD 2. To investigate the clinical insight added by these transcriptomic embeddings, we performed Kaplan-Meier survival analysis on these two groups of patients. The results showed a significant difference in overall survival between the two groups (**Revised Supplementary Fig. 4b**), indicating the prognostic relevance of OmiCLIP's transcriptomic encoder. To further investigate the underlying relevance of the OmiCLIP transcriptomic embeddings, we split the larger LUAD group into LUAD 1 – Prox (50%) and LUAD 1 – Dist (50%), which were close to and distant from the LUAD 2 group in OmiCLIP embedding space, respectively (**Revised Supplementary Fig. 4c**). Kaplan-Meier survival analysis revealed a more significant difference in overall survival between the LUAD 1 – Dist

and LUAD 2 groups, with a p -value of 7×10^{-4} (**Revised Supplementary Fig. 4d**). This analysis indicated OmiCLIP's capability to identify patient clusters with clinical relevance. This capability potentially arises from the incorporation of the biological knowledge derived from the literature: we initialized OmiCLIP's transcriptomic encoder using the pretraining weights of a Bidirectional Encoder Representations from Transformers (BERT) language model, thereby leveraging the power of natural language processing (NLP) models to embed biological information into a high-dimensional space. A similar phenomenon has been reported by recent publications such as GenePT³ and cell2sentence¹³, supporting our approach of utilizing text-based embeddings to capture rich biological information efficiently. We have added this new result to the Supplementary Note and the Discussion section in the manuscript.

Response Figure 11 (Supplementary Figure 4): OmiCLIP transcriptomic embeddings stratify patient groups on TCGA LUAD data. **a**, the PCA projections of the OmiCLIP transcriptomic embeddings of the TCGA LUAD patients (LUAD 1 and LUAD 2). **b**, Kaplan-Meier curves for the LUAD 1 and LUAD 2 patient groups. The log-rank test p -values are provided. **c**, the PCA projections of the OmiCLIP transcriptomic embeddings of the TCGA LUAD patients colored by LUAD 1 – Prox (close to LUAD 2), LUAD 1 – Dist (distant from LUAD 2), and LUAD 2. **d**, Kaplan-Meier curves for the LUAD 1 – Dist and LUAD 2 patient groups.

Potential Overestimation of CLIP Performance:

Question 10. The unexpectedly strong performance of CLIP for single-cell annotation raises questions about the advantages and potential limitations of the gene expression textual representation method. Exploring ways to evaluate these representations independently, such as UMAP visualizations based on CLIP and OmiCLIP embeddings, would clarify the distinct advantages or drawbacks of this approach.

Response: Thank you for this insightful suggestion. As suggested, we have investigated the UMAP visualizations of the CLIP (OpenAI CLIP) and OmiCLIP transcriptomic embeddings (**Revised Supplementary Fig. 1**). Our analysis shows that OmiCLIP effectively clustered cells of the same cell types from multi-sample scRNA-seq data. In contrast, OpenAI CLIP, pre-trained on general-domain text, resulted in fragmented clusters of the same cell type and showed greater overlap among similar cell types. This suggests that OmiCLIP's pretraining on domain-focused data, including gene names and histology image patches, enhances OmiCLIP's capability to group cells of the same type across samples more effectively, reflecting underlying biologically meaningful relationships inferred from transcriptomic and histological information.

Furthermore, the text-based representation of gene expression in OmiCLIP provides unique advantages over models that directly use raw gene expression values. By representing gene expression data as text, OmiCLIP leverages the power of NLP models to embed biological information into a high-dimensional space, facilitating seamless integration of omics data with various biological entities, such as pathways, functional annotations (Nature Methods, 2024 a)¹, and cell types (Nature Methods, 2024 b)². Additionally, representing omics data as text aligns with the architecture of other multimodal foundation models, which provides the opportunity to further incorporate more modalities such as proteomics and metabolomics into the same unified space. Recent studies, such as GenePT³, have demonstrated the effectiveness of using large language models to generate gene embeddings from textual descriptions, achieving comparable or superior performance to models trained on extensive gene expression datasets. This supports our approach of utilizing text-based embeddings to capture rich biological information efficiently.

We have included detailed analyses and discussions in the Discussion section in the revised main text and **Revised Supplementary Note 1**.

Response Figure 12 (Revised Supplementary Figure 1): UMAP visualizations of OmiCLIP/OpenAI CLIP transcriptomic embeddings of scRNA-seq data. **a**, UMAP representation of the OmiCLIP transcriptomic embeddings of kidney cancer and prostate cancer multi-sample scRNA-seq datasets. Each dot represents a cell. **b**, same as in **(a)** with OpenAI CLIP embeddings. Note for the prostate cancer dataset, tumor cells ('invisible') totally overlap with the epithelial cells.

Minor Comments:

Question 1. Visual Clarity in Figures:

Fig. 3d suffers from stretching artifacts and inconsistent color scales across methods, which complicates direct comparisons. Standardizing color bars across methods would facilitate more straightforward interpretation of results.

Response: Thank you for the suggestion. We have standardized color bars across methods.

Question 2. Organization of Results Section:

Loki Annotate, which is arguably the most compelling task within the results, is positioned third despite its prominence in the figures. Reordering this section could better emphasize its significance.

Response: Thank you for highlighting Loki Annotate's significance. We reordered the Loki Annotate to the second position before Loki Decomposition.

*Question 3. Terminology Consistency in Loki Decompose:
Describing Loki Decompose as an application of Tangram with OmiCLIP embeddings
instead of gene expression data would enhance clarity and reflect the underlying
methodology more accurately.*

Response: Thank you for the suggestion. We highlighted Loki Decompose as an application of Tangram with OmiCLIP embeddings instead of gene expression data. (revised manuscript lines 418-423).

Remarks on code availability:

*Question 1. Code base lacks methods to format Visium data:
The code as it is currently given does not have any methods to take in the Visium slide
and properly segment the histology image and convert the gene expression data into the
text-based format. This is a critical piece of processing that could impact the results and
it is absent.*

Response: Thank you for the comment. We have provided a website in the “source_code” file of the Additional Review Material, which includes the codes for preprocessing ST data and whole image in the “src” folder and a notebook “Basic Usage of OmiCLIP Model” in the “website” folder as an example to take in the Visium slide and properly segment the histology image and convert the gene expression data into the text-based format (please see “source_code /README.md” file with instructions for view it).

*Question 2. There also was no README file with instructions for installing (although it
doesn't seem like this is an actual software package yet, since it just uses other libraries
so far). I think it would be much better if they created a python package for Loki, since it
would be much easier to just call methods like loki.retrieve() instead of copying a python
notebook and adapting it to your needs.*

Response: Thank you for the comment. We have provided the README file in the “source_code” file of the Additional Review Material with instructions for installing and creating a Python package for Loki with built-in methods.

*Question 3. Since this is poised as a foundation model with wide applicability (especially
within medical imaging fields), the code should really be more user-friendly.*

Response: Thank you for the comment. We have reorganized the code to enhance the installation and the query of functions to make it more user-friendly.

Response to Reviewer #3:

Remarks to the Author:

The manuscript introduces OmiCLIP, a dual-encoder foundation model designed to integrate histopathology images with transcriptomics data, alongside the Loki platform for tasks such as tissue alignment, cell type decomposition, tissue annotation, and ST gene expression prediction. The authors demonstrate superior performance when compared to several methods across multiple datasets, particularly in integrating visual and omics data. While the modeling ideas are innovative, the authors need to carefully validate the pre-training approach and demonstrate the superiority of their method by comparing it with more recent methods. Additionally, as an infrastructure model, greater transparency around the pre-training data is necessary.

Response: Thank you for your valuable comments and for recognizing the innovation and potential of OmiCLIP and the Loki platform. We appreciate your thoughtful feedback on the pre-training approach, comparisons with recent methods, and transparency of the pre-training data. We have carefully addressed all your concerns through a detailed point-by-point response provided in the sections below.

Question 1. For pre-training, how did the authors handle the batch effect between different data resources? It is quite important to guarantee the model has good generalization ability without fine-tuning.

Response: Thank you for raising this important question. We also recognize that batch effects are a key source of variation that can undermine the generalization ability of any pre-trained model. To address this issue, we adopted rank-based strategies inspired by recent single-cell foundation models such as GeneFormer¹⁴ and scFoundation¹⁵, which successfully eliminate batch effects through rank-based approaches rather than relying directly on raw read counts or normalized gene expression values. Specifically, we sorted genes by their expression ranks to reduce biases resulting from differences in overall expression levels across datasets. In addition, we removed housekeeping genes—those commonly expressed in all cell types at relatively stable levels—to prevent them from disproportionately influencing the learned representations. By focusing on the rank order of variable genes, rather than on potentially inconsistent or platform-dependent expression magnitudes, our model is better able to capture meaningful biological variation while reducing technical noise. Collectively, these steps help ensure that our pre-training procedure is robust to batch effects and promotes strong generalization performance, even in the absence of additional fine-tuning. We have clarified this on lines 95-100.

Question 2 The authors fine-tuned the pre-trained model on many downstream tasks, leaving the meaning of pre-training unclear. First, a table summarizing which downstream task needs further fine-tuning model would help readers. Second, it is important to compare the performance of pure pre-training, pure training from scratch for downstream, and pre-training + fine-tuning to show the benefit of pre-training. Also, It would be better if the author could guide readers on when further fine-tuning is needed.

Response: Thank you for this valuable comment. As suggested, we provided a table (**Response Table 1** shown in response to Reviewer #1's Question 4; **Revised Extended Data Fig. 17**) as a guide to the settings for downstream tasks. Although ST-bank includes 32 organ types, some rare conditions still may be missed in ST-bank. So, we suggested fine-tuning in Alignment and Decomposition tasks to ensure compatibility with datasets that may not be represented in the ST-bank, especially when the user's tissue type is not covered by ST-bank. We added this suggestion in the manuscript on lines 691-695.

To address the reviewer's second suggestion, we compared the performance of pure pre-training, pure training from scratch, and pre-training plus fine-tuning for alignment and decomposition tasks. For the alignment task, we evaluated ovarian carcinosarcoma samples using the three approaches. The results indicated that pre-training plus fine-tuning and pure pre-training achieved similar performance with median PCC of 0.86 and 0.85 and median Kendall's tau coefficient of 0.18 and 0.17, respectively, while training from scratch yielded the lowest performance with median PCC of 0.53 and median Kendall's tau coefficient of 0.06.

For the decomposition task, we conducted the comparison on TNBC patient samples. The analysis showed that pre-training plus fine-tuning had the best performance, achieving a mean SSIM score of 0.30 and a mean JS divergence of 0.40. In contrast, pure pre-training resulted in a mean SSIM score of 0.13 and a mean JS divergence of 0.43, while pure training from scratch performed the worst, with a mean SSIM score of 0.0007 and a mean JS divergence of 0.44. Although pure pre-training achieved a comparable JS divergence score to the pre-training plus fine-tuning method (0.43 vs. 0.40), it showed a significant decline in SSIM (0.13 vs. 0.30). This underscores the importance of fine-tuning, as it significantly improves the SSIM score and overall performance. Therefore, we strongly recommend fine-tuning the model for this task to achieve optimal results.

These results underscore the value of pre-training, particularly when combined with fine-tuning, for optimal performance in downstream tasks. We added these analyses to the manuscript on lines 268-276 and 491-501.

Response Figure 13 (Revised Extended Data Fig. 17, Revised Extended Data Fig. 7 and 13): Summarization of the fine-tune settings for downstream tasks. **a**, Tissue alignment results on 2 adjacent human ovarian carcinosarcoma samples using Loki Align Image-to-ST (fine-tuning, pre-training, and train from scratch). We colored the samples using the top three PCA components of OmiCLIP transcriptomic embeddings, mapped to red, green, blue color channels, respectively. **b**, Comparison of tissue alignment performances, represented by the PCC and Kendall's tau coefficient of the highly expressed gene expression between target sample and source sample after alignment at the same location, using Loki Align Image-to-ST (fine-tuning, pre-training, and train from scratch). Bar plot shows the accuracy of alignment by Loki Align Image-to-ST (fine-tuning, pre-training, and train from scratch). Error bar is standard deviation. **c**, Cell type decomposition results on 3 major cell types of the TNBC sample using Loki Decompose Image-to-ST (fine-tuning, pre-training, and train from scratch). The color of the heatmap reflects the z-score, calculated by the enrichment of each cell type. **d**, Bar plot shows the accuracy of decomposition by Loki Decompose Image-to-ST (fine-tuning, pre-training, and train from scratch). Error bar is standard deviation.

Question 3. The authors use the Calinski-Harabasz (CH) score for evaluation, but it is unclear how the ground truth labels were determined for this metric. A more detailed explanation in the methods section is needed.

Response: Thank you for this valuable comment. We apologize for the lack of clarity regarding the CH score calculation. In the revised manuscript, we provided a more detailed explanation of how the ground truth labels were determined for this metric. To calculate the CH score, we used two types of ground truth labels. First, we added a new benchmark dataset including 95 tissue samples in ST-bank, which have expert-annotated cell types, and used these annotations as ground truth labels. These samples included tissues from breast, healthy heart, kidney cancer, lung, and myocardial infarction heart tissues (**Revised Supplementary Table 2**). Second, for the samples without direct cell type annotations, we applied the Leiden algorithm to cluster spatial transcriptomics data (detail parameters in the Method section). These Leiden clusters

were then treated as the ground truth labels for the CH score calculation. We have added these details to the Methods section on lines 1028-1042 to ensure clarity and transparency.

Question 4.1. Spatial transcriptomics data typically include both image and expression information. It would be valuable to explore whether Loki Align could leverage both modalities simultaneously for alignment, rather than relying on just one.

Response: Thank you for this comment. As suggested, we have examined whether Loki Align could leverage both modalities simultaneously for alignment, rather than relying on just one. To evaluate this, we integrated image embeddings and transcriptomic embeddings by averaging them and used the combined embeddings to align two adjacent tissues in ovarian carcinosarcoma samples. We then calculated the Pearson correlation coefficient (PCC) and Kendall's tau coefficient for the image embeddings, transcriptomic embeddings, and averaged embeddings to assess performance. The results indicated that the averaged embeddings do not outperform single-modality embeddings. This analysis has been added to the manuscript on lines 278-284.

Response Figure 14 (Revised Extended Data Fig. 8): **a**, Comparison of tissue alignment performances, represented by the PCC of the highly expressed gene expression between target sample and source sample after alignment at the same location, using Loki ST&image-to-ST, ST-to-ST, and Image-to-ST, respectively. In the box plots, the middle line represents the median, the box boundaries indicate the interquartile range, and the whiskers extend to data points within 1.5× the interquartile range. **b**, Comparison of tissue alignment performances, represented by the Kendall's tau coefficient of the highly expressed gene expression between target sample and source sample after alignment at the same location, using Loki ST&image-to-ST, ST-to-ST, and Image-to-ST, respectively. In the box plots, the middle line represents the median, the box boundaries indicate the interquartile range, and the whiskers extend to data points within 1.5× the interquartile range.

Question 4.2. The inclusion of more recent methods such as STalign and CAST for comparison is also recommended.

Response: Thank you for this valuable suggestion! As recommended, we included CAST in the comparison in ovarian carcinosarcoma samples. The results show that CAST achieved a median PCC score of 0.71 and a median Kendall's tau coefficient of 0.09, ranking as the third-best method overall, while Loki's ST-to-ST and image-to-ST performed as the top two methods. We have added this analysis in the manuscript on lines 247-251.

We also tried to include STalign in the comparison. However, we found that STalign was originally designed to align single-cell spatial transcriptomic data, such as MERFISH and 10x Xenium data. While it does have a mode for aligning MERFISH data with Visium data, it requires manually placed landmarks to enhance alignment accuracy. This reliance on manual intervention would make the comparison unfair when evaluated against other tools. Therefore, we decided not to include STalign in our analysis.

Response Figure 15 (Revised Fig. 2d-e): **a**, Tissue alignment results on 2 adjacent human ovarian carcinosarcoma samples using Loki (ST-to-ST and Image-to-ST) and baseline methods PASTE (ST-to-ST), GPSA (ST-to-ST), and CAST (ST-to-ST), respectively. We colored the samples as described in (Fig. 2b). **b**, Comparison of tissue alignment performances, represented by the PCC and Kendall's tau coefficient of the highly expressed gene expression between target sample and source sample after alignment at the same location, using Loki (ST-to-ST and Image-to-ST) and baseline methods PASTE (ST-to-ST), GPSA (ST-to-ST), and CAST (ST-to-ST), respectively. In the box plots, the middle line represents the median, the box boundaries indicate the interquartile range, and the whiskers extend to data points within 1.5× the interquartile range.

Question 5. Since Loki Align can be performed based on pure image, it would be better if the authors could demonstrate its performance on aligning 10X Visium and Xenium slides to show the model's contribution as a bridge between image and omic.

Response: Thank you for this valuable comment! As suggested, we tested the performance of Loki Align on aligning 10X Visium and Xenium slides of a human breast cancer sample to demonstrate its utility as a bridge between image and omics data. We generated a simulation data by performing rotations and translations on the Xenium data. Although OmiCLIP is a vision-transcriptome model, it was trained use only H&E image in the image decoder. It cannot directly align Xenium image (DAPI image) and Visium. Therefore, we used an alternative approach to this task which is described below. To perform the alignment, we first calculated transcriptomic embeddings for the Visium slide using the gene sentences derived from Visium transcriptomic data. For the Xenium slide, we created pseudo-Visium data by averaging gene expression values across pseudo-spots. These pseudo-Visium data were then used to calculate transcriptomic embeddings via the transcriptomic encoder of OmiCLIP. Finally, Loki Align was

applied to align the transcriptomic embeddings of the Xenium slide with those of the Visium slide. The alignment performance was quantified by measuring the distance between the Loki-aligned Xenium data and the ground truth, reflecting the consistency between the two transcriptomic datasets. The resulting distance between the aligned Xenium slide and the target Visium slide was 0.08 mm, demonstrating that Loki Align effectively aligns Visium and Xenium slides with high precision. These findings underscore the model's ability to integrate and align diverse transcriptomic data and have been added to the manuscript on lines 254-266.

Response Figure 16 (Revised Extended Data Fig. 6): Tissue alignment results on breast cancer sample using Loki Align. Left: Source Xenium ST data; Middle: Target Visium ST data; Right: Xenium ST data after Loki alignment.

Question 6. PLIP should also be used for benchmarking the retrieval task. Moreover, the benefits and significance of retrieval should be discussed in greater depth. For instance, demonstrating how the retrieved gene expression patterns map onto the queried WSI (whole slide image) would provide more tangible insights into the retrieval functionality.

Response: Thanks for this suggestion. We have now included PLIP for benchmarking the retrieval task in **revised Fig. 6c and d**. Results show that the PLIP model has a similar performance to OpenAI CLIP. We have also added PLIP to the evaluation for Loki Retrieve by calculating the rank of the correct pair using Recall@K, which is the proportion of the data correctly retrieved among the top-K quantile retrieved samples (suggested by Reviewer #2). The validation datasets include brain, heart, kidney, and breast tissue samples. The test dataset consists of various tissue samples from 4 studies (**Revised Supplementary Table 4**). Results demonstrated that Loki significantly outperformed both OpenAI CLIP and PLIP across all validation datasets. Specifically, Loki achieved Recall@5% of 0.125 and Recall@10% of 0.227 for brain (averagely 2.3-fold higher than OpenAI CLIP and 2.5-fold higher than PLIP), Recall@5% of 0.186 and Recall@10% of 0.291 for heart (averagely 3.2-fold higher than OpenAI CLIP and 3.1-fold higher than PLIP), Recall@5% of 0.173 and Recall@10% of 0.297 for kidney (averagely 3.2-fold higher than OpenAI CLIP and PLIP), and Recall@5% of 0.140 and Recall@10% of 0.240 for breast (averagely 2.6-fold higher than OpenAI CLIP and PLIP) (**Revised Fig. 6e**). On the test dataset, Loki further demonstrated substantial improvements,

achieving Recall@5% of 0.117 and Recall@10% of 0.208 (averagely 3.1-fold higher than OpenAI CLIP and 3.0-fold higher than PLIP) (**Revised Fig. 6e** and **Revised Supplementary Table 4**).

For the benefits and significance of retrieval, OmiCLIP is designed as a contrastive learning framework rather than a generative model. While it excels in embedding transcriptomic and histology data at the patch level, it does not inherently generate new data, such as reconstructing a WSI with detailed gene expression patterns. However, leveraging OmiCLIP's patch-level embeddings could serve as a powerful foundation for generative approaches. For instance, these embeddings could potentially be integrated into a diffusion model to reconstruct a more reliable WSI that accurately maps retrieved gene expression patterns. Such an approach would enable the generation of high-resolution WSIs enriched with spatial transcriptomic information, bridging the gap between retrieval functionality and tangible outputs. This possibility highlights OmiCLIP's flexibility as a foundational model, providing embeddings that can be extended to support downstream generative tasks, such as reconstructing WSIs in future studies. We have added these in the Discussion section on lines 680-690.

Dataset	Metric	Loki	OpenAI CLIP	PLIP	Fold change	Fold change
					(Loki versus OpenAI CLIP)	(Loki versus PLIP)
Validation brain dataset	Recall@5%	0.125	0.051	0.048	2.4	2.6
	Recall@10%	0.227	0.103	0.095	2.2	2.4
Validation heart dataset	Recall@5%	0.186	0.052	0.057	3.6	3.3
	Recall@10%	0.291	0.104	0.103	2.8	2.8
Validation kidney dataset	Recall@5%	0.173	0.052	0.053	3.4	3.3
	Recall@10%	0.297	0.100	0.097	3.0	3.0
Validation breast dataset	Recall@5%	0.140	0.049	0.050	2.9	2.8
	Recall@10%	0.240	0.100	0.096	2.4	2.5
Test dataset	Recall@5%	0.117	0.033	0.042	3.5	2.8
	Recall@10%	0.208	0.075	0.067	2.8	3.1

Response Figure 17 (Revised Fig. 6c-e): **a**, Image-to-transcriptomics retrieval similarity scores across the four validation datasets: CRC7K, WSSS4LUAD, LC25000, and PatchCamelyon using Loki, OpenAI CLIP, and PLIP respectively. In the box plots, the middle line represents the median, the box boundaries indicate the interquartile range, and the whiskers extend to data points within 1.5× the interquartile range. **b**, Image-to-transcriptomics retrieval similarity scores across the 8 in-house patient tissues: heart failure (HF), Alzheimer's disease (AD), metaplastic breast cancer (MPBC), and triple-negative breast cancer (TNBC), using Loki, OpenAI CLIP, and PLIP respectively. In the box plots, the middle line represents the median, the box boundaries indicate the interquartile range, and the whiskers extend to data points within 1.5× the interquartile range. **c**, Image-to-transcriptomics retrieval evaluation across four validation datasets and one test dataset using Loki, OpenAI CLIP, and PLIP, respectively. The top-K quantile most similar transcriptomics were retrieved. We report Recall@K for $K \in \{5\%, 10\%\}$ (Methods).

Question 7. New baselines should be included for Loki PredEx. Works like BLEEP (Spatially Resolved Gene Expression Prediction from Histology Images via Bi-modal Contrastive Learning) and mclSTExp (Multimodal contrastive learning for spatial gene expression prediction using histology images) have a similar clip-retrieval style, also scalable and already showed a higher performance than HisToGene. The authors should compare Loki with these methods.

Related to the last comment, I suggest adding Pearson correlation across spots as an additional evaluation metric, which would better reflect the accuracy of recovering spatial gene expression patterns.

Response: Thank you for this valuable comment! We have now included a detailed comparison of Loki PredEx with BLEEP and mclSTExp, alongside the previously evaluated methods (**Revised Extended Data Fig. 16**). Loki PredEx demonstrated superior performance, achieving the best results based on MSE scores in 28 out of 39 cases compared to Hist2ST, HisToGene, BLEEP, and mclSTExp (**Response Figure 18**). In addition, as suggested, we incorporated PCC across spots as an additional evaluation metric to better reflect the accuracy of recovering spatial gene expression patterns. Using this metric, Loki PredEx also outperformed other methods by ranking as the best in 16 out of 39 samples. These results showed the robustness of OmiCLIP in predicting spatial transcriptomic data across diverse datasets. We have incorporated it into the revised manuscript on lines 593-599, illustrated in **Response Figure 18**.

Response Figure 18 (Revised Extended Data Fig. 16): Comparison of ST gene expression prediction performances, represented by MSE and PCC on 39 normal heart tissues using Loki, Hist2ST, HisToGene, BLEEP, and mclSTExp, respectively. In the box plots, the middle line represents the median, the box boundaries indicate the interquartile range, and the whiskers extend to data points within 1.5× the interquartile range.

Question 8. I appreciate the authors on their ST-bank dataset construction and I think it would be a valuable resource to the community. However I only found a demo dataset in code and there is no availability in the manuscript to access the data. I think it should be released as a condition of publication.

Response: Thank you for acknowledging the value of the ST-bank dataset. We provided it on https://drive.google.com/drive/folders/1J15cO-pXTwkTjRAR-v-_nQkqXNfcCNn3?usp=sharing. We introduced the file information in “source_code/ README.md” file in Additional Review Material.

Question 9. An intuitive alternative to the current method is to combine existing single-cell foundation models with pathology foundation models. The authors should discuss why they chose to model omics data as text, rather than directly using gene expression values.

Response: Thank you for this insightful comment. In recent years, single-cell foundation models have shown significant improvement for tasks such as batch effect removal, drug response prediction, rare cell population detection, and perturbation prediction¹⁴⁻¹⁶. Combining existing single-cell and pathology foundation models is an intuitive alternative approach. However, we decided to choose the approach of modeling omics data as the text for its versatility and potential to bridge molecular and visual modalities effectively. By representing gene expression data as text, this approach leverages the power of NLP models to embed biological information into a high-dimensional space. This approach offers several advantages over directly using raw

gene expression values. First, text embeddings enable the seamless integration of omics data with various biological entities, such as pathways, functional annotations (Nature Methods, 2024 a)¹, and cell types (Nature Methods, 2024 b)², by linking textual representations across datasets. This opens opportunities to generalize the model's capabilities beyond tissue alignment and decomposition, making it adaptable to a broader range of biological tasks. Second, representing omics data as text aligns with the architecture of other multimodal foundation models, which provides the opportunity to further incorporate more modalities such as proteomics, metabolomics, and DAPI images into the same unified space. In contrast, raw gene expression values lack the flexibility for direct integration with such models and may require additional preprocessing or specialized architectures to achieve comparable performance. Third, the text-based foundation models have been trained on the billion-level tokens and provided solid text embeddings. For example, recent studies, such as GenePT³, have demonstrated the effectiveness of using large language models to generate gene embeddings from textual descriptions, achieving comparable or superior performance to models trained on extensive gene expression datasets. This supports our approach of utilizing text-based embeddings to capture rich biological information efficiently. Together, while existing single-cell foundation models are optimized for specific tasks, our method aims to create a unified framework that can generalize across modalities and tasks.

We believe this choice enhances the versatility and robustness of OmiCLIP and Loki, enabling the models to serve as bridges between visual and molecular data. This discussion has been added to the manuscript on lines 697-718 for further clarity.

Question 10. I recommend the authors add the introduction of tasks in supplementary notes to the introduction of the main manuscript. This would help better contextualize the generalizable contributions of this work and make it easier for the readers to grasp the scope of the research.

Response: Thank you for this valuable suggestion. As suggested, we have integrated the introduction of tasks in supplementary notes to the introduction of the main manuscript. The revised introduction has been updated on lines 81-83.

Remarks on code availability:

I installed the environment, ran the code on Codeocean, and also reviewed the code in the supplementary material. Codes for downstream applications are executable and documented, but are without the pre-training one.

Response: Thank you for the comment. In the re-submission, we have included pre-train weights on:

<https://drive.google.com/file/d/1UTJKAqoiUT55h9sA1Ee8iJeOcINLBvWD/view?usp=sharing>.

Reference:

1. Hu, M., *et al.* Evaluation of large language models for discovery of gene set function. *Nature methods*, 1-10 (2024).
2. Hou, W. & Ji, Z. Assessing GPT-4 for cell type annotation in single-cell RNA-seq analysis. *Nature Methods*, 1-4 (2024).
3. Chen, Y. & Zou, J. Simple and effective embedding model for single-cell biology built from ChatGPT. *Nature Biomedical Engineering*, 1-11 (2024).
4. Hao, Y., *et al.* Dictionary learning for integrative, multimodal and scalable single-cell analysis. *Nature biotechnology* **42**, 293-304 (2024).
5. Wolf, F.A., Angerer, P. & Theis, F.J. SCANPY: large-scale single-cell gene expression data analysis. *Genome biology* **19**, 1-5 (2018).
6. McCarthy, D.J., Campbell, K.R., Lun, A.T. & Wills, Q.F. Scater: pre-processing, quality control, normalization and visualization of single-cell RNA-seq data in R. *Bioinformatics* **33**, 1179-1186 (2017).
7. Caliński, T. & Harabasz, J. A dendrite method for cluster analysis. *Communications in Statistics-theory and Methods* **3**, 1-27 (1974).
8. Neubeck, A. & Van Gool, L. Efficient non-maximum suppression. in *18th international conference on pattern recognition (ICPR'06)*, Vol. 3 850-855 (IEEE, 2006).
9. Tasic, B., *et al.* Shared and distinct transcriptomic cell types across neocortical areas. *Nature* **563**, 72-78 (2018).
10. Schuhmann, C., *et al.* Laion-5b: An open large-scale dataset for training next generation image-text models. *Advances in Neural Information Processing Systems* **35**, 25278-25294 (2022).
11. Jain, P. & Kar, P. Non-convex optimization for machine learning. *Foundations and Trends® in Machine Learning* **10**, 142-363 (2017).
12. Biancalani, T., *et al.* Deep learning and alignment of spatially resolved single-cell transcriptomes with Tangram. *Nature methods* **18**, 1352-1362 (2021).
13. Levine, D., *et al.* Cell2sentence: Teaching large language models the language of biology. *BioRxiv*, 2023.2009. 2011.557287 (2023).
14. Theodoris, C.V., *et al.* Transfer learning enables predictions in network biology. *Nature* **618**, 616-624 (2023).
15. Hao, M., *et al.* Large-scale foundation model on single-cell transcriptomics. *Nature Methods*, 1-11 (2024).
16. Cui, H., *et al.* scGPT: toward building a foundation model for single-cell multi-omics using generative AI. *Nature Methods*, 1-11 (2024).

Summary: We would like to thank the reviewers for all the suggestions and constructive comments. We appreciate that all reviewers acknowledged the improvements in our work. For example:

- *“The authors have addressed most comments provided.” (Reviewer #1)*
- *“Overall, I am very happy with the progress made on the manuscript. The revisions clearly demonstrate that substantial effort was invested, resulting in a much-improved presentation and analysis.” (Reviewer #2)*
- *“The analysis on Loki Align has been significantly enhanced by the addition of extra metrics and the inclusion of CPD as a baseline, which substantially improves this section.” (Reviewer #2)*
- *“I would like to thank the authors for their thorough responses to my comments; they have addressed most of my concerns. I am also pleased to see that the model weight and pre-training data will be made available to the community.” (Reviewer #3)*
- *“I think the paper is now much stronger” (Reviewer #3)*

For the rest of the concerns, we provided detailed point-by-point responses below.

Response to Reviewer #1:

Remarks to the Author:

The authors have addressed most comments provided while some comments are still not adequately addressed like comment 3. Specifically, the authors have not answered the reason why the results appear unstable, with some seemingly random results.

Response: Thank you for pointing out the perceived instability in our results. We believe several factors contribute to this variability, including biological heterogeneity within tissue patches, technical variability in library preparation and sequencing, and instances of low gene expression that challenge consistent pattern recognition. Each of these elements can introduce noise or bias, making certain outcomes appear random in some cases.

Remarks on code availability:

The authors has provided the codes via private link and jupyter notebook files. README file with enough instructions for installing and running the application was provided.

I haven't run the code due to the lacking of GPU resources. However, it appears the code and data are reliable and reproducible.

Additionally, while the author provided private link, it is still recommended to make code and data public.

Response: Thank you very much for your valuable comments. We plan to release the full codes, platform, pre-trained model, website, and database ST-bank to the public upon acceptance of the manuscript, ensuring accessibility for the broader research community.

Response to Reviewer #2:

Remarks to the Author:

Overall, I am very happy with the progress made on the manuscript. The revisions clearly demonstrate that substantial effort was invested, resulting in a much-improved presentation and analysis.

Response: Thank you very much for your encouraging and constructive comments. We appreciate your recognition of the effort invested in revising the manuscript and are pleased to hear that these improvements have led to a clearer and more robust presentation and analysis. Your feedback has been instrumental in guiding us toward a more comprehensive approach.

Loki Align

The analysis on Loki Align has been significantly enhanced by the addition of extra metrics and the inclusion of CPD as a baseline, which substantially improves this section. However, I am curious as to why only two principal components were used in the PCA analysis. An explanation of this decision would help clarify whether the chosen dimensionality captures the essential variance or if additional components might provide further insights.

Response: Thank you very much for your valuable comments and for recognizing the improvements made in this section. In the case study of human ovarian carcinosarcoma, we observed that the first principal component (PC) captures 84.6% of the variance in the transcriptome features and 83.3% in the image features. Subsequent components, such as the third PC, each contributed a comparatively small fraction (1.6% and 2.5% of the variance for transcriptome and image features, respectively). Consequently, focusing on just the first two principal components was sufficient to capture the essential variance and dominant trends in both data modalities.

Additionally, Loki Align inherently operates in two spatial dimensions (x and y coordinates). Restricting the dimensionality to two principal components offers a balanced way to incorporate both spatial location and the key transcriptome or image signals, without diluting the overall contribution of each source. This approach reduces unnecessary model complexity, streamlines computation, and simplifies visualization—advantages that become especially important when dealing with large datasets or when rapid iteration is needed.

We acknowledge that, in other contexts or datasets, higher-order principal components might reveal subtle patterns. However, given the high proportion of variance captured by the first two components in this particular analysis, we found that additional components did not significantly enhance interpretability or improve alignment performance. We remain open to exploring additional components in future studies if they appear to offer meaningful insights.

Additionally, the final sentence of this section—stating that “Altogether, Loki Align serves as a generalized cross-modality alignment tool for spatial tissue analysis, reducing costs associated with spatial transcriptomics sequencing”—feels somewhat

disconnected from the main motivation. This should be changed to ensure that it aligns clearly with the overall objectives of Loki Align.

Response: Thank you very much for your valuable comments. We will rewrite the final sentence of this section:

“Altogether, by directly addressing the challenges posed by spatial distortions and biological variability, Loki Align enables the accurate alignment of multiple H&E images and ST sections, thereby supporting advanced 3D reconstructions of tissue organization, particularly for cross-modality studies that combine H&E imaging and ST data.”

Loki Decompose

The conclusion for Loki Decompose remains somewhat weak. It would be useful to more explicitly describe the scenarios in which one might prefer using Loki Decompose for extracting cell type proportions from images, particularly in cases where acquiring the transcriptomic profile of the spots is not necessary. Clarifying these use cases would enhance the reader’s understanding of its practical applications.

Response: Thank you very much for your valuable comments. We will add below in the discussion.

“Loki Decompose is particularly valuable when the high costs of sequencing make obtaining full transcriptomic profiles impractical. In these scenarios, researchers may opt to preselect, screen, or batch process samples using imaging alone to estimate cell type proportions. This approach provides a cost-effective alternative that can quickly yield essential information, especially in exploratory studies or large-scale screenings where detailed molecular profiling of every spot is not necessary.”

Loki Retrieve

The new analysis presented for Loki Retrieve raises a couple of points that need further clarification. First, what is the random baseline for Recall@5% in this setting? Including this information would allow readers to better contextualize the performance of Loki Retrieve relative to competing methods such as OpenAI CLIP and PLIP.

Response: Thank you very much for your valuable comments. We will add the random baseline for Recall@5% and Recall@10% in this setting to better contextualize the performance of Loki Retrieve relative to OpenAI CLIP and PLIP.

Dataset	Metric	Loki	OpenAI CLIP	PLIP	Random	Fold change (Loki versus OpenAI CLIP)	Fold change (Loki versus PLIP)
Validation brain dataset	Recall@5%	0.125	0.051	0.048	0.052	2.4	2.6
	Recall@10%	0.227	0.103	0.095	0.101	2.2	2.4
Validation heart dataset	Recall@5%	0.186	0.052	0.057	0.049	3.6	3.3
	Recall@10%	0.291	0.104	0.103	0.098	2.8	2.8
Validation kidney dataset	Recall@5%	0.173	0.052	0.053	0.053	3.4	3.3
	Recall@10%	0.297	0.100	0.097	0.101	3.0	3.0
Validation breast dataset	Recall@5%	0.140	0.049	0.050	0.047	2.9	2.8
	Recall@10%	0.240	0.100	0.096	0.094	2.4	2.5
Test dataset	Recall@5%	0.117	0.033	0.042	0.025	3.5	2.8
	Recall@10%	0.208	0.075	0.067	0.067	2.8	3.1

Response Figure 1: Image-to-transcriptomics retrieval evaluation across four validation datasets and one test dataset using Loki, OpenAI CLIP, and PLIP, respectively, with random baseline. The top-K quantile most similar transcriptomics were retrieved. We report Recall@K for $K \in \{5\%, 10\%\}$ (Methods).

Secondly, while I understand that Loki Retrieve is not a generative model, the manuscript would benefit from a clearer discussion of its biological motivation. Specifically, why is retrieving similar images a valuable downstream task from a biological standpoint, especially in contrast to Loki PredEx? Elaborating on this aspect could help underscore the relevance and novelty of the approach.

Response: Thank you very much for your valuable comments. We will add the below discussion in the Discussion section:

“Loki Retrieve is designed to leverage a curated set of annotated reference images, providing users with reliable ground truth comparisons, especially when training data of prediction models like Loki PredEx is difficult to obtain. Additionally, the reference-based approach allows researchers to validate and interpret their findings with greater confidence.”

Supplementary Figures

In Supplementary Figure 2, I was surprised to see many T cells classified as B cells, given that these are distinct cell types in the UMAP. It would be helpful to provide some intuition or discussion on why this misclassification occurs, highlighting any challenges inherent in distinguishing these cell types. Moreover, both Supplementary Figures 1 and 2 would benefit from a revised color scheme that more clearly differentiates B cells from T cells, as these are key cell types that warrant clear visual separation.

Response: Thank you very much for your valuable comments. The observed misclassification of T cells as B cells potentially arises from overlapping immune cell features, particularly when key marker genes are expressed at low levels or are temporarily downregulated. In such cases, the model may rely on shared immune signatures, making it challenging to distinguish these cell types accurately in the UMAP space. We have discussed this limitation in the manuscripts.

We will revise the color scheme in Supplementary Figures 1 and 2 to ensure that B cells and T cells are visually distinct. We believe this change will provide a clearer depiction of the different cell populations, allowing readers to easily identify these critical cell types. Moving forward, we

also plan to explore additional marker genes and refined classification thresholds to further improve cell-type resolution in our analyses.

Response Figure 2: UMAP visualizations of OmiCLIP/OpenAI CLIP transcriptomic embeddings of scRNA-seq data. **a**, UMAP representation of the OmiCLIP transcriptomic embeddings of prostate cancer multi-sample scRNA-seq datasets. Each dot represents a cell. **b**, same as in **(a)** with OpenAI CLIP embeddings. **c**, The projection of the actual and predicted cell type labels to the original UMAP embedding for the prostate cancer dataset. 1% randomly sampled cells were used for training and the rest were plotted on the UMAP embedding.

Extended Data Figures

The modifications to Extended Data Fig. 2 are a welcome improvement. However, I suggest that Extended Data Fig. 1b be reformatted to more closely resemble Extended Data Fig. 2b, which effectively shows the UMAPs before and after contrastive learning using ground truth labels for coloring. This change would provide a more informative visualization and facilitate a direct comparison of the effects of contrastive learning.

Response: Thank you very much for your valuable comments. We will reformat Extended Data Fig. 1b which shows the UMAPs before and after contrastive learning, to provide a more informative visualization and facilitate a direct comparison of the effects of contrastive learning.

Response Figure 3: Image and transcriptomic embeddings of the lung, kidney cancer, healthy heart, and Myocardial Infarction (MI) heart samples. Each row corresponds to a WSI and showcases information from two modalities. The first column are H&E images showing tissue morphology; the second column are the heatmaps of ST data with the colors indicating the cell types; the third column are the UMAP of image embeddings colored by cell types before and after contrastive learning; the fourth column are the UMAP of transcriptomics embeddings colored by cell types before and after contrastive learning.

Cancer Patient Risk Stratification

The section on cancer patient risk stratification appears to be biologically significant. Considering its potential impact, it might be worth discussing whether this section should remain in the supplementary material or be promoted to the main text, as it could provide valuable insights that are more compelling than some of the other downstream tasks currently highlighted.

Response: Thank you for highlighting the biological significance and potential impact of the cancer patient risk stratification section. We agree that this analysis is valuable; however, our decision to keep it in the supplementary material is primarily driven by our desire to maintain the main text's focus on the core methodology and its broad applicability. We also aim to manage the length and clarity of the main manuscript, ensuring that readers can easily follow the central narrative without being overwhelmed by additional details.

That said, we recognize the value of these risk stratification insights. For balance, we prefer to reference this section more prominently in the main text, ensuring that interested readers are aware of the supplementary material and can readily access its findings. We believe this approach allows us to highlight the methodological framework in the main paper while still

providing a dedicated space for a deeper discussion of the risk stratification results for those who wish to explore them further.

In summary, while the manuscript has improved considerably, addressing these specific points would further enhance its clarity, impact, and overall scientific contribution.

Response: Thank you for your thorough review and insightful suggestions. We truly appreciate the time you took to help us refine our manuscript. Your specific points will guide our further improvements, ensuring that our work is as clear, impactful, and scientifically rigorous as possible.

Remarks on code availability:

The additional links for the source code gave me confidence that it will be possible to build off of Loki and reproduce their results. While I did not install the code, a README is available and has detailed instructions. Furthermore, the inclusion of python notebooks for many of the downstream tasks serve as good tutorials.

Response: Thank you for your encouraging feedback. We're glad that the additional source code links, detailed README, and accompanying Python notebooks instill confidence in the reproducibility and extensibility of Loki.

Reviewer #3:

Remarks to the Author:

I would like to thank the authors for their thorough responses to my comments; they have addressed most of my concerns. I am also pleased to see that the model weight and pre-training data will be made available to the community.

I think the paper is now much stronger, but I have a few additional comments.

Response: Thank you very much for your positive feedback. We are pleased to hear that our revisions have addressed most of your concerns, and we appreciate your recognition of our efforts to make both the model weights and pre-training data openly available. Below is a point-by-point response to the additional comments.

1. In Revised Extended Data Fig. 17, the Loki PredEx module is missing. Could the authors please include it?

Response: Thank you very much for pointing this out. We truly appreciate your interest in the Loki PredEx module! We will include the Loki PredEx module in Revised Extended Data Fig. 17.

	Function	Software	Fine-tune
Loki Align	H&E image to ST alignment	Loki only	Suggested
Loki Align	ST to ST alignment	Loki PASTE ...	No need
Loki Decompose	scRNA-seq to H&E image	Loki only	Suggested
Loki Decompose	scRNA-seq to ST mapping	Loki Tangram ...	No need
Loki Annotate	Tissue annotation by Bulk RNA-seq	Loki only	No need
Loki Annotate	Tissue annotation by marker genes	Loki only	No need
Loki Retrieve	H&E image-to-ST retrieval	Loki only	No need
Loki PredEx	ST gene expression prediction by H&E image	Loki HisToGene ...	Suggested

Response Figure 4: Summary of the fine-tuning settings for downstream tasks. Recommendation settings for downstream tasks.

2. In the Loki align part, why the two modalities embeddings didn't outperform the single one? I suggest adding an explanation for this result.

Response: Thank you for your suggestion. While combining two modalities can theoretically capture more comprehensive information, in practice, it may also introduce noise or misalignment that overshadows the potential benefits. Additionally, if one modality has a stronger signal, it can dominate the overall outcome, resulting in minimal or no performance gains when both are combined. We will add these to the Discussion.

3. On line 1005, the authors state, "The OmiCLIP model was fine-tuned on the training set for 10 epochs." However, line 606 mentions that "Loki PredEx avoids these resource-intensive training needs, providing a more efficient alternative." Since OmiCLIP is a foundation model, it would typically require more resources to train compared to smaller models. Could the authors clarify this?

Response: Thank you for bringing this up. Although OmiCLIP is indeed a large foundation model, in our work we only fine-tuned it using pre-trained weights from a sizable histopathology dataset, which substantially reduces the training overhead. By contrast, some smaller transformer-based models (e.g., HisToGene and Hist2ST) require training from scratch, and others (e.g., mclSTExp and BLEEP) use less specialized pre-trained encoders—both of which can still be resource-intensive. Loki PredEx, on the other hand, avoids most of these training steps altogether, offering a more efficient alternative. We will modify the conclusion to state that "Loki PredEx avoids these resource-intensive training needs, providing a more efficient alternative based upon the use of pre-trained weights."

4. It is difficult to compare the performance across methods from Extended Data Fig. 16. It would be helpful to summarize the performance across all samples in a single figure for better comparison. Additionally, Since there is not a significant performance gain in the PredEx module, I believe it would be valuable for the authors to provide some insights into the success and failure cases of OmiCLIP.

Response: Thank you very much for your valuable comments. We will summarize the performance across all samples in a single figure for better comparison using MSE and PCC respectively. The use of diverse pre-training data and a large model architecture potentially results in the successful performance of Loki PredEx. These factors may enable the model to capture a broad range of features and subtle variations, which can help reduce the failure rate and improve robustness. We will add this discussion in the Discussion section.

Response Figure 5: Summarized comparison of ST gene expression prediction performances, represented by MSE and PCC respectively across all samples using Loki, HisToGene, mclSTExp, BLEEP, and Hist2ST respectively. In the box plots, the middle line represents the median, the box boundaries indicate the interquartile range, and the whiskers extend to data points within 1.5× the interquartile range.

5. Since fine-tuning is necessary for several of the Loki functions, I suggest the authors make the code more user-friendly. Upon reviewing the code, I noticed that the fine-tuning code is available via the command line but not integrated into the API.

Response: Thank you for your comment. We recognize that having an API-based approach can be convenient for some users. However, our experience has shown that command-line fine-tuning offers greater flexibility and control (adapted from scGPT Nature Methods, 2024 and Prov-GigaPath Nature, 2024), especially for advanced workflows where researchers want to customize parameters in a more granular way. We do provide a detailed notebook and comprehensive documentation for the fine-tuning process to ensure its accessibility for those who need it. We believe this balance—keeping specialized functionality in the command line while providing a clear guide—best serves both casual and power users. Nonetheless, we are open to revisiting this approach if we see broader demand for an API-based fine-tuning solution.